# Rainfall intensity estimations based on degradation characteristics of images taken with commercial cameras

Akito Kanazawa[1], Taro Uchida[2]

[1]National Institute for Land and Infrastructure Management, Tsukuba, Japan
[2]Faculty of Life and Environmental Sciences, University of Tsukuba, Tsukuba, Japan

*Correspondence to*: Akito Kanazawa (kanazawa.akito@gmail.com)

**Abstract.** Camera-based rainfall observation is a useful technology that contributes to the densification of rainfall observation networks because it can measure rainfall with high spatio-temporal resolution and low cost. To verify the applicability of existing theories, such as computer vision and meteorological studies, to static weather effects caused by rain

in outdoor photography systems, this study proposed relational equations representing the relationship between image information, rainfall intensity, and scene depth by linking the theoretically derived rainfall intensity with a technique proposed in the computer vision field for removing static weather effects. This study also proposed a method for estimating rainfall intensity from images using those relational equations. Since the method only uses the camera image taken of the background over a certain distance and background scene depth information, it is a highly versatile and accessible method.

The proposed equations and the method for estimating rainfall intensity from images were applied to outdoor images taken by commercial interval cameras at the observation site in a mountainous watershed in Japan. As a result, it was confirmed that transmission calculated from the image information decreases exponentially according to the increase in rainfall intensity and scene depth, as assumed in the proposed equations. Furthermore, rainfall intensity can be estimated from the image using the proposed relational equations. On the other hand, the calculated extinction coefficient tended to be

overestimated at small scene depth. Although there are several problems at present that need to be resolved for the technology proposed in this study, this technology has the potential to help the development of a camera-based rainfall observation technology that is accurate, robust, versatile, and accessible.

## 1 Introduction

The water cycle regulates local, regional, and global climate change, and precipitation is an important component of this

cycle (Eltahir and Bras, 1996). Reliable precipitation data are therefore critical for local, regional, and global water resource management and weather, climate, and hydrologic forecasting (Jiang et al., 2019; Sun et al., 2018). Rainfall is difficult to observe adequately due to large spatial and temporal variations (Kidd et al., 2016). In order to properly observe such variations, a dense observation network is necessary on a fine temporal and spatial scale. Especially in mountainous areas where flash floods and debris flow occur, rainfall should be measured on fine spatial and temporal scales for effective early

warning against these disasters (e.g., Kidd et al., 2016). Currently, rainfall data are mainly obtained from ground observation such as rain gauges, and remote sensing such as weather radar and satellites. Rainfall data obtained from ground observation are used for both direct measurement and indirect measurement calibrations. However, rainfall data is often limited in terms of spatio-temporal resolution due to the sparseness of the ground observation networks (Notarangelo et al., 2021). In addition, it has been noted that near-real-time rainfall data has reasonable coverage in Europe and East Asia, including Japan, but

observation sites are sparse in other regions (Kidd et al., 2016), and due to the high cost of observation, a high-resolution, ground-level rainfall monitoring network still has limited use (Jiang et al., 2019). Therefore, innovative methods to achieve higher density in the ground-level rainfall observation network have been the focus of recent hydrological research (Tauro et al., 2018).

As an initiative to overcome the issues mentioned above, techniques have been proposed to build sensors using low-cost

equipment not used for its intended use and to combine a variety of not fully utilized technologies to make opportunistic observations (Tauro et al., 2018). For these techniques, an approach has been adopted in the form of aggregating data obtained from a high-density network built using a large number of low-cost sensors that are less accurate (Notarangelo et al., 2021). While such an approach is not as accurate as conventional rain gauges in most cases, it could provide valuable additional information when combined with conventional techniques (Tauro et al., 2018). Haberlandt and Sester (2010) and

Rabiei et al. (2016) reported that the idea of considering moving vehicles as rain gauges and windshield wipers as sensors to detect rainfall may enable better areal rainfall estimation than using several accurate rain gauges by making numerous observations, even if they are somewhat inaccurate. The microwave link in the cellular phone communication network, which focuses on the relationship between rain attenuation of electromagnetic signals transmitted from one cellular tower to another and the average rainfall along the path, has been proposed as a promising new rainfall measurement technology

(Leijnse et al., 2007; Messer et al., 2006; Overeem et al., 2011; Rahimi et al., 2006; Tauro et al., 2018; Upton et al., 2005; Zinevich et al., 2009). It has been indicated that such opportunistic sensors have the potential to be utilized in geographic regions where the density of conventional rainfall measurement devices is low, namely mountainous areas and developing countries (Uijlenhoet et al., 2018). Further, since a large number of video monitoring cameras have been installed outdoors in recent years for security and safety reasons, techniques have been reported to use these cameras to estimate the

environment and weather of scenes (Jacobs et al., 2009). As techniques that use cameras to monitor surrounding conditions, techniques to observe river levels and flow rates (Gilmore et al., 2013; Muste et al., 2008; Tauro et al., 2018), and rainfall (Allamano et al., 2015; Dong et al., 2017; Jiang et al., 2019; Yin et al., 2023; Zheng et al., 2023) have also been reported, and are attracting great interest in the hydrologic field. In addition, such a camera-based technique for understanding the surrounding situation has the potential to serve as a sensor that can measure multiple types of physical quantities with a

single camera and is a very reasonable and meaningful technique for obtaining various types of information all at once. Since rainfall measurement using cameras enables high spatio-temporal resolution and extremely low-cost measurement, it is possible to say that it has opened a novel avenue toward higher-density rainfall observation (Tauro et al., 2018).

The development of camera-based rain gauges requires clarification of the effects of rainfall on images. The effects of adverse weather conditions, such as rainfall, on images have conventionally been studied mainly in the fields of computer vision and image processing (Narasimhan and Nayar, 2002). In outdoor photography systems used for monitoring, navigation, and other purposes, various algorithms such as feature detection, stereo correspondence, tracking, segmentation, and object recognition are used and these algorithms require visual clues and feature information (Garg and Nayar, 2007). Since the adverse weather conditions lead to the loss of those visual clues and feature information due to the effects of poor visibility, the objective of studies was to remove the effects of adverse weather conditions on the images and obtain clear images (Jiang et al., 2019; Tripathi and Mukhopadhyay, 2014). On the other hand, in reference to such image processing techniques, studies on camera-based rain gauges quantified the degree of performance degradation due to adverse weather in outdoor photography systems as a change in weather conditions (Garg and Nayar, 2007). Such studies broadly categorize adverse weather into static weather, such as fog and haze, and dynamic weather, such as rain and snow, based on physical properties and types of visual effects (Garg and Nayar, 2007). In the case of static weather, the constituent water droplets are small, ranging from 1 to 10 μm, and cannot be detected individually by a camera. The intensity produced in the pixel is therefore due to the cohesive effect of the numerous water droplets within the pixel's solid angle (Garg and Nayar, 2007). Accordingly, studies have been conducted to represent static weather and remove the effects of static weather from images by using models of atmospheric scattering such as direct attenuation and airlight (Narasimhan and Nayar, 2002, 2003). In the studies on removing static weather effects from images, methods based on priors from natural image statistics have conventionally been used (Fattal, 2008; He et al., 2011; Tan, 2008). Recently, deep machine learning-based methods that extract image features from a large amount of learning data have been adopted (Qin et al., 2020; Shao et al., 2020; Zhou et al., 2021). On the other hand, in dynamic weather, water droplets are composed of particles 1,000 times larger than in static weather, ranging from 0.1 to 10 mm, and individual particles are visible to cameras. For this reason, the image processing research to remove dynamic weather effects has primarily studied techniques to extract rain by discriminating water droplets that appear as rain streaks from other backgrounds, and previous studies on camera-based rain gauges are also utilizing such techniques (Bossu et al., 2011; Garg and Nayar, 2007; Luo et al., 2015). A lot of deep machine learning-based methods have been proposed in recent years as with the trend of studies about static weather effect (e.g., Lin et al., 2020; Lin et al., 2022; Yin et al., 2023; Zheng et al., 2023).

In the previous studies, static and dynamic weather have been treated as separate themes because of the different characteristics of their effects on images. In particular, rain has been studied primarily as a dynamic weather topic (Allamano et al., 2015; Dong et al., 2017; Jiang et al., 2019; Yin et al., 2023; Zheng et al., 2023). However, the following practical challenges remain in these studies that treat rainfall as dynamic weather. They are effective only for static backgrounds of outdoor photography (Allamano et al., 2015), require special equipment (Dong et al., 2017), need to use video rather than still images to estimate rainfall intensity (Jiang et al., 2019), and need for a variety of rainfall images and corresponding rainfall intensity value data in advance to train the deep learning model (Yin et al., 2023; Zheng et al., 2023). In particular, it

has been pointed out that the limitation of deep machine learning-based methods is the lack of training data rather than the design of network structure and learning manners (Wang et al., 2021; Yan et al., 2023).

On the other hand, even if the absolute size of raindrops is constant within the camera's angle of view, the size of raindrops in the image varies with their distance from the camera. In particular, raindrops over a certain distance from the camera induce a visual effect as if they were in static weather conditions, because their fall distance within the camera's exposure time is sufficiently small compared to the pixel size that the camera's sensor cannot detect individual raindrops. In fact, it has been pointed out that rain streaks over a certain distance from the camera accumulate on the image and appear as fog (Garg and Nayar, 2007; Li et al. 2018; Li et al., 2019). This implies that rain causes static weather effects. Such raindrops over a certain distance from the camera are likely to induce static weather effects when the camera is mainly capturing a relatively undisturbed background such as rivers, scenery, trees. On the other hand, in the case of a disturbed background with people, animals, or traffic moving around, the static weather effects of raindrops may be difficult to discern because the original background may be disturbed by their movement. Thus, in an outdoor photography system that captures a relatively undisturbed background over a certain distance, not only the dynamic weather effects caused by rain but also the static weather effects caused by rain may be apparent in the images. In Japan, many cameras have been installed by public organizations to monitor watershed conditions with an angle of view that allows the viewer to see into the background at a certain distance for disaster prevention purposes. In other words, it is easy to obtain images that show static weather effects. Therefore, to utilize more images effectively, we construct a method to measure rainfall intensity using static weather effects from such images that are not intended for rainfall measurement but for monitoring watershed conditions.

So far, not enough is known about the details of the static weather effects caused by rain. Therefore, the main objective of this study is to verify the applicability of existing theories, such as computer vision and meteorological studies, to static weather effects caused by rain in outdoor photography systems. In this study, we analyzed the effects of rainfall intensity on the appearance of the background. Using the extinction coefficient as information source, we linked the technique of removing static weather effects reported in many computer vision studies with the theory of rainfall intensity expressed in atmospheric radiology and meteorology. We then proposed equations for the relationship between image information, rainfall intensity, and the distance from the camera to the background, hereinafter referred to as scene depth. Using the proposed equations, rainfall observations can be performed with an image of the background at a certain distance and information on the scene depth to the background, even if the image is not intended for rainfall observations. Therefore, by applying the outdoor images taken by commercial interval cameras at observation sites in mountainous watersheds in Japan and rainfall observations to the proposed relational equations, the relationship between image information, rainfall intensity, and scene depth was analyzed, and the validity of the extinction coefficient obtained from the images was verified. Furthermore, we also attempted to estimate rainfall intensity using the proposed relational equations. The estimation of rainfall intensity used over 3,000 images from rain events, and this data size is a unique aspect of this study.

This paper is structured as follows. Section 2 describes the proposed relational equations for the relationship between image information, rainfall intensity, and scene depth. Section 3 describes the outdoor observations and the processing of the

130 captured images. Section 4 presents the results of observations, image processing, and analysis. Section 5 discusses the extinction coefficient and rainfall intensity estimated from the image information, and section 6 describes the conclusion.

## 2 Relational equations for the relationship between image information, rainfall intensity, and scene depth

### 2.1 Image information and extinction coefficient

Effects of static weather are mainly caused by two scattering phenomena: "direct attenuation" and "airlight" (Fattal, 2008;
He et al., 2011; Narasimhan and Nayar, 2002, 2003; Tan, 2008). "Direct attenuation" is the attenuated light received by the camera from the background along the line of sight, caused by the scattering of light by particles such as water droplets in the atmosphere. "Direct attenuation" reduces the contrast of a scene (Tripathi and Mukhopadhyay, 2014). "Airlight" is the total amount of environmental illumination reflected into the line of sight by atmospheric particles, typically direct and diffuse radiation from the sun interacting with the atmosphere in the case of daytime outdoors. "Airlight" results in a shift in
color (Tripathi and Mukhopadhyay, 2014). Static weather effects can be represented as a function of the scene depth and vary spatially on a single image (He et al., 2011; Tripathi and Mukhopadhyay, 2014). In the case of static weather, since the size of constituent particles such as water droplets is large compared to the wavelength of light, the "scattering coefficient", which represents the ability of a unit volume of atmosphere to scatter light in all directions, is not dependent on wavelength. For this reason, all wavelengths are equally scattered, giving the appearance of a whitish fog (Narasimhan and Nayar, 2003).
Therefore, the static weather effect that appears on the image by rainfall can be considered as image whitening, where the luminance increases and contrast decreases, depending on rainfall intensity and scene depth.

Many studies on computer vision have reported techniques for removing static weather effects from images (Fattal, 2008; He et al., 2011; Tan, 2008). In these studies, the effect of a hazy background due to fog or haze is represented by the following Image Degradation Model, using Koschmieder's model, which shows the relationship between visibility and atmospheric
extinction coefficient (Fattal, 2008; Koschmieder, 1924).

$$I(x) = J(x)t(x) + A\big(1 - t(x)\big) \tag{1}$$

Where $I$ is observed intensity, $J$ is scene radiance, $A$ is global atmospheric light, and $t$ is transmission, which represents the ratio of light that reaches the camera without being scattered. $x$ indicates the pixel position. $A$ is independent of $x$ and is generally constant in a single image (Tan, 2008). Eq. (1) is defined on the three RGB color channels. $I(x)$, $J(x)$, and $A$ are
155 three-dimensional RGB vectors and are represented by integer pixel intensity. $t(x)$ is scalar between 0 and 1. These four variables have no units.

In Eq. (1), the right-hand side $J(x)t(x)$ is direct attenuation, and $A(1-t(x))$ is airlight. Direct attenuation represents the attenuation of scene radiance by the medium in the air, while airlight represents light scattered by myriad particles suspended in the atmosphere.

If the atmosphere is uniform, transmission $t$ is expressed as follows.

$$t(x) = \exp(-\beta d(x)) \tag{2}$$

Where $d$ (m) is scene depth. $x$ indicates the pixel position as in Eq. (1).

$\beta$ (m$^{-1}$) is called the atmospheric extinction coefficient and represents the ability of the atmosphere to dissipate light in a unit volume of the atmosphere. Extinction refers to the combined effect of light scattering and absorption. In this paper, the terms extinction and scattering are used synonymously because water absorbs virtually no light in the visible light wavelength range.

Equation (2) shows that transmission attenuates exponentially according to the increase in scene depth, subject to the effect of the extinction coefficient. The principle is based on Beer-Lambert law, which means that as light passes through matter, in this case transparent atmosphere, its intensity attenuates exponentially.

The following is a variant of Eqs. (1) and (2).

$$\beta = -\frac{\log_e(t(x))}{d(x)} \tag{3}$$

$$t(x) = \frac{A - I(x)}{A - J(x)} \tag{4}$$

*Where* $A - J(x) \neq 0$, *and* $0 \leq t(x) \leq 1$

## 2.2 Rainfall intensity and extinction coefficient

With the theory of atmospheric radiation, the extinction coefficient under rainfall conditions can be expressed as follows using the raindrop diameter, the particle size distribution of raindrops, and extinction efficiency (Grabner and Kvicera, 2011).

$$\beta = \int_0^\infty \frac{\pi D^2}{4} N(D) Q dD \tag{5}$$

Where $D$ (m) is the raindrop diameter and $N(D)$ (m$^{-3}$) is the particle size distribution of raindrops. $D^2/4$ represents the surface area of raindrops projected in the optical path direction. $Q$ is called extinction efficiency and is a dimensionless parameter that expresses the ratio of the extinction cross-sectional area of the raindrop to the geometric cross-sectional area of the raindrop. The extinction cross-sectional area is the quantity that expresses the intensity of extinction of a single particle with the dimension of area. Under the Mie scattering theory, the extinction efficiency $Q$ is expressed as 2, given the relationship between raindrop size and the wavelength of visible light (Chylek, 1977; Uijlenhoet et al., 2011).

Since the particle size distribution of raindrops is known to be related to rainfall intensity (Marshall and Palmer, 1948), the extinction coefficient can be expressed using rainfall intensity as follows.

$$\beta = 5.80 \times 10^{-5} \pi Q R^{0.63} \tag{6}$$

Where $R$ (mm h$^{-1}$) is rainfall intensity. The detailed derivation process of Eq. (6) is described in Appendix A.

As shown in Appendix A, the particle size distribution of raindrops used in this study is that presented by Marshall and Palmer (1948), hereafter referred to as the M-P distribution. The M-P distribution is a very good approximation to the raindrop size distribution referred to natural rainfall and widely used for describing the midlatitude particle size distribution that are characterized by low to moderate intensity (e.g., Serio et al., 2019). However, particle size distribution is known to vary between rainfall types and climates. Therefore, it should be noted that when using the particle size distribution different from the M-P distribution, the change of the value of $N_0$ and $\lambda$ may lead to the change in the value of extinction coefficient $\beta$, resulting in an overestimation or underestimation of rainfall intensity estimate.

### 2.3 Relationship between image information, rainfall intensity, and scene depth

The extinction coefficient of the Image Degradation Model shown in Eq. (2) is the extinction coefficient obtained from the image information as shown in Eqs. (3) and (4). If the images were taken under rainfall conditions, the extinction coefficient in Eq. (2) will reflect rainfall intensity. On the other hand, the extinction coefficient using the rainfall intensity shown in Eq. (6) is a theoretically derived value, although it is approximate, based on the atmospheric radiation theory. Therefore, by substituting Eq. (6) into Eq. (2), the relationship between image information, rainfall intensity, and scene depth can be obtained as follows:

$$t(x) = \exp\left(-5.80 \times 10^{-5}\pi Q R^{0.63} d(x)\right) \tag{7}$$

$$t(x) = \frac{A - I(x)}{A - J(x)} \tag{8}$$

*Where $A - J(x) \neq 0$, and $0 \leq t(x) \leq 1$*

Equation (7) shows a relationship where transmission $t$ decreases exponentially as rainfall intensity $R$ increases and as scene depth $d$ increases.

Equations (7) and (8) can be transformed as follows:

$$R = \left[-\frac{1}{5.80 \times 10^{-5}\pi Q d(x)}\log_e\left(\frac{A - I(x)}{A - J(x)}\right)\right]^{\frac{1}{0.63}} \tag{9}$$

*Where $A - J(x) \neq 0$, and $0 \leq t(x) \leq 1$*

Equation (9) is a formula for estimating rainfall intensity from image information. The applicability of these relational equations will be examined in subsequent sections.

## 3 Materials and Methods

### 3.1 Rainfall photography and observation

We captured outdoor conditions including rain events and observed rainfall intensity by installing three cameras at observation sites (35° 45' 53" N, 138° 18' 42" E, 758 m a.s.l.) along the banks of the Omu River, which flows through Yamanashi Prefecture in central Japan. A plan view of the observation site is shown in Figure 1. The meteorological observations in 2021 around the observation site are shown in Figure 2. The figure shows data from a weather station about 24 km southeast of those cameras. The Köppen climate classification of the area around the observation site is humid subtropical climate, with hot, humid and heavy precipitation in summer and cool to mild in winter. Photography was taken using three commercially available interval cameras (TLC200Pro Brinno inc., Taiwan). The camera has a 1/3-inch HDR sensor with a resolution of 1.3 megapixels and a pixel size of 4.2 μm. The F-number, field of view, and focal length of the lens are F2.0, 112 degrees, and 19 mm in 35 mm format, respectively. The focus distance is from 40 cm to infinity. The resolution of the image is 1280 pixels wide by 720 pixels high. Images of the upstream, opposite bank, and downstream of the river were taken at one-minute intervals from the same point. Camera 1 took the upstream direction of the river, Camera 2 took the opposite bank direction, and Camera 3 took the downstream direction. The photography period was 235 days from April 19, 2021, to December 9, 2021. Images taken at night were excluded from the analysis because it was difficult to distinguish rainfall.

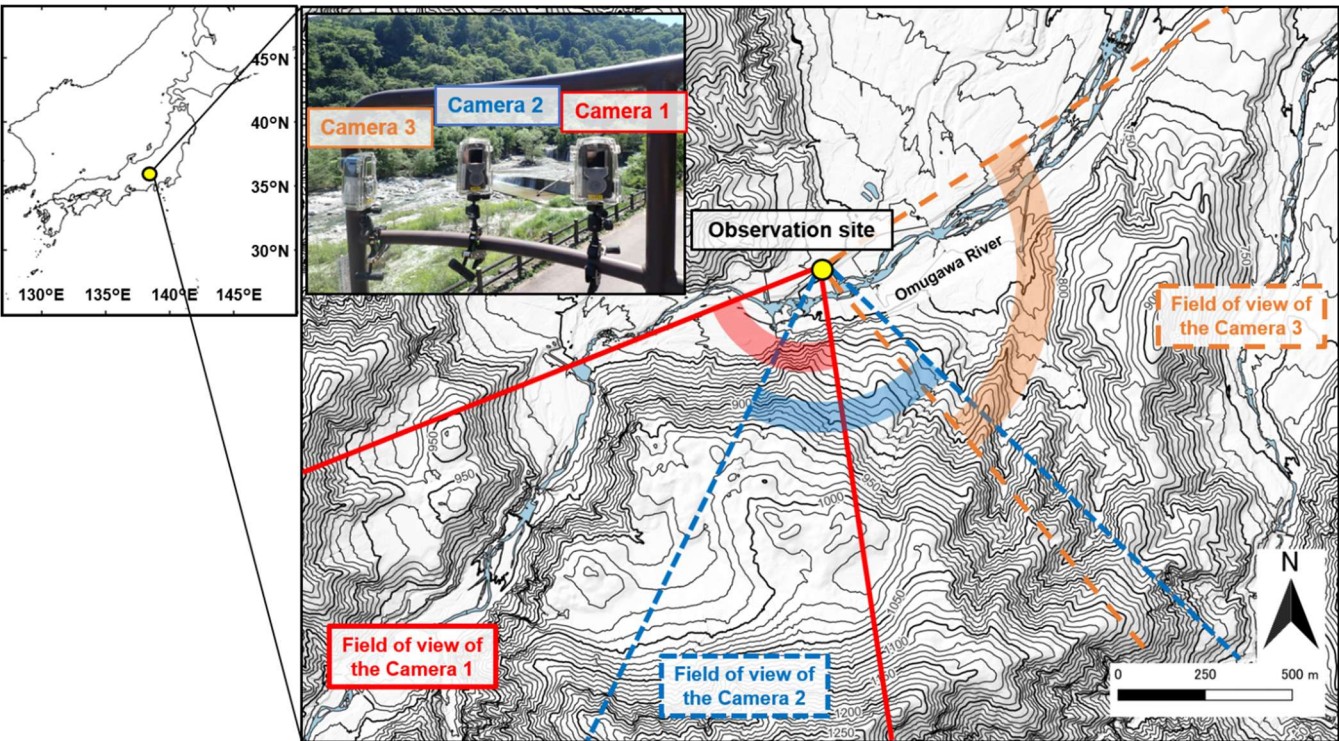

**Figure 1. Observation site plan. Coastline map made with Natural Earth (2018).**

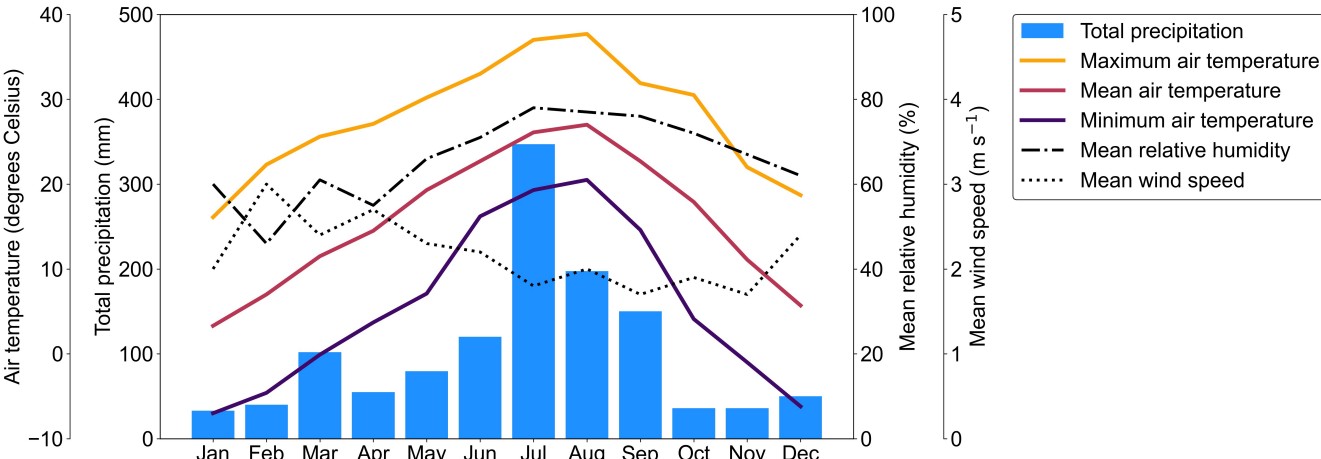

**Figure 2. The meteorological observations in 2021 around the observation site.**

One-minute rainfall intensity was also observed using a tipping bucket rain gauge (RG3-M Onset Computer Corporation, USA) at almost the same locations where the cameras were installed. In estimating rainfall intensity based on camera images, it is essential to consider the instantaneous intensity at the time of shooting. In contrast, when observing rainfall using a traditional tipping bucket, it is not possible to measure rainfall until it reaches the capacity of one tipping bucket. In other words, it is difficult to measure instantaneous values with a tipping bucket rain gauge with sufficient precision. However, in this study, to validate the accuracy of rainfall intensity estimated based on camera images, we decided to obtain data from a tipping bucket rain gauge with as fine a resolution as possible (one minute). The resolution and calibration accuracy of the tipping bucket rain gauge used was 0.2 mm and $\pm 1.0\%$, respectively. In the tipping bucket rain gauge, the number of tips in a unit of time is affected by the amount of water stored in the bucket in the previous unit of time due to the characteristics of the mechanism. Therefore, even if one tip occurs in a unit of time, the actual rainfall in a unit of time is considered to have a range from a value slightly larger than 0 to a value less than 0.4 mm. However, since the range is constant, we consider that a broad trend can be discussed. The total rainfall during the observation period was 1,257 mm, and the total daytime rainfall for the analysis was 685 mm. The maximum one-minute daytime rainfall intensity during the observation period was 0.8 mm min$^{-1}$. The number of images used for the analysis by rainfall intensity is shown in Table 1. Although the number of images at 0.8 mm min$^{-1}$ is small, there are more than 100 images at 0.4 mm min$^{-1}$ and above, so a broad trend can be discussed.

**Table 1. The number of images**

| Rainfall intensity (mm min$^{-1}$) | Camera 1 | Camera 2 | Camera 3 |
|---|---|---|---|
| 0.0 | 151,823 | 133,970 | 151,771 |
| 0.2 | 3,141 | 2,908 | 3,141 |
| 0.4 | 87 | 75 | 87 |
| 0.6 | 21 | 20 | 21 |
| 0.8 | 12 | 12 | 12 |

### 3.2 Image data preprocessing and processing

For the images of landscapes taken, background objects, such as sky, vegetation, and riverbeds, and their respective scene depths are different according to the angle of view of the camera and the area of the image. Then, to analyze the influence of background objects and scene depth, patches to be analyzed were set on the image. The analysis patch was defined as the center area of 30 × 30 pixels in each area of the image divided into 64 areas of 8 × 8. Serial numbers were assigned to 64 patches as shown in Figure 3. The magnitudes of image degradation with increasing rainfall should be related to the scene depth. If a relatively wide area is analyzed, the scene depth should vary considerably. Therefore, the limited number of pixels are set as analysis patches for each area. The representative value of each analysis patch was the mean value of the analysis patch.

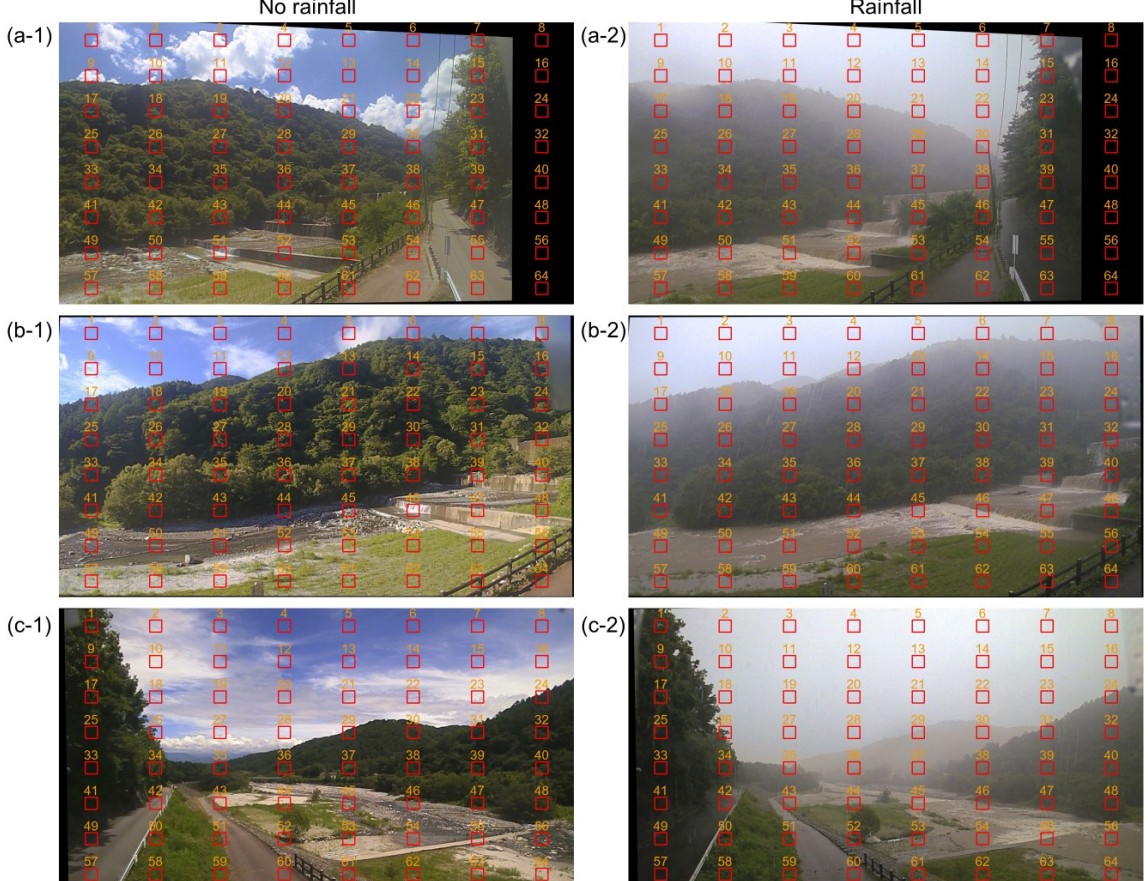

**Figure 3.** Analysis patches of the three cameras: (a-1), (b-1), and (c-1), respectively, show the images taken by Camera 1, Camera 2, and Camera 3 during no rainfall. Likewise, (a-2), (b-2), and (c-2) show the images taken by Camera 1, Camera 2, and Camera 3 during rainfall, respectively.

Concerning the parameters obtained from the images to be used in Eq. (8), observed intensity $I$ was the luminance value of the image taken. Global atmospheric light $A$ and scene radiance $J$ were calculated from observed intensity $I$ using the Dark Channel Prior method proposed by He et al. (2011), hereinafter referred to as DCP. DCP is a method of recovering an image with the effects of static weather removed, scene radiance $J$, using a single hazy image, observed intensity $I$. The procedure for recovering scene radiance $J$ from observed intensity $I$ by DCP is described in Appendix B.

DCP is not a machine learning-like method that requires a large amount of prior learning but is a method that can simply estimate global atmospheric light $A$ and scene radiance $J$ from a single image with relatively little calculation amount. Therefore, this study has adopted a method using DCP. In addition, since the angle of view may change even with the same camera in long-term photography, image registration was performed so that the angle of view was the same throughout the

entire term. Image registration was performed by combining feature detection using the Accelerated-KAZE (Alcantarilla et al., 2013) algorithm and image warping by homography.

Scene depth $d$ was calculated as the oblique distance from the camera to the intersection of (i) the light path in the camera's
line-of-sight direction obtained from the camera's latitude, longitude, height above sea level, azimuth angle, and elevation angle information and (ii) the background 5-m digital elevation models created from the aerial laser survey data (Geospatial Information Authority of Japan, 2018). The scene depth of each analysis patch was defined as the scene depth at the center position of each patch.

The values of parameters global atmospheric light $A$, scene radiance $J$, observed intensity $I$, and scene depth $d$ calculated for
each image were applied to the proposed relational equations (Eqs. (7), (8), and (9)) to analyze the relationship between transmission $t$, rainfall intensity $R$, and scene depth $d$ in each analysis patch. The flowchart of estimating rainfall intensity from image information by Eq. (9) is shown in Figure. 4. The image processing was performed using OpenCV4.0.1, an open-source library in the Python 3.8.12 programming language. For DCP calculation, we referred to the source code in Zhang (2021).

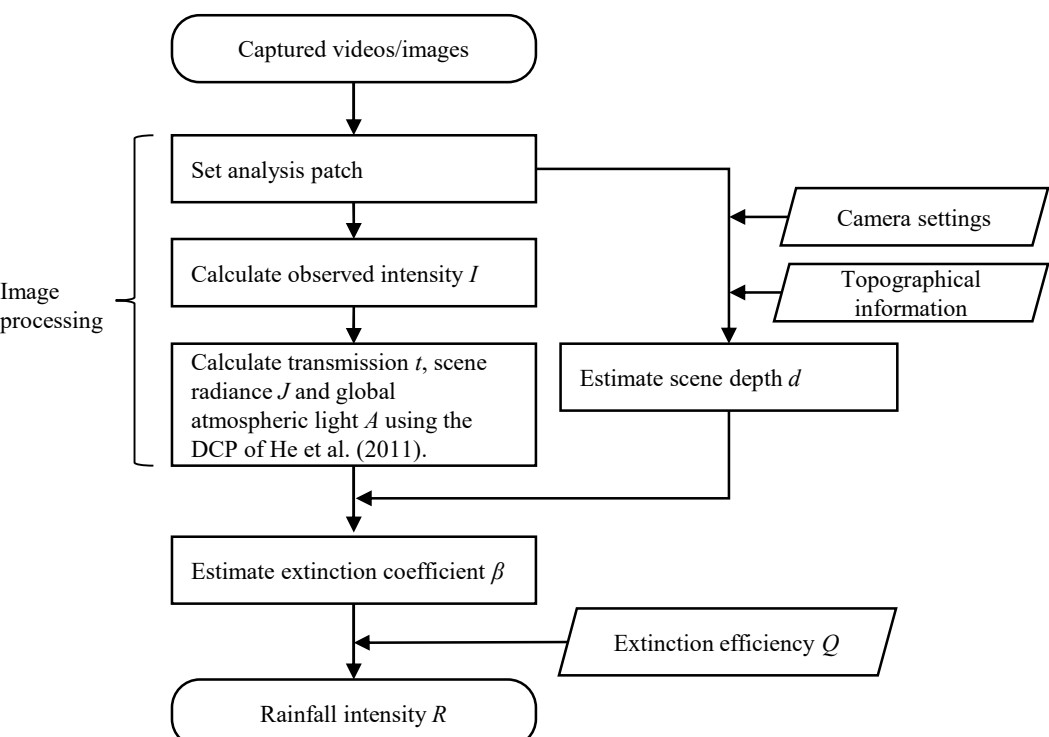

**Figure 4. The flowchart of estimating rainfall intensity from image information.**

## 4 Results

### 4.1 Distribution of observed intensity *I*, scene radiance *J*, global atmospheric light *A*, and transmission *t*

Figures 5, 6, and 8 show the distribution of observed intensity $I$, scene radiance $J$, and transmission $t$ for each rainfall intensity, respectively. These figures show the top three patches of scene depth for each camera. This is because the greater the scene depth, the more likely static weather effects are to appear, making it easier to understand the characteristics of static weather effects. Figures which include all patches are shown in Appendix C-1, C-2, and C-3. As shown in Appendix C-1, C-2, and C-3, Patches where the appropriate scene depth could not be calculated due to the presence of sky background and the application of geometric corrections in the image registration process, such as the upper and rightmost patch of Camera 1, were excluded from the analysis. Figures 7 show the distribution of global atmospheric light $A$ for each rainfall intensity. Global atmospheric light $A$ is set to one value per image, regardless of the patch. Furthermore, the slope of the regression line by single regression analysis in the relationship between rainfall intensity and the mean values of observed intensity $I$, scene radiance $J$, global atmospheric light $A$, and transmission $t$ are shown in Figures 5, 6, 7, and 8. Although an exponential relationship between rainfall intensity and observed intensity $I$, scene radiance $J$, global atmospheric light $A$, and transmission $t$ is expected as shown in Eqs. (7) and (8), a simple regression analysis was conducted here to analyze a simple trend.

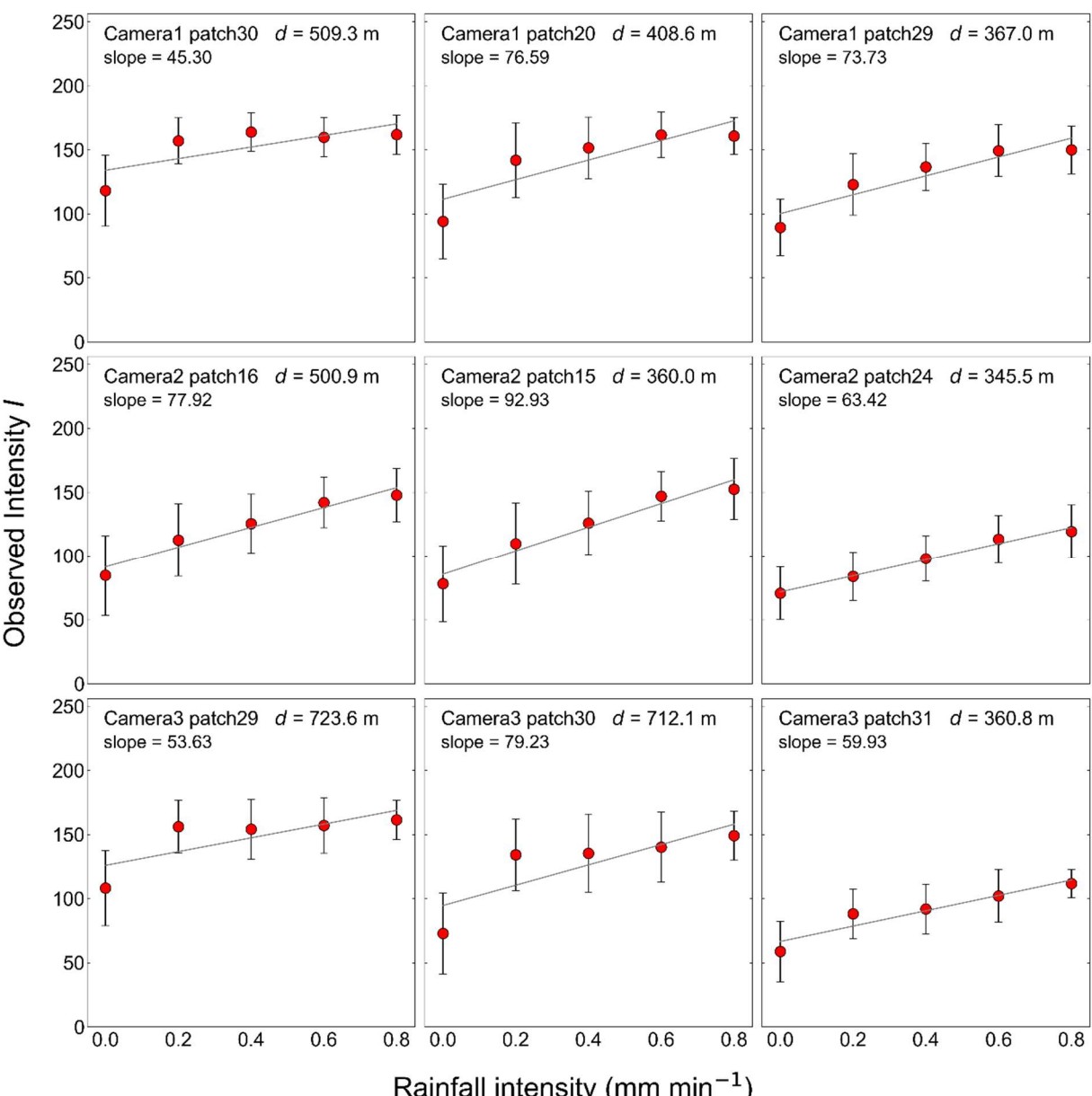

Figure 5. Distribution of observed intensity *I* by rainfall intensity. Figures show the top three patches of scene depth for each camera. Figures which include all patches are shown in Appendix C-1. Each figure is marked with a camera name, patch number, scene depth and slope of the linear regression line for the relationship between rainfall intensity and observed intensity *I*. The upper three figures are Camera 1, the middle three figures are Camera 2, and the lower three figures are Camera 3. The plots and error bars show the mean value and standard deviation of all data during the observation period. The straight lines show linear regression line for the relationship between rainfall intensity and observed intensity *I*. Rainfall intensity is observed by the rain gauge.

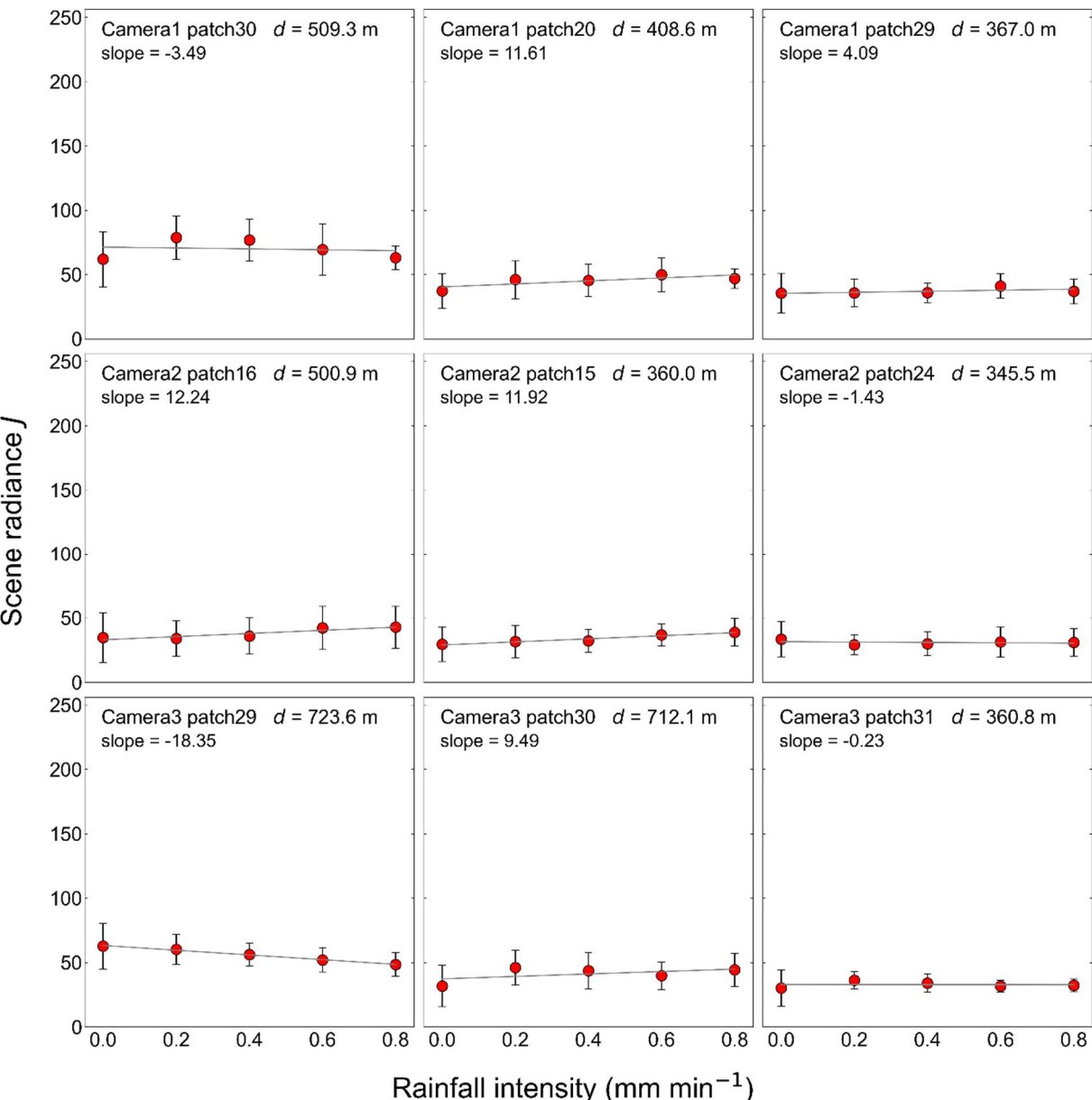

**Figure 6. Distribution of scene radiance *J* by rainfall intensity. Figures show the top three patches of scene depth for each camera. Figures which include all patches are shown in Appendix C-2. Each figure is marked with a camera name, patch number, scene depth and slope of the linear regression line for the relationship between rainfall intensity and scene radiance *J*. The upper three figures are Camera 1, the middle three figures are Camera 2, and the lower three figures are Camera 3. The plots and error bars show the mean value and standard deviation of all data during the observation period. The straight lines show linear regression line for the relationship between rainfall intensity and scene radiance *J*. Rainfall intensity is observed by the rain gauge.**

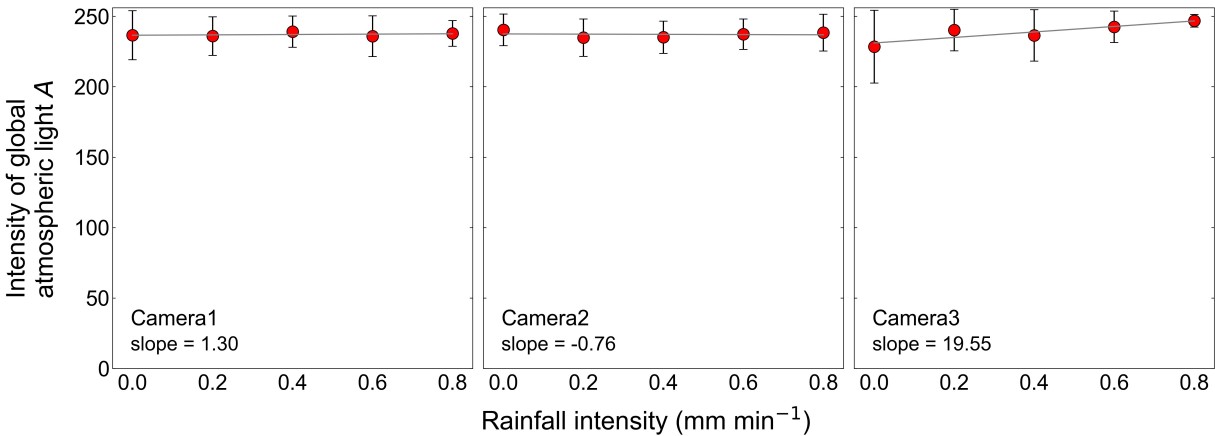

**Figure 7. Distribution of global atmospheric light $A$ by rainfall intensity. Each figure is marked with a camera name and slope of the linear regression line for the relationship between rainfall intensity and global atmospheric light $A$. The left figure is Camera 1, the center figure is Camera 2, and the right figure is Camera 3. The plots and error bars show the mean value and standard deviation of all data during the observation period. The straight lines show linear**

**regression line for the relationship between rainfall intensity and global atmospheric light $A$. Rainfall intensity is observed by the rain gauge.**

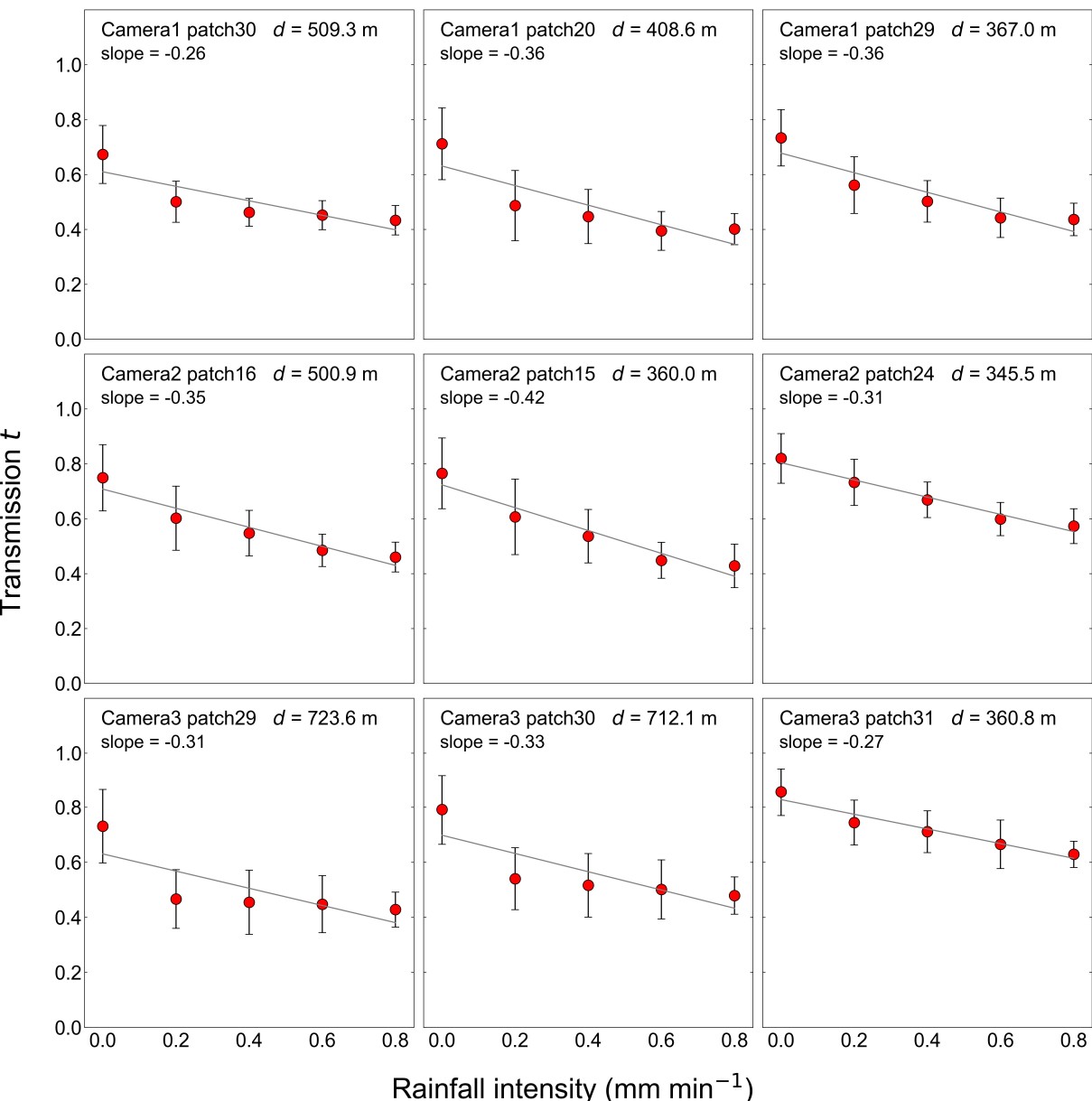

**Figure 8. Distribution of transmission *t* by rainfall intensity. Figures show the top three patches of scene depth for each camera. Figures which include all patches are shown in Appendix C-3. Each figure is marked with a camera name, patch number, scene depth and slope of the linear regression line for the relationship between rainfall intensity and transmission *t*. The upper three figures are Camera 1, the middle three figures are Camera 2, and the lower three figures are Camera 3. The plots and error bars show the mean value and standard deviation of all data during the observation period. The straight lines show linear regression line for the relationship between rainfall intensity and transmission *t*. Rainfall intensity is observed by the rain gauge.**

As shown in Figure 5, the mean values of observed intensity $I$ range from approximately 50 to 170, and the value and distribution range of observed intensity $I$ vary for each patch. For example, comparing the figures for patches 30 and 29 of camera 1 in the upper panel of Figure 5, when the rainfall intensity is 0.0 mm min$^{-1}$, the mean value of observed intensity $I$ in patch 30 is approximately 120, while in patch 29, it is approximately 90. Similarly for the other cameras, each patch has a different observed intensity when the rainfall intensity is 0.0 mm min$^{-1}$. This is because the background color and luminance are inherently different due to the different background objects in each patch. On the other hand, the tendency of increasing observed intensity $I$ as the rainfall intensity increases is consistent across all patches although the slope of the regression line is different in each patch somewhat. This means that, as an overall trend, the whiteness of the image increases as rainfall intensity increases. Furthermore, in patches with particularly large scene depths, such as patch 30 of Camera 1, patch 29 of Camera 3, and patch 30 of Camera 3, observed intensity $I$ tend to increase significantly when the rainfall intensity is 0.2 mm min$^{-1}$, and then remains almost constant even if the rainfall intensity increases further. This suggests that there is an upper limit to the observed intensity $I$, and that scene depth and rainfall intensity may interrelate to determine the extent of increase in observed intensity $I$.

Next, as shown in Figure 6, the mean values of scene radiance $J$ range from approximately 20 to 80. The tendency of scene radiance $J$ is different from the tendency of observed intensity $I$, and the effect of rainfall intensity is limited and varies little in any of the cameras. Moreover, as shown in Figure 7, the mean values of global atmospheric light $A$ range from approximately 220 to 240, and the effect of rainfall intensity on global atmospheric light $A$ is also limited and varies little in any of the cameras.

Finally, as shown in Figure 8, the mean values of transmission $t$ range from approximately 0.4 to 0.9, and the value and distribution range of transmission $t$ vary for each patch. On the other hand, the tendency of decreasing transmission $t$ as the rainfall intensity increases is consistent across all patches and is opposite to the tendency of the relationship between observed intensity $I$ and rainfall intensity. This tendency between transmission $t$ and rainfall intensity quantitatively indicates that the background becomes gradually hazy and less visible as the rainfall intensity increases. Furthermore, in patches with particularly large scene depths, such as patch 30 of Camera 1, patch 29 of Camera 3, and patch 30 of Camera 3, transmission $t$ tends to decrease significantly when the rainfall intensity is 0.2 mm min$^{-1}$, and then remains almost constant even if the rainfall intensity increases further. As with the case of observed intensity $I$, this suggests that there is a lower limit to the transmission $t$, and that scene depth and rainfall intensity may interrelate to determine the extent of decrease in transmission $t$.

## 4.2 Relationship between transmission $t$, rainfall intensity $R$, and scene depth $d$

Figure 9 shows the relationship between transmission $t$ calculated by Eq. (8), observed rainfall intensity $R$, and scene depth $d$ for each patch. In all cameras, if observed rainfall intensity is constant, transmission $t$ gradually decreases as scene depth increases. Similarly, if scene depth is constant, transmission $t$ will gradually decrease as rainfall intensity increases. These

data clearly show that transmission $t$ decreases exponentially according to the increase in rainfall intensity $R$ and scene depth

$d$, as shown in Eq. (7). As described in 4.1, the fact that scene depth and rainfall intensity may interrelate to determine the extent of decrease in transmission $t$ is also in the same sense. Therefore, the proposed relationship, Eqs. (7) and (8), are considered applicable to images taken outdoors in practice. Furthermore, in the Figures at the time of rainfall in each camera such as rainfall intensity $R$ from 0.2 to 0.8 mm min$^{-1}$, the plots generally ranged between the theoretical lines of $Q = 0.5$ to 2.0. However, in patches where scene depth $d$ was less than approx. 100 m, the plots often ranged below the line of $Q = 2.0$.

In the patches ranging below the $Q = 2.0$ line, the ratio of scene radiance $J$ to global atmospheric light $A$ tends to be higher. In addition, theoretically, if there is no rainfall, i.e., $R = 0.0$ mm min$^{-1}$, transmission $t$ should always be 1.0 without decreasing. However, even in the case of no rainfall, transmission $t$ tends to decrease according to scene depth.

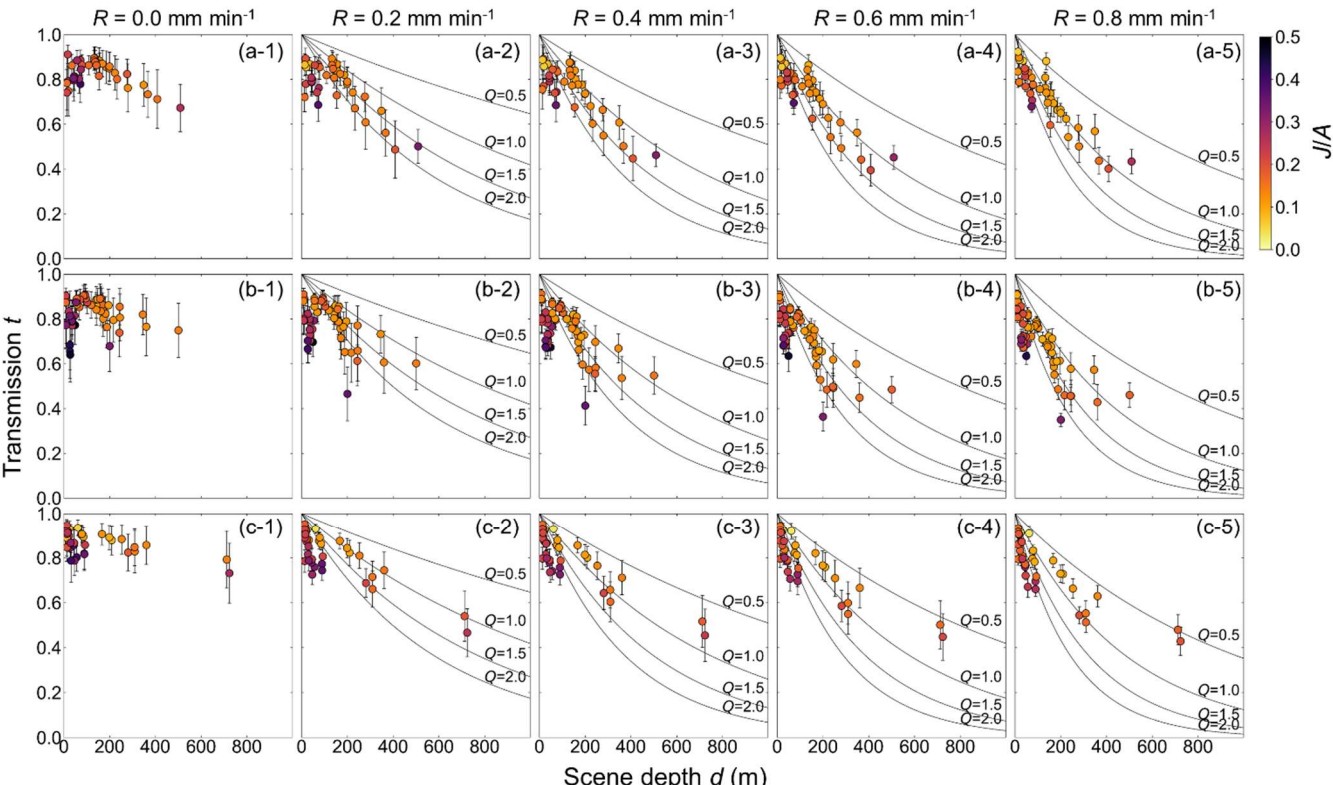

**Figure 9. Relationship between transmission $t$ and scene depth $d$: (a-1)–(a-5), respectively, show the results of Camera 1 by rainfall intensity ((a-1) $R$=0.0 mm min$^{-1}$, (a-2) $R$=0.2 mm min$^{-1}$, (a-3) $R$=0.4 mm min$^{-1}$, (a-4) $R$=0.6 mm min$^{-1}$, and (a-5) $R$=0.8 mm min$^{-1}$). Likewise, (b-1)–(b-5) show the results of Camera 2 by rainfall intensity, and (c-1)–(c-5) show the results of Camera 3 by rainfall intensity, respectively. The plots show the mean value of all image data in each**

**patch, and the error bars show the standard deviation. The theoretical relationship between transmission $t$ and scene**

depth *d* is shown as a curve when extinction efficiency *Q* is given in Eq. (7) for four patterns: 0.5, 1.0, 1.5, and 2.0 for each rainfall intensity. The theoretical transmission *t* is not shown because the transmission *t* is always 1 when *R*=0.0 mm min$^{-1}$. Each plot is shown in a different color depending on the ratio of scene radiance *J* to global atmospheric light *A*.

## 5 Discussion

### 5.1 Factors of the value and the variation of transmission *t* according to rainfall intensity

As shown in Eq. (4), transmission *t* is determined by the relationship between observed intensity *I*, scene radiance *J*, and global atmospheric light *A*. However, as shown in Figures 5, 6, 7, and 8, the values and trend of variation for observed intensity *I*, scene radiance *J*, global atmospheric light *A*, and transmission *t* vary according to rainfall intensity. Therefore, it was verified which of the following factors, observed intensity *I*, scene radiance *J*, or global atmospheric light *A*, strongly affected the value and the variation of transmission *t* according to rainfall intensity.

Figure 10 shows the relationship between (i) the mean value of observed intensity *I*, scene radiance *J*, and global atmospheric light *A* according to rainfall intensity in each patch for the three cameras shown in Figures 5, 6, 7, Appendix C-1, and  Appendix C-2, and (ii) the mean value of transmission *t* shown in Figure 8, and Appendix C-3. Table 2 shows the slope of the regression line and the value of the coefficient of determination R$^2$ obtained by simple regression analysis. Figure 10 and Table 2 clearly show a negative correlation between observed intensity *I* and transmission *t*, where transmission *t* decreases as observed intensity *I* increases in all three cameras. In the results of the single regression analysis, the coefficient of determination was 0.47 to 0.69 in the case of no rainfall and 0.74 to 0.90 in the case of rainfall, which indicates a strong negative correlation. That is, the value of transmission *t* has a strong relationship with the value of observed intensity *I*. In addition, the absolute value of the slope of the regression line gradually increases as rainfall intensity increases. This indicates that as rainfall intensity becomes greater, the value of transmission *t* tends to respond to the value of observed intensity *I* more sensitively and vary more. Furthermore, in each patch, especially patches where the range of variation of transmission *t* is large, observed intensity *I* increases and transmission *t* decreases as rainfall intensity increases. From this, it can be said that in patches where the range of variation of transmission *t* is large, as rainfall intensity increases, the apparent whiteness of the image tends to increase.

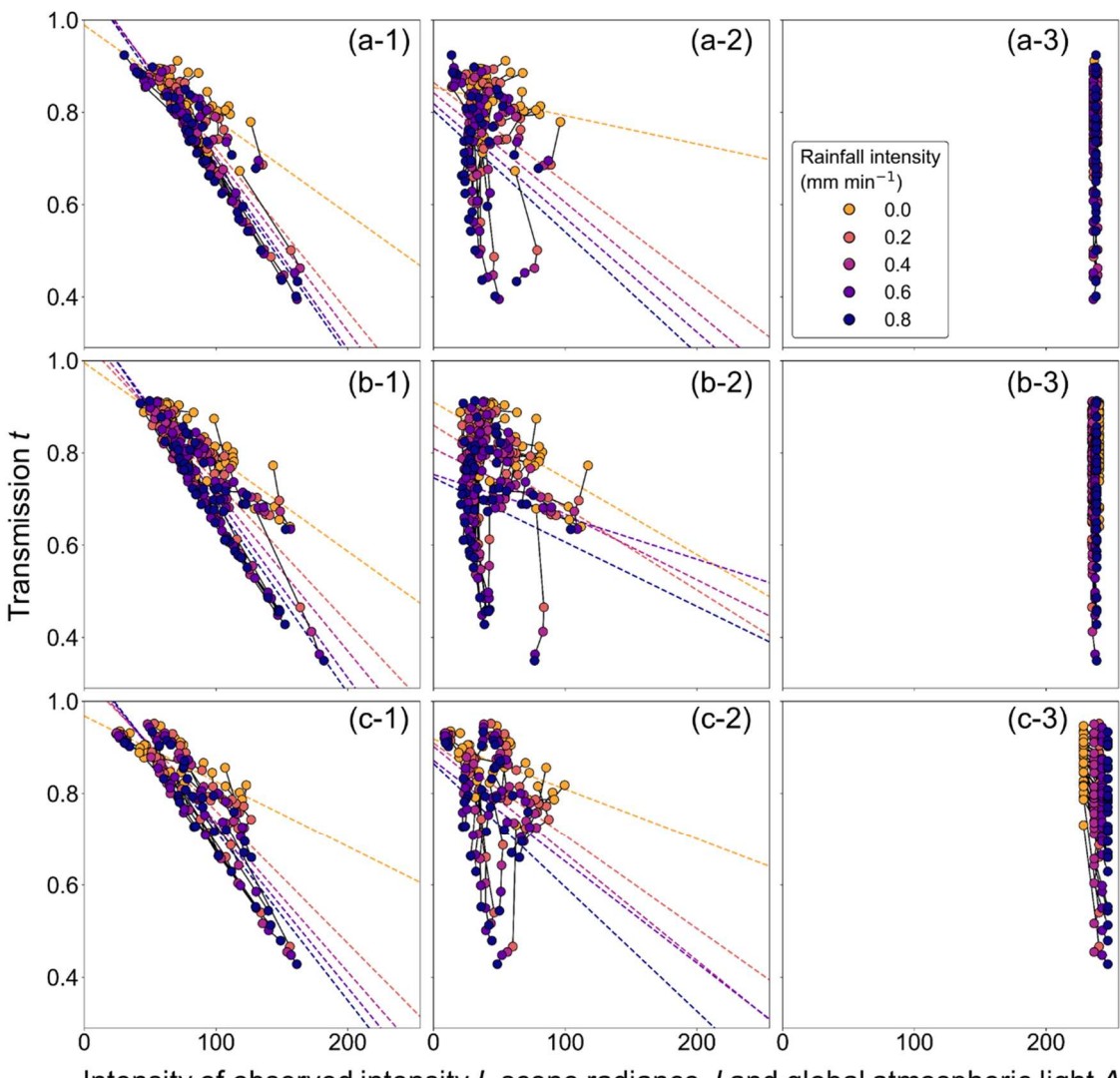

**Figure 10. Relationship between observed intensity *I*, scene radiance *J*, global atmospheric light *A* and transmission *t***
**by analysis patch and rainfall intensity:**

**(a-1)–(a-3), respectively, show the relationship between observed intensity *I*, scene radiance *J*, global atmospheric**
**light *A* and transmission *t* in Camera 1. Likewise, (b-1)–(b-3) show the relationship in Camera 2, and (c-1)–(c-3) show**
**the relationship in Camera 3, respectively. The plots by rainfall intensity for each patch were connected by straight**
**lines to show the transition associated with changes in rainfall intensity in one patch. Global atmospheric light *A* is set**
**to one value per image, so the values are all the same in each patch. In the Figures of observed intensity *I* and scene**
**radiance *J*, the regression lines from the single regression analysis by rainfall intensity are shown as dotted lines that**
**match the colors of the scatter diagram.**

**Table 2. Slope and coefficient of determination $R^2$ of the linear regression line for the relationship between observed intensity $I$, scene radiance $J$ and transmission $t$ by rainfall intensity**

| | | Slope (×10⁻³) | | | | | Coefficient of determination $R^2$ | | | | |
|---|---|---|---|---|---|---|---|---|---|---|---|
| | Rainfall intensity (mm min⁻¹) | 0.0 | 0.2 | 0.4 | 0.6 | 0.8 | 0.0 | 0.2 | 0.4 | 0.6 | 0.8 |
| $I$ vs $t$ | Camera 1 | -2.04 | -3.53 | -3.79 | -4.03 | -4.06 | 0.47 | 0.81 | 0.86 | 0.88 | 0.90 |
| | Camera 2 | -2.04 | -3.05 | -3.47 | -3.92 | -4.09 | 0.69 | 0.74 | 0.77 | 0.81 | 0.86 |
| | Camera 3 | -1.42 | -2.88 | -3.25 | -3.48 | -3.66 | 0.56 | 0.74 | 0.79 | 0.82 | 0.87 |
| $J$ vs $t$ | Camera 1 | -0.61 | -2.16 | -2.38 | -2.48 | -2.63 | 0.04 | 0.12 | 0.10 | 0.08 | 0.08 |
| | Camera 2 | -1.65 | -1.78 | -1.42 | -0.92 | -1.39 | 0.36 | 0.14 | 0.07 | 0.02 | 0.03 |
| | Camera 3 | -1.09 | -2.02 | -2.33 | -2.20 | -2.69 | 0.27 | 0.16 | 0.14 | 0.09 | 0.11 |

Next, in the relationship between scene radiance $J$ and transmission $t$, the slope of the regression line was negative in all three cameras. However, the coefficient of determination was 0.04 to 0.36 in the case of no rainfall and 0.02 to 0.16 in the case of rainfall, which indicates a generally weak negative correlation or almost no correlation. In each patch, changes in scene radiance $J$ and transmission $t$ according to changes in rainfall intensity were also not clear. In the patch where scene radiance $J$ is relatively high when rainfall intensity is 0.0 mm min⁻¹, scene radiance $J$ tends to decrease as rainfall intensity increases. However, since it is not clearly linked to changes in transmission $t$, it can be said that the effect of changes in scene radiance $J$ associated with changes in rainfall intensity on transmission $t$ is limited. Then, in the relationship between global atmospheric light $A$ and transmission $t$, the relationship between global atmospheric light $A$ and transition of transmission $t$ according to changes in rainfall intensity was not clearly found because global atmospheric light $A$ was almost constant at 200 or more in all three cameras. These results suggest that the value and the variation of transmission $t$ according to the increase in rainfall intensity are strongly influenced mainly by the value of observed intensity $I$.

## 5.2 Validity of the extinction coefficient $\beta$ determined from images

### 5.2.1 Rationale for rainfall causing static weather effects

As indicated in Section 1, it has been suggested that rain causes static weather effects because individual raindrops cannot be identified by the camera's sensor when they are more than a certain distance away from the camera. Therefore, this section briefly examines the validity of treating rain as static weather in this study.

The actual height and width of the background in the image varies with the distance from the camera. The height and width are smaller for scenes closer to the camera and larger for scenes farther away from the camera. Therefore, if the image resolution is constant, the actual height and width of the scene occupied by a single pixel also vary with the distance from the camera. In this section, we examine the actual width of the scene occupied by a single pixel in images taken with our camera. It should be noted that the results are approximations since lens distortion is not considered here.

The angle of view of the camera used in this study is 112°. Therefore, at a distance of $d$ (m) from the camera, a width of $2 \times d \times \tan(112/2)$ (m) appears in the image. At a distance of 1 m from the camera, the width is approximately 3 m. The resolution of images captured by this camera is 1280 pixels wide by 720 pixels high. Thus, at a distance of $d$ (m) from the camera, a single pixel occupies a width of $2 \times d \times (\tan(112/2))/1280$ (m). The radius of raindrops is 0.1-10 mm (Narasimhan and Nayar, 2002). If the radius of a raindrop is 1 mm, the distance where the width of a single pixel and the

diameter of a single raindrop are the same is about 0.86 m. Therefore, raindrops further than about 0.86 m from the camera are smaller than a single pixel and cannot be identified by the camera's sensor. In other words, raindrops further than about 0.86 m from the camera are considered to cause static weather effects. The fact that the cameras used in the field in this study captured scenes from several 10 to several 100 meters away suggests that it is reasonable to treat rainfall as static weather.

**5.2.2 Values and trends of the extinction coefficient $\beta$ determined from images**

In this study, as shown in section 2, we linked the extinction coefficient obtained from image information with the rain extinction coefficient approximately obtained from the atmospheric radiation theory. Since there are few examples of rain extinction coefficient values obtained from images in the past, the validity of the values is verified below.

Figure 11 shows the relationship between the value of extinction coefficient $\beta$ calculated from the image and scene depth $d$

for each rainfall intensity. The extinction coefficient obtained from the image was calculated by Eq. (3) after determining transmission $t$ from observed intensity $I$, global atmospheric light $A$, and scene radiance $J$ of the image, as shown in Eq. (4). The Figure at the time of rainfall in each camera such as rainfall intensity $R$ from 0.2 to 0.8 mm min$^{-1}$ shows the values of extinction coefficient for the extinction efficiency $Q$ of 0.5, 1.0, 1.5, and 2.0 and the values of extinction coefficient given in the previous study to be discussed in section 5.2.3. In all three cameras, the value of extinction coefficient $\beta$ in the case of no

rainfall, i.e., rainfall intensity $R = 0.0$ mm min$^{-1}$, is the order of $10^{-4}$ to $10^{-2}$, while the value of extinction coefficient $\beta$ in the case of rainfall is the order of $10^{-3}$ to $10^{-2}$. In addition, in all rainfall intensities, a trend is seen that extinction coefficient $\beta$ decreases as scene depth increases in patches where scene depth $d$ is less than approx. 100 m, while it remains nearly constant when scene depth $d$ is more than approx. 100 m. These values and trends of extinction coefficient $\beta$ will be discussed in the following sections.

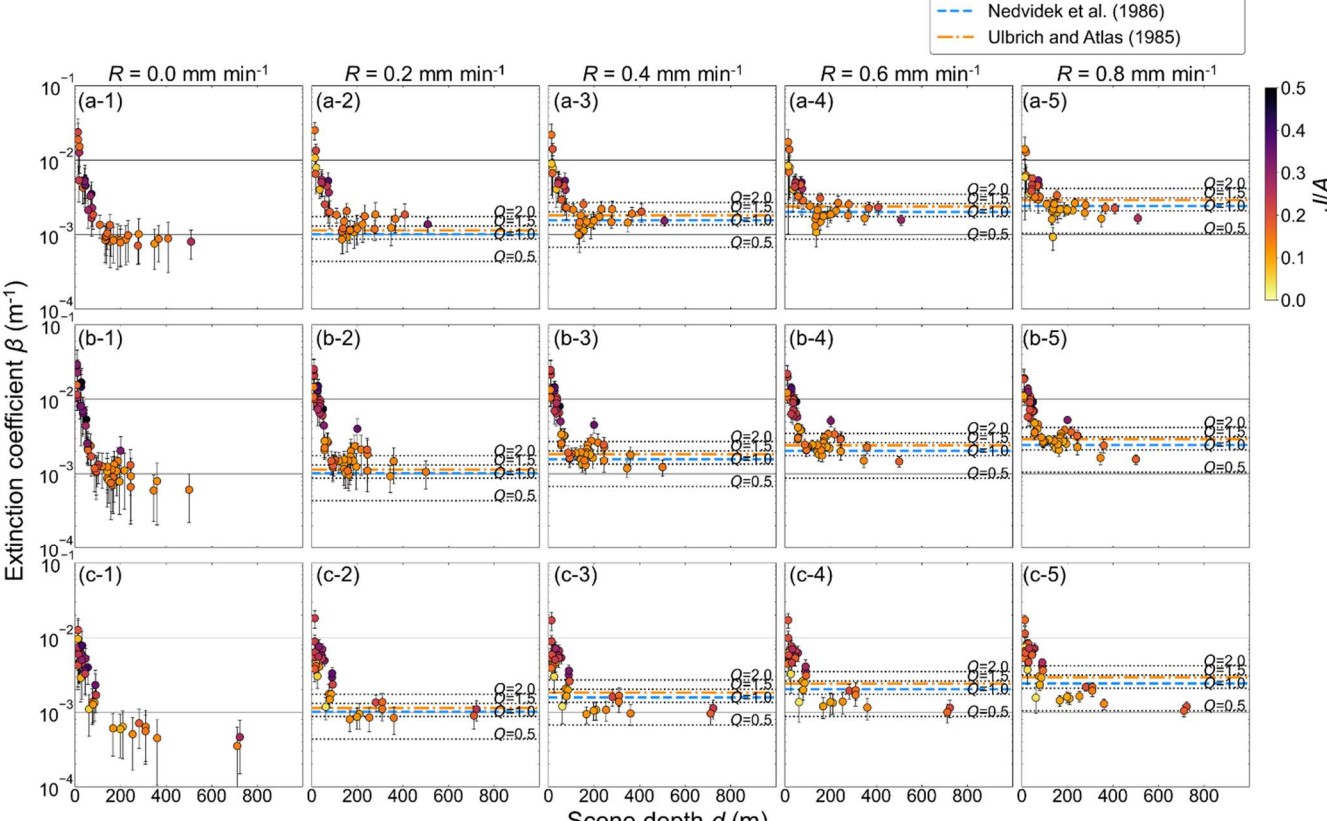

**Figure 11. Relationship between extinction coefficient $\beta$ and scene depth $d$: (a-1)–(a-5), respectively, show the results of Camera 1 by rainfall intensity ((a-1) $R$=0.0 mm min$^{-1}$, (a-2) $R$=0.2 mm min$^{-1}$, (a-3) $R$=0.4 mm min$^{-1}$, (a-4) $R$=0.6 mm min$^{-1}$, and (a-5) $R$=0.8 mm min$^{-1}$). Likewise, (b-1)–(b-5) show the results of Camera 2 by rainfall intensity, and (c-1)–(c-5) show the results of Camera 3 by rainfall intensity, respectively. The plots show the mean value of all image data in each patch, and the error bars show the standard deviation. The values of extinction coefficient $\beta$ is shown as dotted lines when extinction efficiency $Q$ is given in Eq. (6) for four patterns: 0.5, 1.0, 1.5, and 2.0 for each rainfall intensity. The values of extinction coefficient $\beta$ shown in previous studies is shown as blue line (Nedvidek et al., 1986) and orange line (Ulbrich and Atlas, 1985). Each plot is shown in a different color depending on the ratio of scene radiance $J$ to global atmospheric light $A$.**

### 5.2.3 Validity of extinction coefficient $\beta$ determined from images in the case of rainfall

Although no research has been conducted to determine the extinction coefficient of rain from images, there are many examples in the field of radar meteorological observation and telecommunications where the extinction coefficient is

determined from the attenuation of electromagnetic waves due to rain using electromagnetic waves with wavelengths in the visible light and near-infrared regions(Bradley et al., 2000; Nedvidek et al., 1986; Shipley et al., 1974; Suriza et al., 2013; Ulbrich and Atlas, 1985; Zaki et al., 2019). Visible light is an electromagnetic wave with a wavelength of approx. 360 nm to 830 nm and a camera can be regarded as a sensor that detects electromagnetic waves in that wavelength range. Uijlenhoet et al. (2011) indicated that both theoretically and experimentally the attenuation of visible and near-infrared signals over paths ranging from a few hundred meters to several kilometers can be used to estimate the average rainfall over a path. The concept of attenuation and extinction coefficients of electromagnetic waves due to rain in such previous studies can apply to this study. According to previous studies, the extinction coefficient of electromagnetic waves due to raindrops can be expressed by the following equation (e.g., Ulbrich and Atlas, 1985).

$$\beta = aR^b \tag{10}$$

The two parameters $a$ and $b$ in Eq. (10) represent the difference in the particle size distribution of raindrops. Comparing the extinction coefficient of Eq. (6) and Eq. (10), we obtain $a = 5.80 \times 10^{-5} \pi Q$, $b = 0.63$. In the previous studies, for example, Ulbrich and Atlas (1985) proposed the theoretical values $a = 2.12 \times 10^{-4}$ and $b = 0.68$ based on the results of previous experiments on rainfall intensity and optical attenuation, including the experiment of Shipley et al. (1974). On the other hand, Nedvidek et al. (1986) proposed the values $a = 2.12 \times 10^{-4}$ and $b = 0.63$ based on the results of experiments using near-infrared light sources and reflectors. All the values of extinction coefficients shown in the unit of dB km$^{-1}$ in the previous studies were converted to m$^{-1}$. Figure 11 shows the results of calculating the extinction coefficient $\beta$ using the values of $a$ and $b$ shown in these previous studies. The values of extinction coefficient $\beta$ shown in these previous studies are in the order of $10^{-3}$. The values of extinction coefficient $\beta$ obtained from the images in this study in the case of rainfall are almost constant with the order of $10^{-3}$ in patches where scene depth $d$ is more than approx. 100 m. Therefore, the results show that the extinction coefficient $\beta$ in patches where scene depth $d$ is more than approx. 100 m is almost consistent with the value shown in the previous study. However, the extinction coefficient $\beta$ in patches where scene depth $d$ is less than approx. 100 m is a significant overestimate compared to the previous studies. The reasons for this overestimate are discussed in 5.2.5. As indicated in section 2, extinction efficiency $Q$ is ideally 2 (Chylek, 1977; Uijlenhoet et al., 2011), but the values of extinction coefficient in the previous studies ranged between 1.0 and 1.5. It has been indicated that the reason for this difference in the value of $Q$ is that the ideal case of $Q = 2$ tends to overestimate the number of very small raindrops in the raindrop population (Bradley et al., 2000; Rogers et al., 1997).

### 5.2.4 Validity of extinction coefficient $\beta$ determined from images in the case of no rainfall

In the case of no rainfall, as seen from Eq. (6), the rain extinction coefficient approximately obtained from the atmospheric radiation theory is expected to be normally zero, and the extinction coefficient obtained from the image is also expected to be zero (synonymous with the transmission $t$ of 1). However, as shown in the no-rainfall Figure in Figure 11 in the case of

no rainfall, the extinction coefficient indicated almost the same trend in the three cameras, decreasing between the order of $10^{-2}$ and $10^{-3}$ in patches where scene depth was less than approx. 100 m, and remaining almost constant between $10^{-3}$ and $10^{-4}$ when scene depth was more than approx. 100 m. It is noted that since the extinction coefficient is expressed as an exponential function of transmission and scene depth as in Eq. (3), the facts that transmission $t$ exponentially decreases in the range where scene depth is more than approx. 100 m in the no-rainfall Figure in Figure 9 and that the extinction coefficient is constant in the range where scene depth is more than approx.100 m in Figure 11 have the same meaning.

The reason why the extinction coefficient is not zero when there is no rainfall may be due to the effect of aerosols in the atmosphere. In outdoor photography, not only hydrometeors, such as rain and fog, which are the subject of this study, but also lithometeors, such as smoke and dust, degrade visibility and change the appearance of the background. Therefore, images taken during no rainfall do not show the effects of rain but may show the effects of hydrometeors and lithometeors that are not observed as rainfall intensity. In this paper, hydrometeors and lithometeors that are not observed as rainfall intensity are collectively referred to as aerosols.

Because of the importance of atmospheric aerosols to air pollution and the human health impacts caused by it, traffic and airport safety, and climate change, many studies have been conducted to understand the characteristics of aerosols (Kim and Noh, 2021). Some of these studies have reported on the relationship between atmospheric aerosols and atmospheric extinction coefficients (Kim and Noh, 2021; Ozkaynak et al., 1985; Shin et al., 2022; Uchiyama et al., 2014; Uchiyama et al., 2018). Ozkaynak et al. (1985) calculated the values of the extinction coefficient from the results of visibility observation in 12 airports at large cities in the U.S. and reported that they were $4.0 \times 10^{-5} - 7.8 \times 10^{-4}$ m$^{-1}$. Uchiyama et al. (2014) reported that the mode of extinction coefficients observed at Tsukuba, Japan, using an integrating nephelometer and one- and three-wavelength absorption spectrometers were $2.5 \times 10^{-5}$ m$^{-1}$, and most values were not more than $2.0 \times 10^{-4}$ m$^{-1}$. Uchiyama et al. (2018) also observed extinction coefficients in two cities, Fukuoka, Japan, and Beijing, China, using an integrating nephelometer and an aethalometer, and found that the annual mean for Fukuoka was $7.46 \times 10^{-5}$ m$^{-1}$ and for Beijing, $4.12 \times 10^{-4}$ m$^{-1}$. Kim and Noh (2021) obtained the extinction coefficients of atmospheric aerosols from camera images and reported that the estimated range was $5.0 \times 10^{-5}$ to $1.0 \times 10^{-3}$ m$^{-1}$ and the optimal aerosol extinction coefficient was approx. $5.0 \times 10^{-4}$ m$^{-1}$. Furthermore, Shin et al. (2022) reported that the range obtained from the camera images and visibility data was $2.0 \times 10^{-6}$ to $1.1 \times 10^{-3}$ m$^{-1}$. In reference to these reports, although there are differences in the air pollution conditions at the observation sites and the observation methods used, the value of the atmospheric extinction coefficient is expected to be the order of $10^{-6}$ to $10^{-3}$ in m$^{-1}$ unit due to aerosol effects even if there is no rainfall. In the results of this study, the extinction coefficient is the order of $10^{-3}$ to $10^{-4}$ in patches where scene depth is more than approx. 100 m, as shown in the no-rainfall Figure in Figure 11. This result is a slight overestimation compared to the results observed in Japan in recent years, i.e., Uchiyama et al. (2014) and Uchiyama et al. (2018), but is considered to be generally appropriate. Therefore, the effect of aerosol is considered to appear in the extinction coefficient of no rainfall in patches where the scene depth is more than approx. 100 m. However, in patches where scene depth $d$ is less than approx. 100 m, the results show a significant overestimate compared to the previous studies as well as the case of rainfall.

### 5.2.5 Causes of overestimates of extinction coefficients obtained from images

In patches where scene depth is less than approx. 100 m, the extinction coefficients calculated from images resulted in overestimates, regardless of the presence or absence of rain. This implies that the static weather effect was strongly represented in the image, contrary to the fact, even though the static weather effect was actually absent or small. One possible reason for this could be the influence caused by DCP, the method used in this study to calculate extinction coefficients. DCP assumes that dark channel images of the outdoor images without static weather effects will have zero pixel values in most patches and that transmission will decrease according to an increase in scene depth and static weather effects (rainfall intensity in this study) (He et al., 2011). In other words, it is assumed that the increase in scene depth and static weather effects will make the image whiter. Therefore, although DCP can properly determine transmission $t$ if the background of the image meets the assumption, it has been pointed out that there are many actual outdoor images that violate the assumption, and it is often difficult to estimate the appropriate transmission $t$ (Qin et al., 2020; Qu et al., 2019; Ren et al., 2018; Wu et al., 2020). It has been reported that especially in backgrounds with white objects that are essentially similar to the color of global atmospheric light, DCP often fails because it violates the assumed prior distribution (Qin et al., 2020; Ren et al., 2018; Yang and Sun, 2018).

In Figure 9 and Figure 11, the closer the ratio of scene radiance $J$ to global atmospheric light $A$ is to 1, the more the background has a color that is essentially similar to the color of global atmospheric light, and the more difficult it is to estimate transmission $t$ by DCP. From Figures 9 and 11, it can be seen that in all the cameras and all rainfall intensity Figures, the values of the ratio of scene radiance $J$ to global atmospheric light $A$ in the patches within approx. 100 m of scene depth are larger than in the patches above approx. 100m of scene depth. Therefore, many patches within approx. 100 m of scene depth were likely to violate the assumption of the expected prior distribution, which suggests that it was an inconvenient patch for the estimation of transmission. This indicates that the cause of the overestimates of the value of the extinction coefficient in these patches was due to the misidentification of the white-colored background as a static weather effect, which tends to violate the DCP's assumption of prior distribution.

It has been pointed out that the ambiguity between image color and scene depth is often a problem with image fog removal techniques such as the one referenced in this study (Meng et al., 2013). In other words, the inability to determine whether the whiteness of the image is due to the color of the background object itself or to the increase in scene depth is an issue for the techniques to remove static weather effects. Therefore, it is important to consider in advance the reason for the whiteness of the image, even with the method proposed in this study. Since some techniques have been proposed to express Eq. (1) from images (e.g., Fattal, 2008; Tan, 2008) in addition to the method using DCP, it is a future issue to study which method can be used to obtain appropriate extinction coefficients and transmission.

Furthermore, In Figures 9 and 11, some plots overestimate extinction coefficients even if the value of the ratio of scene radiance $J$ to global atmospheric light $A$ is not necessarily larger, especially in the Figures with higher rainfall intensity.

Therefore, it can be inferred that the cause of the overestimates of extinction coefficients is not only due to the effect caused by DCP. At present, other causes have not yet been identified, and the issue in the future is to determine these causes.

## 5.3 Estimates of rainfall intensity

Based on the previous discussion, we attempted to estimate rainfall intensity using Eq. (9), which determines rainfall
intensity from image information. In Eq. (9), the parameters needed to estimate the rainfall intensity $R$ are extinction efficiency $Q$, global atmospheric light $A$, observed intensity $I$, scene radiance $J$, and scene depth $d$. Concerning the extinction efficiency $Q$, as shown in 5.2.3, the value of parameter $a$ in Eq. (10) was proposed to be $5.80 \times 10^{-5} \pi Q$ using extinction efficiency $Q$ in this study. On the other hand, previous studies proposed the value of parameters $a$ of $2.12 \times 10^{-4}$ (Nedvidek et al. 1986; Ulbrich and Atlas, 1985). Therefore, assuming that the values of both parameters $a$ are identical, the following
equations obtain the extinction efficiency $Q$.

$$5.80 \times 10^{-5} \pi Q = 2.12 \times 10^{-4} \tag{11}$$

$$\therefore \quad Q = \frac{2.12 \times 10^{-4}}{5.80 \times 10^{-5} \pi} \approx 1.16 \tag{12}$$

The same values used in the previous discussion were applied for global atmospheric light $A$, observed intensity $I$, scene radiance $J$, and scene depth $d$. The flow for estimating rainfall intensity is shown in Figure 4.
Figure 12-a, 12-b, and 12-c show the relationships between the observed and estimated rainfall intensity for each camera. Figure 12-a, 12-b, and 12-c show that there are patches where the observed and estimated rainfall intensities generally coincide, such as patch 42 in Camera 1, patch 29 in Camera 2, and patch 39 in Camera 3, suggesting that it is possible to estimate the rainfall intensity from the image. These example patches are those with the lowest mean absolute percentage error (MAPE) of rainfall intensity estimates in cases using data with observed rainfall intensity of 0.2 mm min$^{-1}$ or greater
throughout the observation period. Furthermore, in many of the patches with scene depths of less than 100 m hatched in yellow, the estimated rainfall intensity was overestimated. This may be due to the overestimation of the extinction coefficients, as we have mentioned before. Similarly, patches 12, 13, 17, 18, and 19 in Camera 2 also overestimate the estimated rainfall intensity due to overestimation of the extinction coefficient. This suggests that to estimate rainfall intensity from an image, it is necessary to select an appropriate background for which the extinction coefficient is not overestimated
or underestimated.

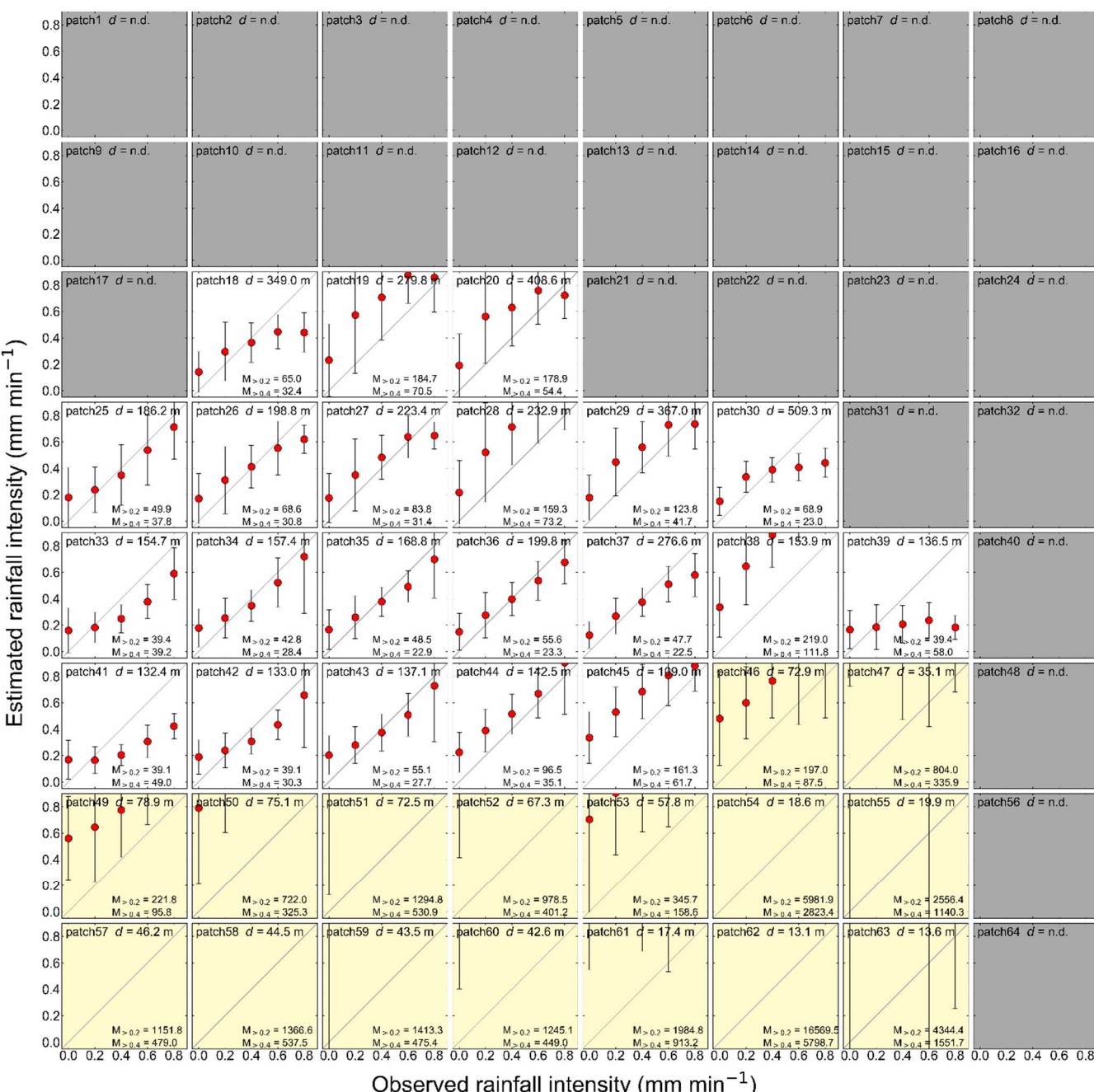

**Figure 12-a. Relationship between observed rainfall intensity and estimated rainfall intensity of Camera 1. Each figure is marked with a patch number, scene depth and two MAPE values of rainfall intensity estimates throughout the observation period. $M_{>0.2}$ means MAPE value in cases using data with observed rainfall intensity of 0.2 mm min$^{-1}$ or greater and $M_{>0.4}$ means MAPE value in cases using data with observed rainfall intensity of 0.4 mm min$^{-1}$ or greater. The plots and error bars show the mean value and standard deviation of all data during the observation**

period. Patches hatched in yellow indicate patches with scene depth of less than 100 m. Patches hatched in gray are patches where the appropriate scene depth could not be obtained due to the presence of sky background and the 635 application of geometric corrections in the image registration process.

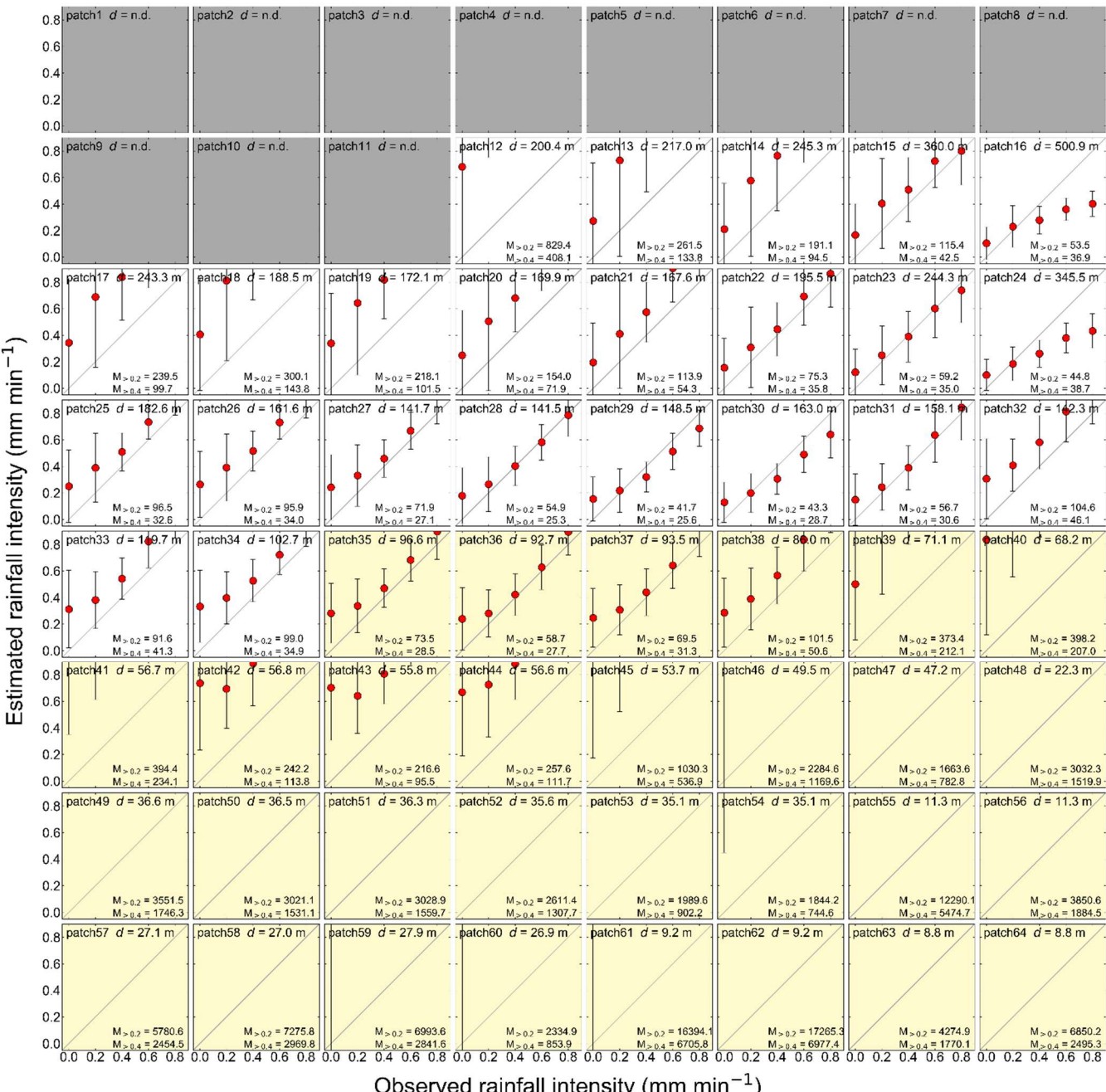

**Figure 12-b. Relationship between observed rainfall intensity and estimated rainfall intensity of Camera 2. Each figure is marked with a patch number, scene depth and two MAPE values of rainfall intensity estimates throughout the observation period. $M_{>0.2}$ means MAPE value in cases using data with observed rainfall intensity of 0.2 mm min$^{-1}$ or greater and $M_{>0.4}$ means MAPE value in cases using data with observed rainfall intensity of 0.4 mm min$^{-1}$ or**

**greater. The plots and error bars show the mean value and standard deviation of all data during the observation period. Patches hatched in yellow indicate patches with scene depth of less than 100 m. Patches hatched in gray are patches where the appropriate scene depth could not be obtained due to the presence of sky background and the application of geometric corrections in the image registration process.**

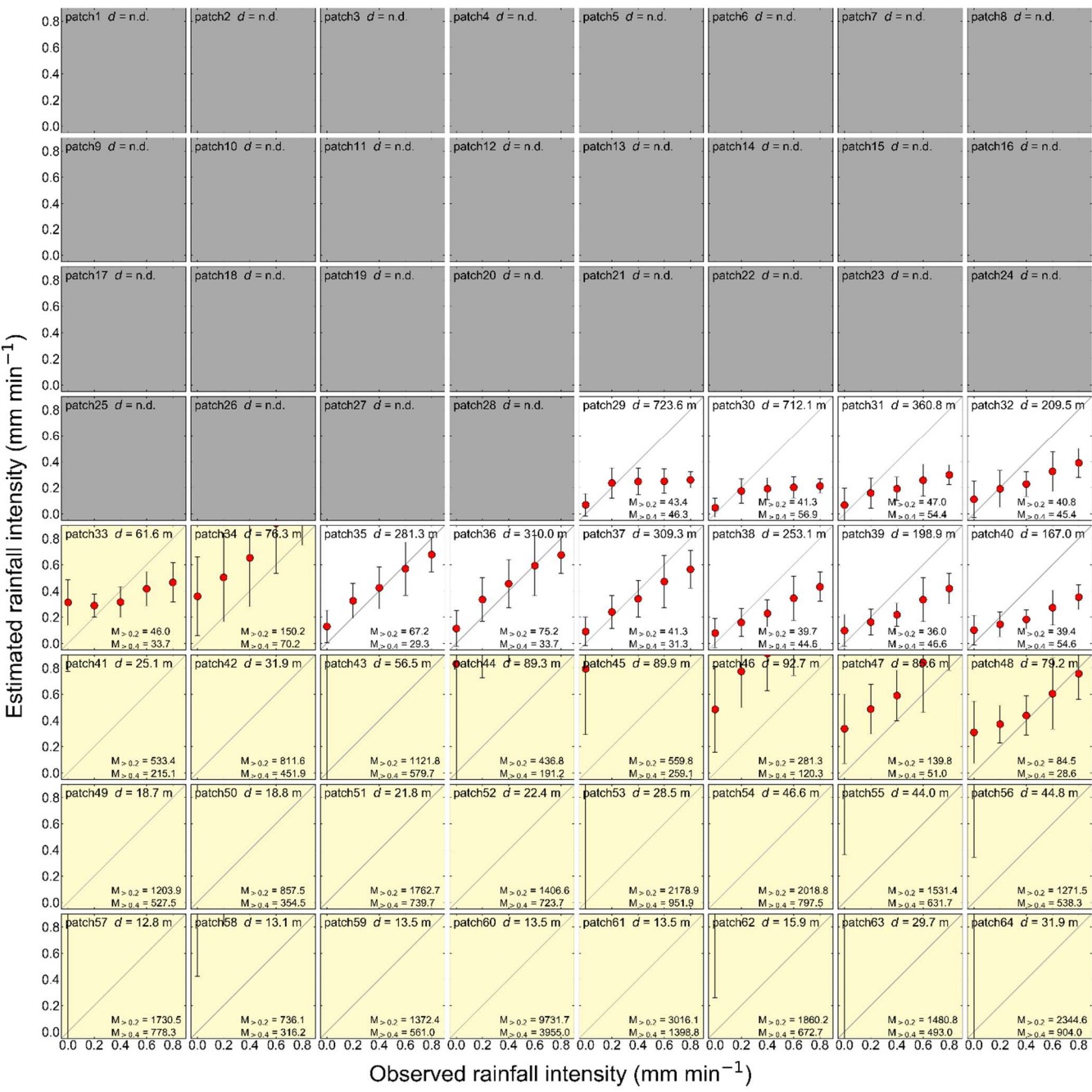

**Figure 12-c. Relationship between observed rainfall intensity and estimated rainfall intensity of Camera 3. Each figure is marked with a patch number, scene depth and two MAPE values of rainfall intensity estimates throughout the observation period. $M_{>0.2}$ means MAPE value in cases using data with observed rainfall intensity of 0.2 mm min$^{-1}$**

or greater and $M_{>0.4}$ means MAPE value in cases using data with observed rainfall intensity of 0.4 mm min$^{-1}$ or greater. The plots and error bars show the mean value and standard deviation of all data during the observation period. Patches hatched in yellow indicate patches with scene depth of less than 100 m. Patches hatched in gray are patches where the appropriate scene depth could not be obtained due to the presence of sky background and the application of geometric corrections in the image registration process.

Figure 13 shows the time series variation of rainfall intensity estimates for the three rain events for the patch with the lowest MAPE for each camera: patch 42 in Camera 1, patch 29 in Camera 2, and patch 39 in Camera 3. The scene depth of patch 42 in Camera 1, patch 29 in Camera 2, and patch 39 in Camera 3 were respectively 133.0 m, 148.5 m, and 198.9 m. The background of all these patches was vegetation. The rain events shown in Figure 13 are those with the maximum one-minute rainfall intensity of 0.8 mm min$^{-1}$ throughout the observation period. The time series variation of rainfall intensity estimates

for all camera patches during these rain events were stored at the storage locations indicated in the Supplement. In Figure 13, during the period when the one-minute rainfall intensity was observed to be 0.4 mm min$^{-1}$ or greater for each rain event, it can be seen that the estimated rainfall intensity variation for all cameras followed the observed rainfall intensity variation, although the absolute values varied slightly. Therefore, it can be said that this method can capture short-term variations in rainfall intensity.

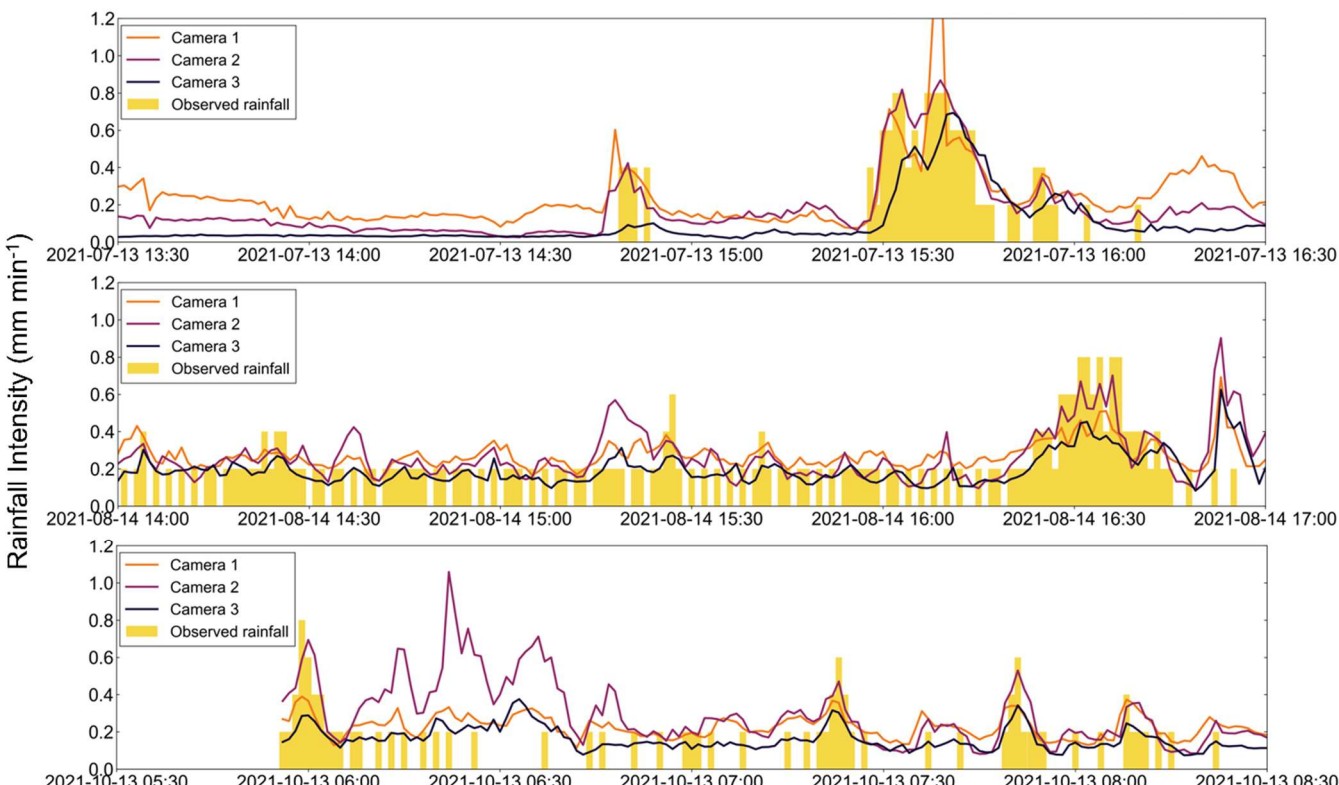

**Figure 13. Time series variation of observed and estimated rainfall intensity. The patch for each camera is the patch with the lowest MAPE of the rainfall intensity estimate in cases using data with observed rainfall intensity of 0.2 mm**

**min-1 or greater throughout the observation period, with patches 42 in Camera 1, 29 in Camera 2, and 39 in Camera 3, respectively.**

Table 3 shows the results of the comparison of the accuracy between the five previous studies (Allamano et al., 2015; Dong et al., 2017; Jiang et al., 2019; Yin et al., 2023; Zheng et al., 2023) and this study. All five of these previous studies focused on the dynamic weather effects of rainfall, and no studies have been conducted on the static weather effects caused by rain. Allamano et al. (2015) and Dong et al. (2017) identified rain streaks on images based on temporal properties, excluded unfocused rain streaks, and estimated rainfall intensity from the identified rain streak information. Jiang et al. (2019) incorporated visual properties in addition to temporal properties in identifying rain streaks on images. Yin et al. (2023) estimated rainfall intensity by constructing an image-based supervised convolutional neural network model called irCNN. Zheng et al. (2023) estimated rainfall intensity by constructing a two-stage algorithm that extract raindrop information from the image and then perform convolutional neural networks using the extracted raindrop information as inputs. Table 3 shows that although only Jiang et al. (2019) was conducted in Monsoon influenced humid subtropical climate, this study and all five previous studies were conducted in a humid subtropical climate, and that there are no significant climatic differences. Furthermore, regarding the mean rainfall intensity during the observation period, the mean rainfall intensities of Allamano et al. (2015) and Dong et al. (2017) were slightly lower than those in this study, while the mean rainfall intensities of Jiang et al. (2019), Yin et al. (2023), and Zheng et al. (2023) were comparable to this study. Overall, there is no significant difference in mean rainfall intensity between this study and the five previous studies. Moreover, in all studies, MAPE, a metrics of model performance, was calculated from observed values and model prediction. Given these facts, it seems reasonable to compare this study with the five previous studies. All five of these studies did not separate the patches. This is because these previous studies focused on the dynamic weather effects and scene depth was not relevant. As shown in Table 3, the mean value of MAPE using data with observed rainfall intensity of 0.2 mm min$^{-1}$ or greater for all patches was 1163.4 %, 2131.4 %, and 1087.2 % for Camera 1, Camera 2, and Camera 3, respectively, and the median value of MAPE using data with observed rainfall intensity of 0.2 mm min$^{-1}$ or greater for all patches was 170.1 %, 242.2 %, and 546.6 % for Camera 1, Camera 2, and Camera 3, respectively. The mean value of MAPE was considerably larger than the median value of MAPE, because it was heavily influenced by larger values such as the maximum value of MAPE. On the other hand, the mean value of MAPE using data with observed rainfall intensity of 0.2 mm min$^{-1}$ or greater for patches with scene depth of more than 100 m was 88.9 %, 148.3 %, and 47.1 % for Camera 1, Camera 2, and Camera 3, respectively, and the median value of MAPE using data with observed rainfall intensity of 0.2 mm min$^{-1}$ or greater for all patches was 65.0 %, 96.5 %, and 41.3 % for Camera 1, Camera 2, and Camera 3, respectively. Thus, the results indicate that the accuracy of rainfall intensity estimation can be improved by restricting the data to patches with scene depth of more than 100 m. Therefore, it is important to select patches with a scene depth of more than 100 m for rainfall intensity estimation. Next, we compare results of patches with scene depth of more than 100 m in this study with results of the five previous studies. The median, 25th percentile, and minimum value of MAPE using data with observed rainfall intensity of 0.2 mm min$^{-1}$ or greater for patches with scene depth of more than 100 m was higher than the MAPE value in the five previous studies. In contrast, the median value of MAPE using data with observed rainfall intensity of 0.4 mm min$^{-1}$ or greater for patches with scene depth of more than 100 m was slightly

higher than the MAPE value in the five previous studies, but the 25th percentile and minimum value of MAPE using data with observed rainfall intensity of 0.4 mm min$^{-1}$ or greater for patches with scene depth of more than 100 m was similar to those of the five studies. Therefore, the proposed method in this study is considered to have a certain degree of effectiveness as a rainfall intensity estimation method, although there may be some error when the rainfall intensity is small. The proposed method is also considered to be sufficiently robust because it was validated for all rain events with observed rainfall intensity of 0.2 mm min$^{-1}$ or greater during the 235-day observation period in this study. In addition, the similarity of the estimated rainfall intensity variations for all cameras suggests that the proposed method is sufficiently versatile.

**Table 3. Comparison of accuracy between the five previous studies and this study: The "City, Country" row indicates the city and country where each study was conducted, and the "Köppen climate classification" row indicates the Köppen climate classification of the city. In the "Used data" row, "> 0.2 mm min$^{-1}$" means that data with observed rainfall intensity of 0.2 mm min$^{-1}$ or greater were used, and "> 0.4 mm min$^{-1}$" means that data with observed rainfall intensity of 0.4 mm min$^{-1}$ or greater were used. The "Observed rainfall intensity" row indicates the mean, maximum and minimum rainfall intensity during the observation period for each study. The range of "Observed rainfall intensity" indicates the range due to multiple rain events in each study. The rainfall intensity of Jiang et al. (2019) was converted from the duration and the accumulated rainfall of the rain event. MAPE is the mean absolute percentage error, and data with observed rainfall intensity of 0 mm min$^{-1}$ were excluded by the definition of MAPE. MAPE values are shown for the case where all patches were used and for the case where only patches with a scene depth of more than 100 m were used. The number of patches used for each camera is shown in parentheses. In the column of the minimum MAPE, the patch number indicating the minimum value is shown in parentheses.**

| | This study | | Allamano et al. (2015) | Dong et al. (2017) | Jiang et al. (2019) | Yin et al. (2023) | Zheng et al. (2023) |
|---|---|---|---|---|---|---|---|
| City, Country | Yamanashi, Japan | | Torino, Italy | Nanjing, China | Shenzhen, China | Hangzhou, China | Hangzhou, China |
| Köppen climate classification | Humid subtropical climate (Cfa) | | Humid subtropical climate (Cfa) | Humid subtropical climate (Cfa) | Monsoon influenced humid subtropical climate (Cwa) | Humid subtropical climate (Cfa) | Humid subtropical climate (Cfa) |
| Used data | > 0.2 mm min$^{-1}$ | > 0.4 mm min$^{-1}$ | - | - | - | - | - |

| Camera name | | Camera 1 | Camera 2 | Camera 3 | Camera 1 | Camera 2 | Camera 3 | - | - | - | - | - |
|---|---|---|---|---|---|---|---|---|---|---|---|---|
| Data size for validation: Video length (min) | | 3261 | 3015 | 3261 | 120 | 107 | 120 | 104 | 9 | 403 | 170 | 357 |
| Observed rainfall intensity (mm h$^{-1}$) | Mean | 12.6 | 12.6 | 12.6 | 28.5 | 28.9 | 28.5 | 2.8 - 9.3 | 1.1 - 6.5 | 4.1 - 29.5 | 11.0 - 23.6 | 9.7 - 39.3 |
| | Maximum | 48.0 | 48.0 | 48.0 | 48.0 | 48.0 | 48.0 | 6.0 - 38.2 | - | - | 36.0 - 66.0 | 24.0 - 156.0 |
| | Minimum | 12.0 | 12.0 | 12.0 | 24.0 | 24.0 | 24.0 | 1.3 - 3.2 | - | - | - | - |
| Accuracy: MAPE (%) | All patches (Camera 1: 37 patches, Camera 2: 53 patches, Camera 3: 36 patches) — Mean | 1163.4 | 2131.4 | 1087.2 | 459.7 | 923.8 | 466.3 | | | | | |
| | Maximum | 5981.9 | 12290.1 | 2178.9 | 2823.4 | 5474.7 | 951.9 | | | | | |
| | 75th percentile | 783.5 | 1989.6 | 1296.7 | 333.2 | 853.9 | 565.7 | | | | | |
| | Median | 170.1 | 242.2 | 546.6 | 66.1 | 111.7 | 237.1 | | | | | |
| | 25th percentile | 55.6 | 95.9 | 62.2 | 32.4 | 36.9 | 49.9 | | | | | |
| | Minimum | 39.1 (patch 42) | 41.7 (patch 29) | 36.0 (patch 39) | 22.5 (patch 37) | 25.3 (patch 28) | 28.6 (patch 48) | | | | | |
| | Patches with scene depth of more than 100 m (Camera 1: 21 patches, Camera 2: 23 patches, Camera 3: — Mean | 88.9 | 148.3 | 47.1 | 43.1 | 70.6 | 44.3 | 26.0 | 31.8 | 21.8 | 13.5 - 21.9 | 10.4 - 18.0 |
| | Maximum | 219.0 | 829.4 | 75.2 | 111.8 | 408.1 | 56.9 | | | | | |
| | 75th percentile | 123.8 | 172.5 | 46.1 | 54.4 | 83.2 | 52.5 | | | | | |
| | Median | 65.0 | 96.5 | 41.3 | 35.1 | 38.7 | 45.9 | | | | | |
| | 25th percentile | 47.7 | 58.0 | 40.0 | 28.4 | 33.3 | 36.4 | | | | | |
| | Minimum | 39.1 (patch 42) | 41.7 (patch 29) | 36.0 (patch 39) | 22.5 (patch 37) | 25.3 (patch 28) | 29.3 (patch 35) | | | | | |

## 5.4 Ways forwards

**5.4.1 Limitation of the proposed method**

There are still several technical problems that need to be solved in the method of this study. The first problem is how to select an appropriate background for rainfall intensity estimation (i.e., the analysis area to be used for rainfall intensity estimation). As shown in Table 3, the accuracy of rainfall intensity estimation varies greatly depending on the background patch selected. Therefore, background patches with the highest estimation accuracy possible should be selected. One solution

to this problem is to select patches with a scene depth of more than 100 m. As shown in Table 3, selecting analysis regions from patches with scene depth of more than 100 m is more accurate overall than selecting analysis regions from all background patches. On the other hand, it may also be important that the scene depth is not too large because even relatively small rainfall intensity may cause the transmission to reach the lower limit as shown in Figure 8. It is necessary to further study in detail what scene depth is appropriate for rainfall intensity estimation. In addition, in terms of background objects, a

relatively undisturbed background is desirable for the analysis area. Therefore, it is preferable to choose a static background such as building walls, tree canopies, and ground surface without people or vehicles, especially when applying this method in urban areas. However, at this time, the selection of appropriate backgrounds has not been analyzed in detail, and further study is needed on the effects of scene depth, background texture, and dynamic subject exposure on estimation accuracy.

The second problem is how to remove the effects of dew formation and raindrops on the camera lens itself from the image.

Dew formation and raindrops on the camera lens itself could cause significant blurring of the image and affect the rainfall estimation results, but this effect has not been analyzed at this time. Therefore, it is necessary to consider how to physically protect the camera lens (e.g., by covering the camera with a cover) and how to remove the effect from the image if dew or raindrops get on the lens.

The third problem is the identification of fog and precipitation types (e.g., rain, snow). Figure 13 shows that the variation of

755 the estimated rainfall intensity of Camera 2 around 6:30 on October 13 was different from that of the observed rainfall intensity. The images from Camera 2 during this period were validated to be foggy in the selected patches. Therefore, the variation in the estimated rainfall intensity for Camera 2 can be attributed to the whitening of the background due to fog. Since this method estimates rainfall intensity from image whiteness, image whiteness caused by fog is misidentified as the effect of rainfall. At present, however, there is no method to determine whether it is fog or rain. Therefore, as a further study,

it is necessary to investigate a method to determine whether the whiteness in the image under bad weather conditions is caused by rain or fog.

Finally, the fourth problem is the development of a nighttime rainfall estimation method. The method of this study is not applicable to nighttime images because it was difficult to distinguish rainfall. Therefore, rainfall estimation methods using nighttime images should be also considered separately. An idea for a rainfall estimation method using nighttime images is to use dynamic weather effects, such as counting the number of rain streaks that appear around the light source or near the lens, if the image is illuminated at night. Furthermore, recently, methods using infrared and near-infrared cameras to estimate rainfall intensity at night have also been proposed, and such methods can be utilized (Lee et al., 2023; Wang et al., 2023). Thus, there are still several technical problems in the method of this study.

### 5.4.2 Possibility for practical use

The camera used in this study was a relatively inexpensive commercially available outdoor camera (approximately 300 US dollars per unit at the time of purchase), and cameras with similar performance have become even less expensive in recent years. Although the durability of the camera needs to be validated in the future, it is expected that data acquisition will be possible at the same level or lower cost than that of a traditional tipping bucket rain gauge. Furthermore, cameras have already been installed outdoors for various purposes other than rainfall observation. The proposed method in this study can utilize images even without a special installation environment for rainfall observation purposes if there is a certain distance to the background and the background is relatively undisturbed. In other words, it is expected that by effectively utilizing images from existing cameras, it will be possible to acquire a vast amount of rainfall data on the ground surface. Therefore, this method potentially become a gap filler for areas in lacking surface rainfall observations. Moreover, if past images have been accumulated, it may be possible to go back in time and recover surface rainfall data. On the other hand, data processing time may be an issue in utilizing the data for real-time observations. However, the proposed method is extremely simple, requiring less than one minute to process one image using a typical commercial computer. Although we use the computer having specifications of 80 GB RAM and Intel core i7-10700 @2.90 GHz CPU in this study, such RAM capacity is not necessary for this process. In other words, it is considered that instantaneous rainfall intensity can be estimated with a time resolution of one minute or less using a typical commercial computer. Therefore, there is potential for various fields where rainfall observation can be effectively utilized, such as countermeasures against flash flood and debris flow, flood forecasting, and irrigation system operation, from a cost perspective. However, there are still several technical problems to be addressed to take advantage of this method, as indicated in 5.4.1. Furthermore, there are concerns about privacy issues in the actual use of this method. In many outdoor surveillance cameras, it may be inevitable that persons will be captured. Therefore, when making data public, it is necessary to pay careful attention to privacy issues. Thus, it is important to understand that there are technical problems and privacy issues before practically using this method.

## 6 Conclusions

In this study, to verify the applicability of existing theories to static weather effects caused by rain in outdoor photography systems, we analyzed the effects of rainfall intensity on the appearance of the background. Using the extinction coefficient as information source, we proposed relational equations representing the relationship between image information, rainfall intensity, and scene depth by linking the theoretically derived rainfall intensity with a technique proposed in the computer vision field for removing static weather effects. We also proposed a method for estimating rainfall intensity from images using those relational equations. Then, the proposed relational equations were applied to outdoor images taken by commercial interval cameras at observation sites in a mountainous watershed in Japan. As a result, the following findings were obtained.

(1) In the images taken outdoors, generally as shown in the proposed relational equations, transmission $t$ decreased exponentially according to the increase in rainfall intensity $R$ and scene depth $d$.

(2) The value and the variation of transmission $t$ according to the increase in rainfall intensity were considered to be strongly influenced mainly by the value of observed intensity $I$.

(3) Extinction coefficient $\beta$ calculated from the rainfall images was reasonable compared to the previous studies in the patches where scene depth $d$ was more than approx. 100 m.

(4) Extinction coefficient $\beta$ calculated from the no-rainfall images may have been affected by aerosols in the patches where scene depth $d$ was more than approx. 100 m. Therefore, extinction coefficient $\beta$ was not zero despite the assumption from the proposed equations.

(5) Regardless of the presence or absence of rainfall, extinction coefficients $\beta$ calculated from the images were overestimated in the patches where scene depth $d$ was less than approx. 100 m. It was suggested that one of the reasons for this was the influence caused by the method used to calculate the extinction coefficient.

(6) By selecting a background with an appropriate value for the extinction coefficient, rainfall intensity can be estimated from the image using the proposed relational equations. This method can also be used to capture short-term variations in rainfall intensity from the image.

(7) Based on the validation results of three cameras over 235 days of observations, the proposed method is considered sufficiently robust and versatile.

These findings are extremely important information regarding the rain-induced static weather effects of images and will lead to further advances in the development of camera-based rain gauges. Overall, these findings suggest that the relational equations representing the relationship between image information, rainfall intensity, and scene depth are generally effective for outdoor images. The method of estimating rainfall intensity from images using the relational equations is also effective for outdoor images. Since this method estimates rainfall intensity from a single static image, it can be applied to video cameras in principle, and real-time rainfall information can also be obtained. In addition, since the method requires little prior preparation or training data, and only uses the camera image taken of the background over a certain distance and

background scene depth information, it is a highly versatile and accessible method. In this study, the scene depth was obtained using a digital elevation model, but it would be possible to obtain the scene depth using a simpler method, such as measuring distances in a GIS. Therefore, this method potentially become a gap filler for areas in lacking surface rainfall observations. Furthermore, this method is also accurate and robust. On the other hand, there are still several problems to be studied, such as finding the details of the reasons for the overestimation of the extinction coefficient, methods to eliminate the overestimation, methods to remove the effects of aerosols, methods to select an appropriate background for rainfall intensity estimation, and methods to identify of fog and rain. Furthermore, this study examined the overall trend in the applicability of the method across the entire data set, but the specific causes of the errors in each individual image were not validated. For example, the presence of dew formation and raindrops on the camera lens itself could cause significant blurriness on the image and affect the rainfall estimation results, but this was not validated in this study. Therefore, validation of the specific causes of the errors when the proposed method is applied to each individual image is a problem to be addressed in the future. Moreover, this method is not applicable to nighttime images because it was difficult to distinguish rainfall. Therefore, rainfall estimation methods using nighttime images should be also considered separately.

Rainfall information is very important for water resource management, weather, climate, hydrological forecasting, and countermeasures against disasters caused by rainfall. Especially, in mountainous areas where flash floods and debris flow occur, for countermeasures against these disasters, it is desirable to have information on rainfall with high spatio-temporal resolution. In such areas, even if rain gauges are not installed, monitoring cameras may be in place. This study attempts to observe rainfall by effectively utilizing such cameras already installed for other purposes. We expect that our research results can be applied on a practical and real world scale in such category of disaster prevention. For this purpose, it is important to further accumulate knowledge about the effects of rainfall on images.

**Appendix A: Derivation process of Eq. (6)**

Rainfall intensity is defined as the amount of rainfall collected per unit time interval (World Meteorological Organization, 2023). Therefore, rainfall intensity is expressed as follows using the particle size distribution of raindrops, raindrop volume, and falling velocity per unit volume (Uijlenhoet, 2001).

$$R = 3.6 \times 10^6 \int_0^\infty \frac{\pi D^3}{6} N(D) U(D) \mathrm{d}D \tag{A1}$$

Where $R$ (mm h$^{-1}$) is rainfall intensity, $D$ (m) is raindrop diameter, $N(D)$ (m$^{-3}$) is the particle size distribution of raindrops, and $U(D)$ (m s$^{-1}$) is the terminal falling velocity of raindrops.

Then, as shown in 2.2, with the theory of atmospheric radiation, the extinction coefficient under rainfall conditions can be expressed as follows using the raindrop diameter, the particle size distribution of raindrops, and extinction efficiency (Grabner and Kvicera, 2011).

$$\beta = \int_0^\infty \frac{\pi D^2}{4} N(D) Q \, dD \tag{A2}$$

Where $D^2/4$ represents the surface area of raindrops projected in the optical path direction and $Q$ is extinction efficiency.

From Eqs. (A1) and (A2), both rainfall intensity and extinction coefficient can be expressed by the particle size distribution of raindrops, but analytically, rainfall intensity cannot be expressed with extinction coefficient. Therefore, the relationship between rainfall intensity and extinction coefficient is approximately related using the relational equations between rainfall intensity and particle size distribution presented by Marshall and Palmer (1948). Using the M-P distribution, the particle size distribution of raindrops can be expressed by the following equation.

$$N(D) = N_0 \exp(-\lambda D) \tag{A3}$$

$$N_0 = 8 \times 10^6 \tag{A4}$$

$$\lambda = 4.1 \times 10^3 R^{-0.21} \tag{A5}$$

Where units of $N_0$ and $\lambda$ are m$^{-4}$ and m$^{-1}$, respectively.

Substituting Eq. (A3) into Eq. (A2), we obtain:

$$\beta = \int_0^\infty \frac{\pi D^2}{4} N_0 \exp(-\lambda D) Q \, dD$$

$$= \frac{\pi N_0 Q}{4} \int_0^\infty D^2 \exp(-\lambda D) \, dD \tag{A6}$$

Here, we introduce the gamma function, which represents the generalization of the factorial.

$$\Gamma(z) = \int_0^\infty a^{z-1} \exp(-a) \, da = (z-1)! \tag{A7}$$

Applying Eq. (A7) to Eq. (A6), we obtain:

$$\beta = \frac{\pi N_0 Q}{4\lambda^3} \Gamma(3) = \frac{\pi N_0 Q}{4\lambda^3} (3-1)!$$

$$= \frac{\pi N_0 Q}{2\lambda^3} \tag{A8}$$

Substituting Eqs. (A4) and (A5) into Eq. (A8), extinction coefficient $\beta$ can be expressed as follows using rainfall intensity $R$.

$$\beta = \frac{8 \times 10^6 \pi Q}{2(4.1 \times 10^3 R^{-0.21})^3}$$

$$= 5.80 \times 10^{-5} \pi Q R^{0.63} \tag{A9}$$

## Appendix B: The procedure for the Dark Channel Prior method

He et al. (2011) defined the concept of a dark channel as follows.

$$J^{dark}(x) = \min_{y \in \Omega(x)} \left( \min_{c \in \{r,g,b\}} J^c(y) \right) \tag{B1}$$

Where $J^{dark}(x)$ is the dark channel at pixel position $x$, $\Omega(x)$ is a local patch centered at pixel position $x$, $y$ is the pixel position and an element of $\Omega(x)$, $c$ is the index of the color channel, and $J^c(y)$ is the color channel at pixel position y. The dark channel is the result of two minimum operators.

The Dark Channel Prior method is based on the statistical prior distribution in which some pixels have at least one color channel with very low intensity in almost all non-sky patches of a certain size in outdoor images without static weather effects. That is, an image that has been dilation-processed for each patch with the lowest intensity color channel values, which is called a dark channel image, is assumed to have zero pixel values in most patches. This is expressed by the following equation.

$$J^{dark}(x) = \min_{y \in \Omega(x)} \left( \min_{c \in \{r,g,b\}} J^c(y) \right) \approx 0 \tag{B2}$$

Using Eq. (B2), the first term on the right-hand side of Eq. (B3) below, which is transformed from Eq. (1), can be regarded as zero.

$$\min_{y \in \Omega(x)} \left( \min_{c \in \{r,g,b\}} \frac{I^c(y)}{A^c} \right) = t(x) \min_{y \in \Omega(x)} \left( \min_{c \in \{r,g,b\}} \frac{J^c(y)}{A^c} \right) + 1 - t(x) \tag{B3}$$

That is, Eq. (B3) is transformed into the following Eq. (B4) when Eq. (B2) is applied.

$$\min_{y \in \Omega(x)} \left( \min_{c \in \{r,g,b\}} \frac{I^c(y)}{A^c} \right) = 1 - t(x) \tag{B4}$$

Eq. (B4) can be rearranged for transmission $t$ to yield the following Eq. (B5).

$$t(x) = 1 - \min_{y \in \Omega(x)} \left( \min_{c \in \{r,g,b\}} \frac{I^c(y)}{A^c} \right) \tag{B5}$$

In Eq. (B5), $I^c(y)$ is obtained from observed intensity $I$, so transmission $t$ can be obtained by setting global atmospheric light $A$ separately. He et al. (2011) selected pixels with the top 0.1 percent intensity in the dark channel image and set the pixel with the highest intensity of observed intensity $I$ among these pixels as global atmospheric light $A$.

Scene radiance $J$ can be recovered by substituting the calculated transmission $t$ using Eq. (B5), the observed intensity $I$, and the global atmospheric light $A$, which is set separately, into Eq. (1).

**Appendix C-1: Figures including all patches showing the distribution of observed intensity $I$ by rainfall intensity. Figure C-1-a, Figure C-1-b and Figure C-1-c show the distribution of observed intensity $I$ by rainfall intensity of Camera 1, Camera 2 and Camera 3, respectively. Each figure is marked with a camera name, patch number, scene depth and slope of the linear regression line for the relationship between rainfall intensity and observed intensity $I$. The plots and error bars show the mean value and standard deviation of all data during the observation period. The**
**straight lines show linear regression line for the relationship between rainfall intensity and observed intensity $I$. Rainfall intensity is observed by the rain gauge. Patches hatched in gray are patches where the appropriate scene depth could not be obtained due to the presence of sky background and the application of geometric corrections in the image registration process.**

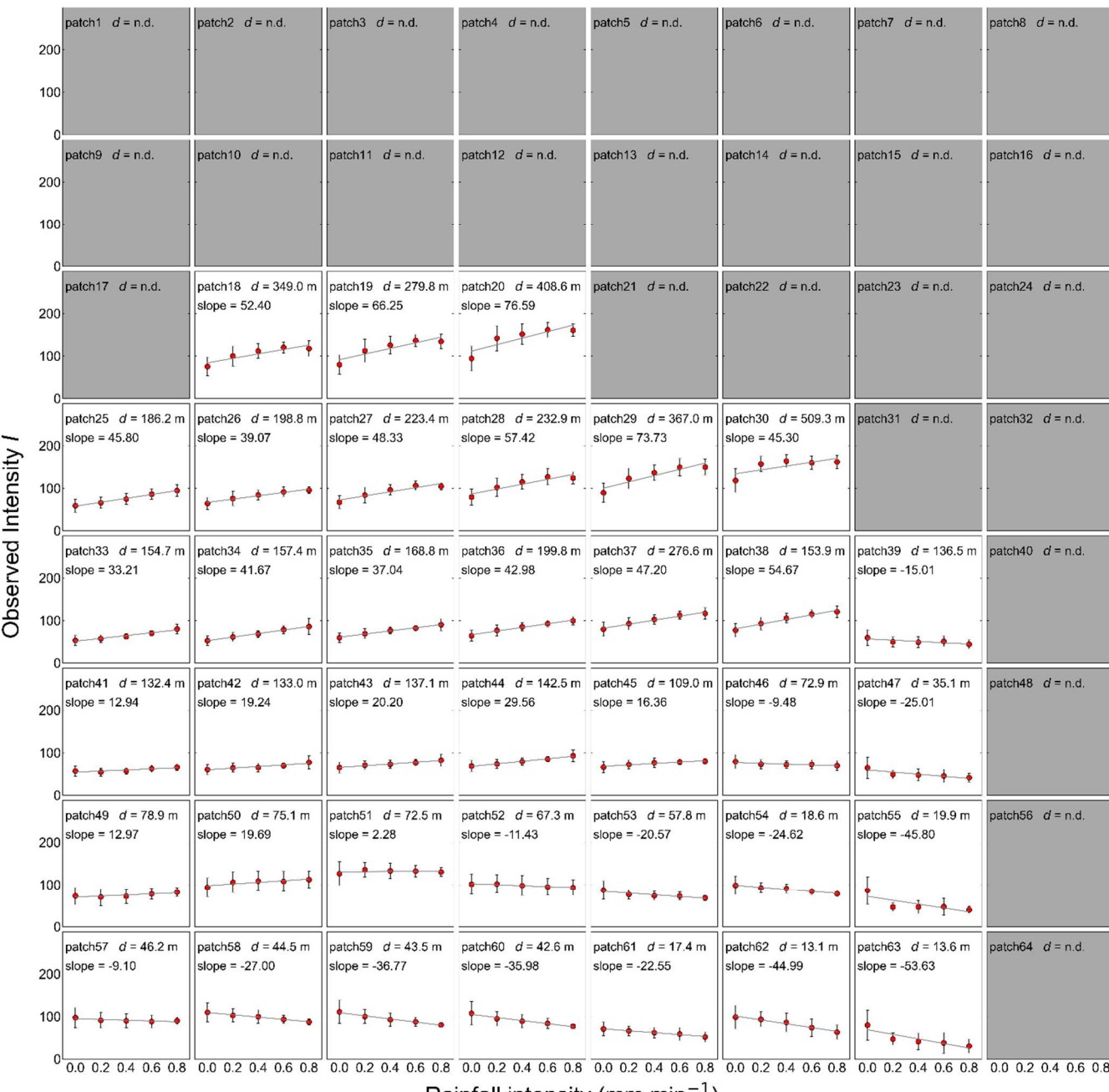

**Figure C-1-a. Distribution of observed intensity *I* by rainfall intensity of Camera 1.**

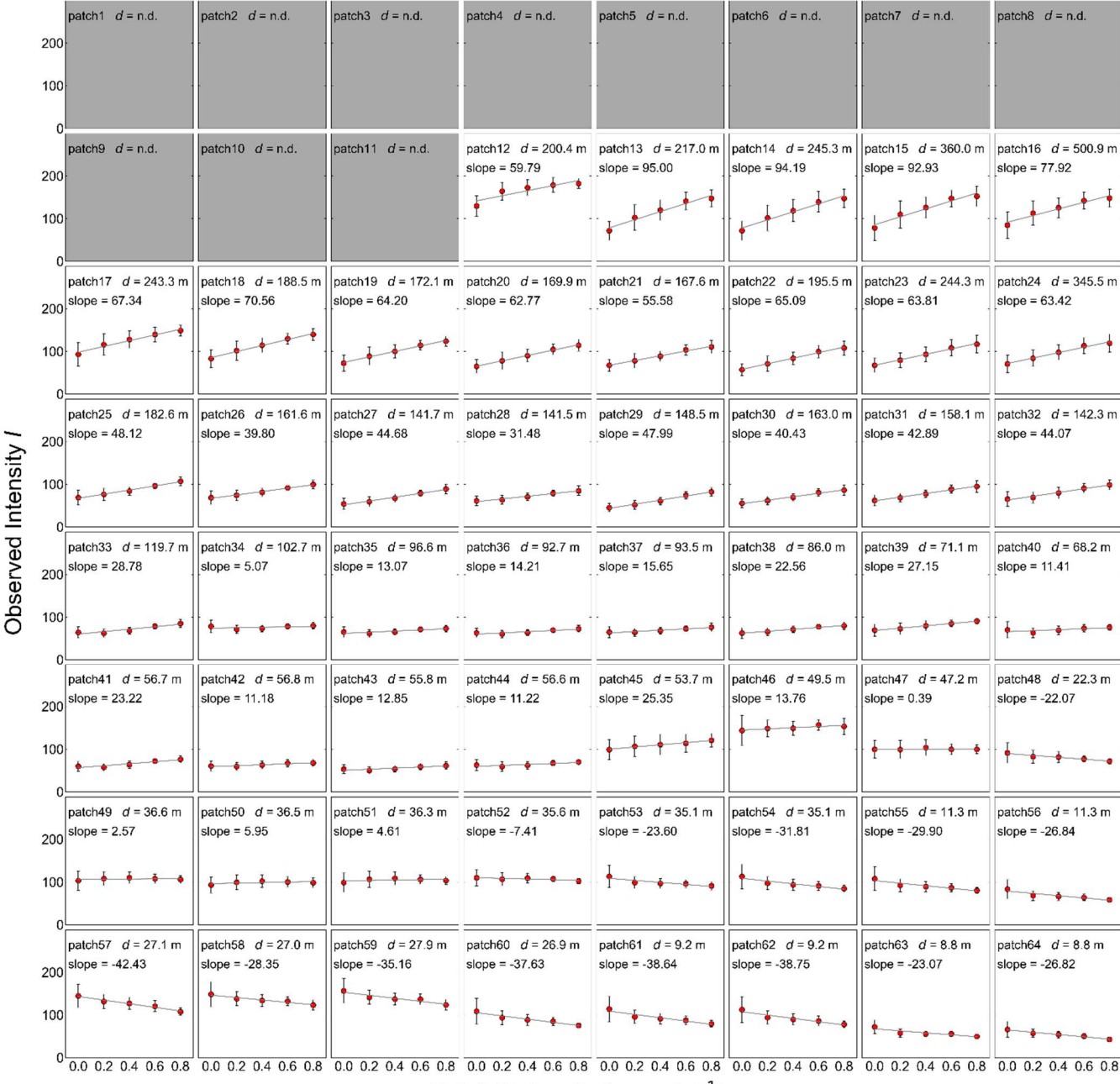

**Figure C-1-b. Distribution of observed intensity *I* by rainfall intensity of Camera 2.**


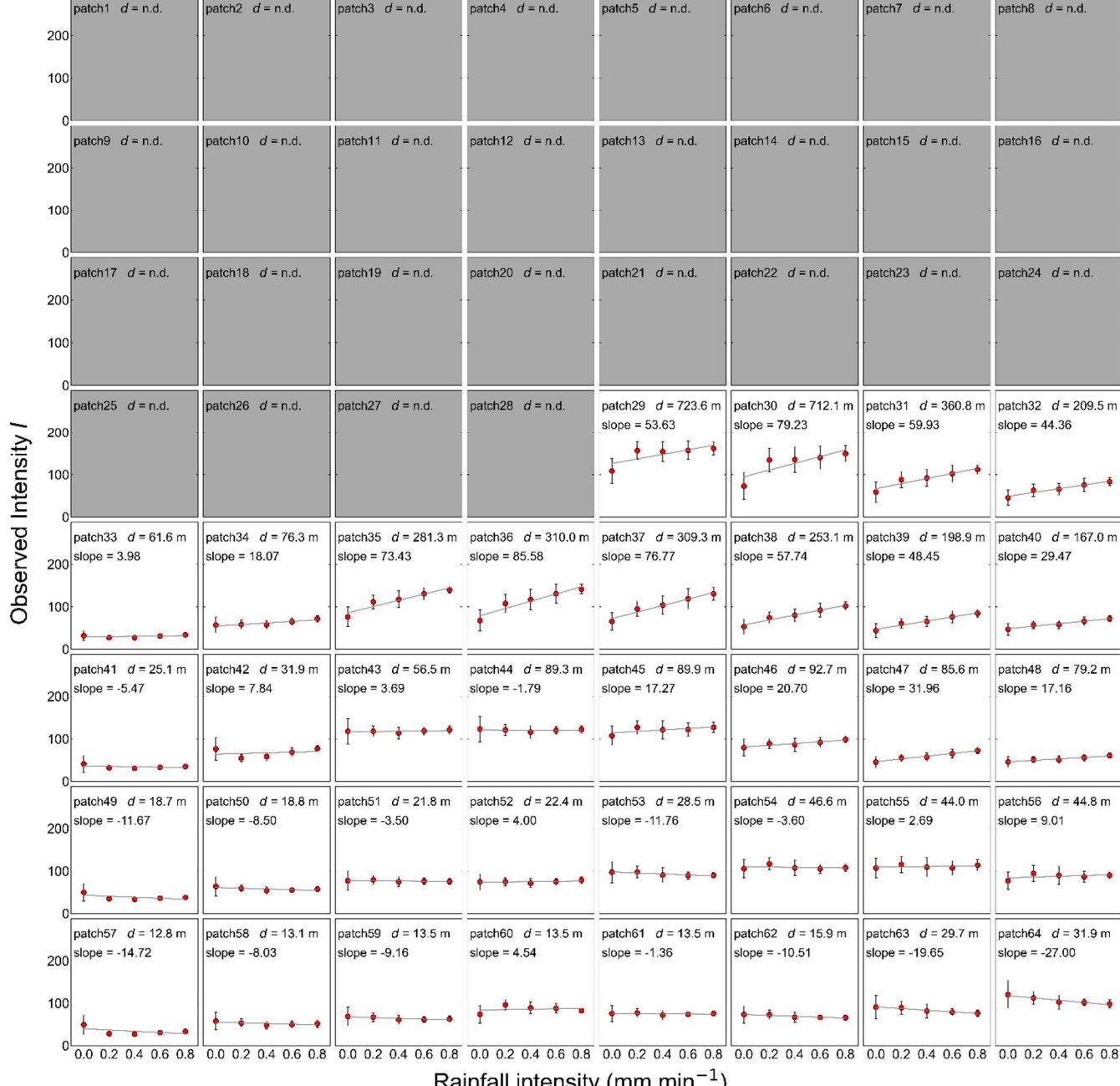

**Figure C-1-c. Distribution of observed intensity _I_ by rainfall intensity of Camera 3.**

**Appendix C-2: Figures including all patches showing the distribution of scene radiance _J_ by rainfall intensity. Figure**

**C-2-a, Figure C-2-b and Figure C-2-c show the distribution of scene radiance _J_ by rainfall intensity of Camera 1, Camera 2 and Camera 3, respectively. Each figure is marked with a camera name, patch number, scene depth and slope of the linear regression line for the relationship between rainfall intensity and scene radiance _J_. The plots and error bars show the mean value and standard deviation of all data during the observation period. The straight lines show linear regression line for the relationship between rainfall intensity and scene radiance _J_. Rainfall intensity is**

**observed by the rain gauge. Patches hatched in gray are patches where the appropriate scene depth could not be obtained due to the presence of sky background and the application of geometric corrections in the image registration process.**

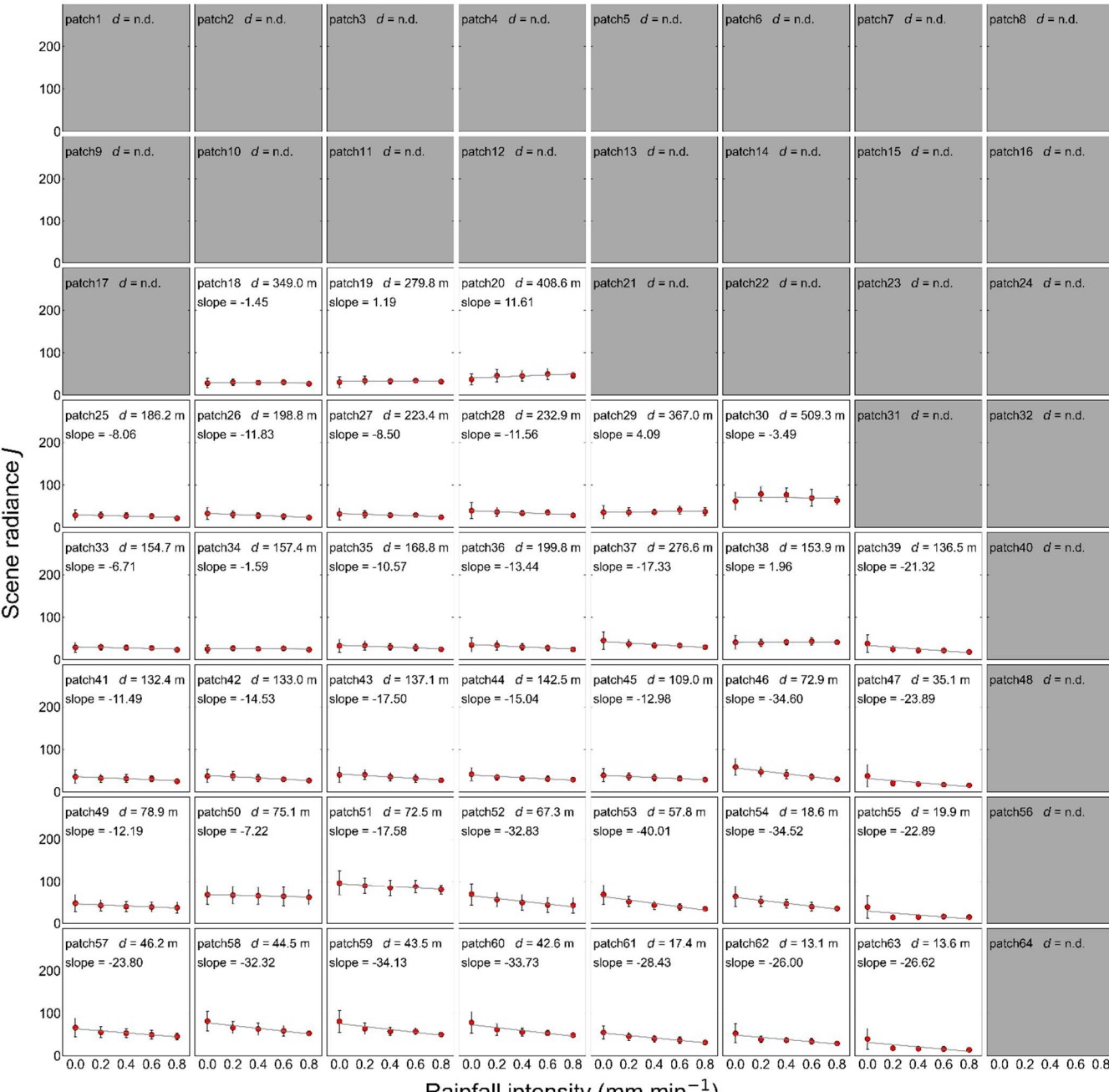

**Figure C-2-a. Distribution of scene radiance _J_ by rainfall intensity of Camera 1.**


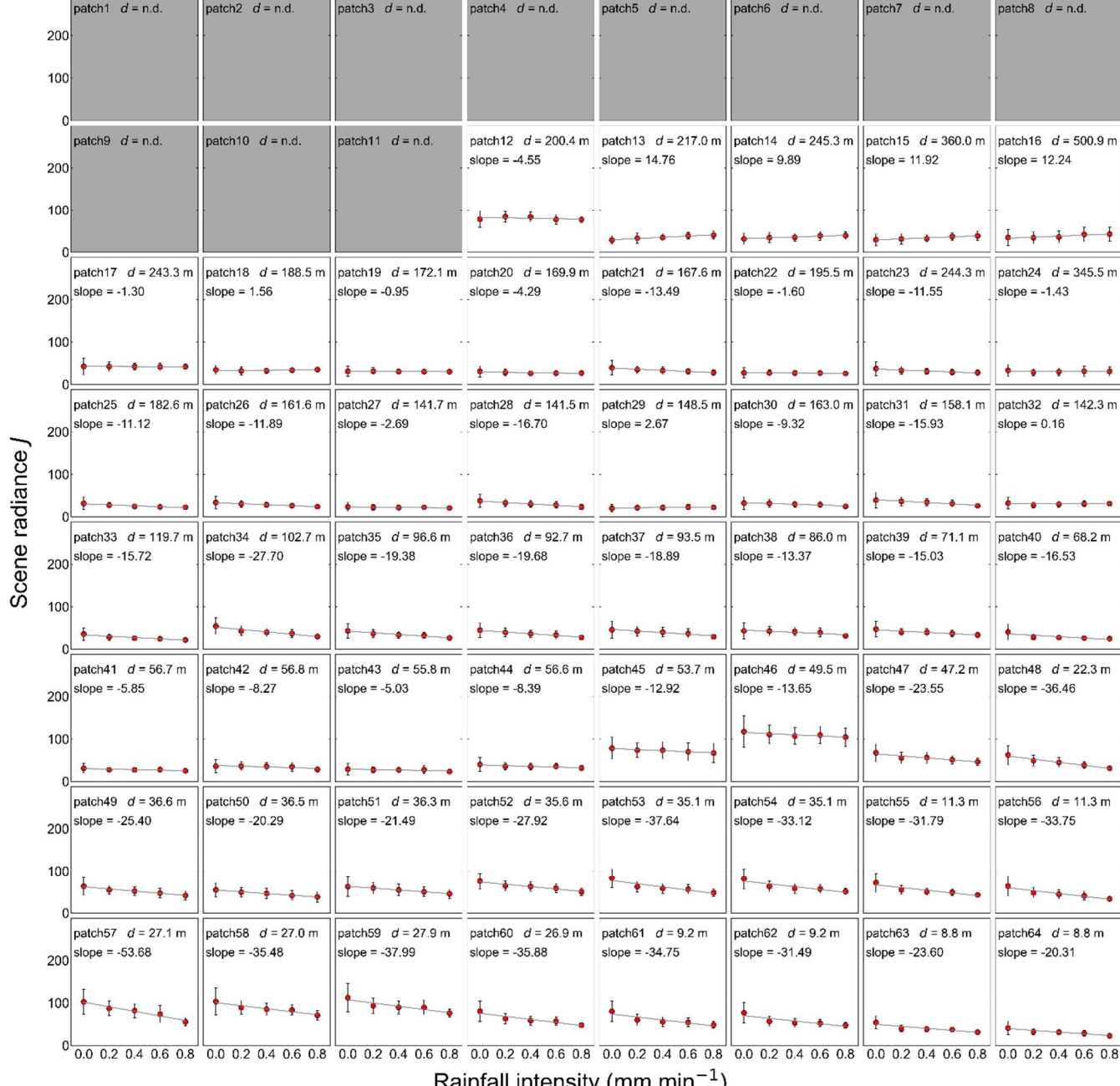

**Figure C-2-b. Distribution of scene radiance *J* by rainfall intensity of Camera 2.**

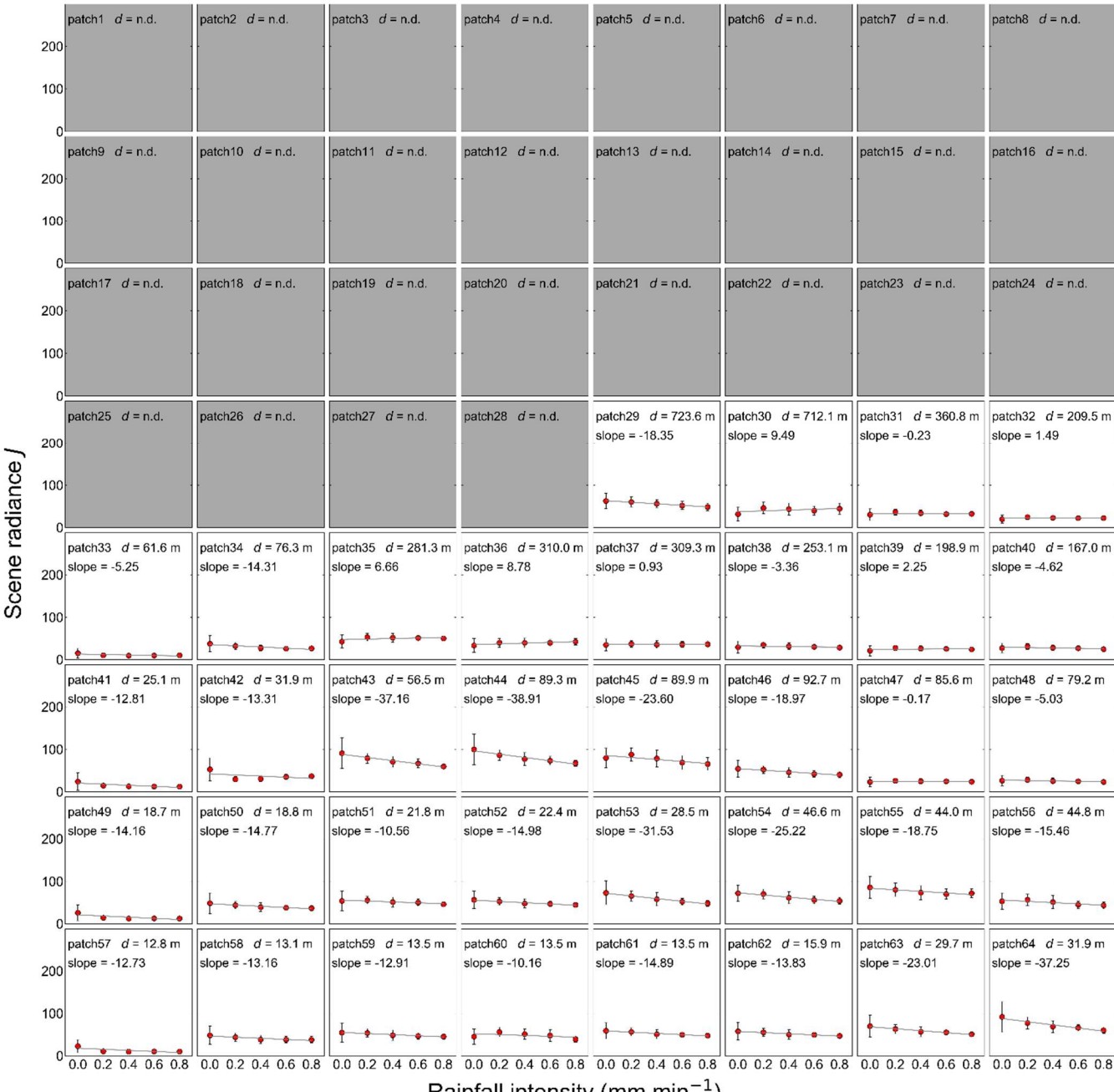

Figure C-2-c. Distribution of scene radiance *J* by rainfall intensity of Camera 3.

**Appendix C-3: Figures including all patches showing the distribution of transmission _t_ by rainfall intensity. Figure C-3-a, Figure C-3-b and Figure C-3-c show the distribution of transmission _t_ by rainfall intensity of Camera 1, Camera 2 and Camera 3, respectively. Each figure is marked with a camera name, patch number, scene depth and slope of the linear regression line for the relationship between rainfall intensity and transmission _t_. The plots and error bars show the mean value and standard deviation of all data during the observation period. The straight lines show linear regression line for the relationship between rainfall intensity and transmission _t_. Rainfall intensity is observed by the rain gauge. Patches hatched in gray are patches where the appropriate scene depth could not be obtained due to the presence of sky background and the application of geometric corrections in the image registration process.**

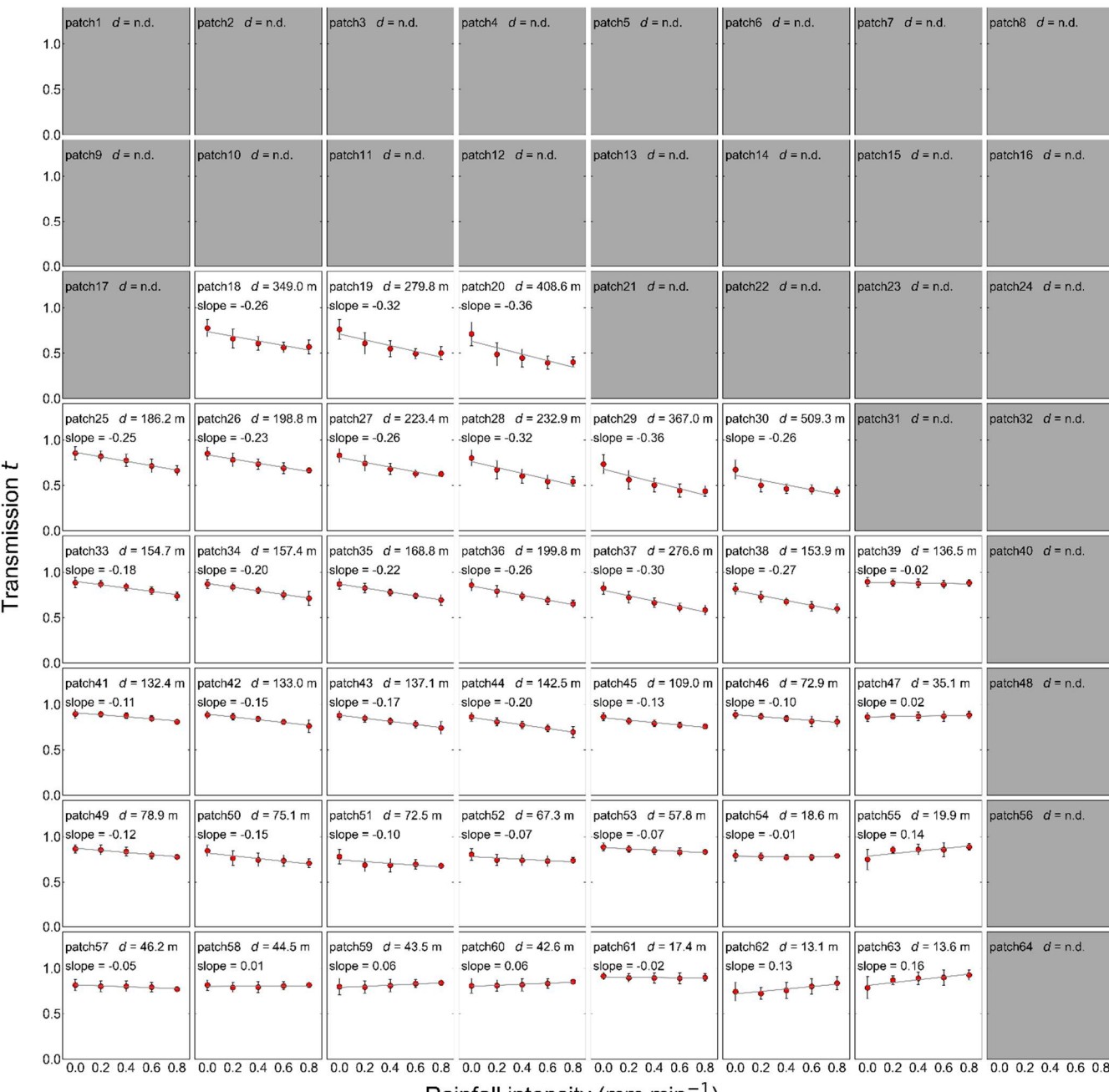


**Figure C-3-a. Distribution of transmission *t* by rainfall intensity of Camera 1.**

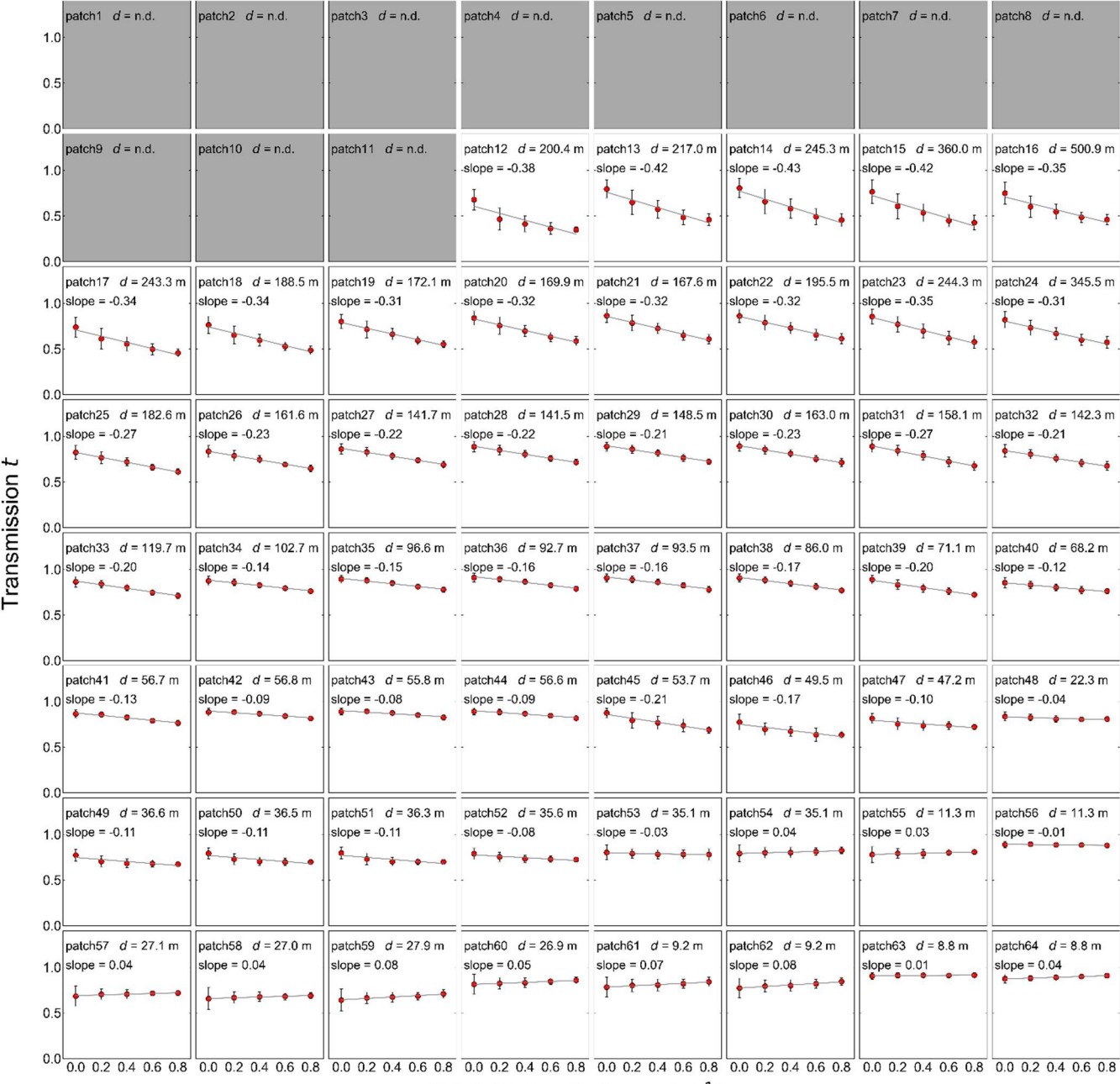

**Figure C-3-b. Distribution of transmission *t* by rainfall intensity of Camera 2.**

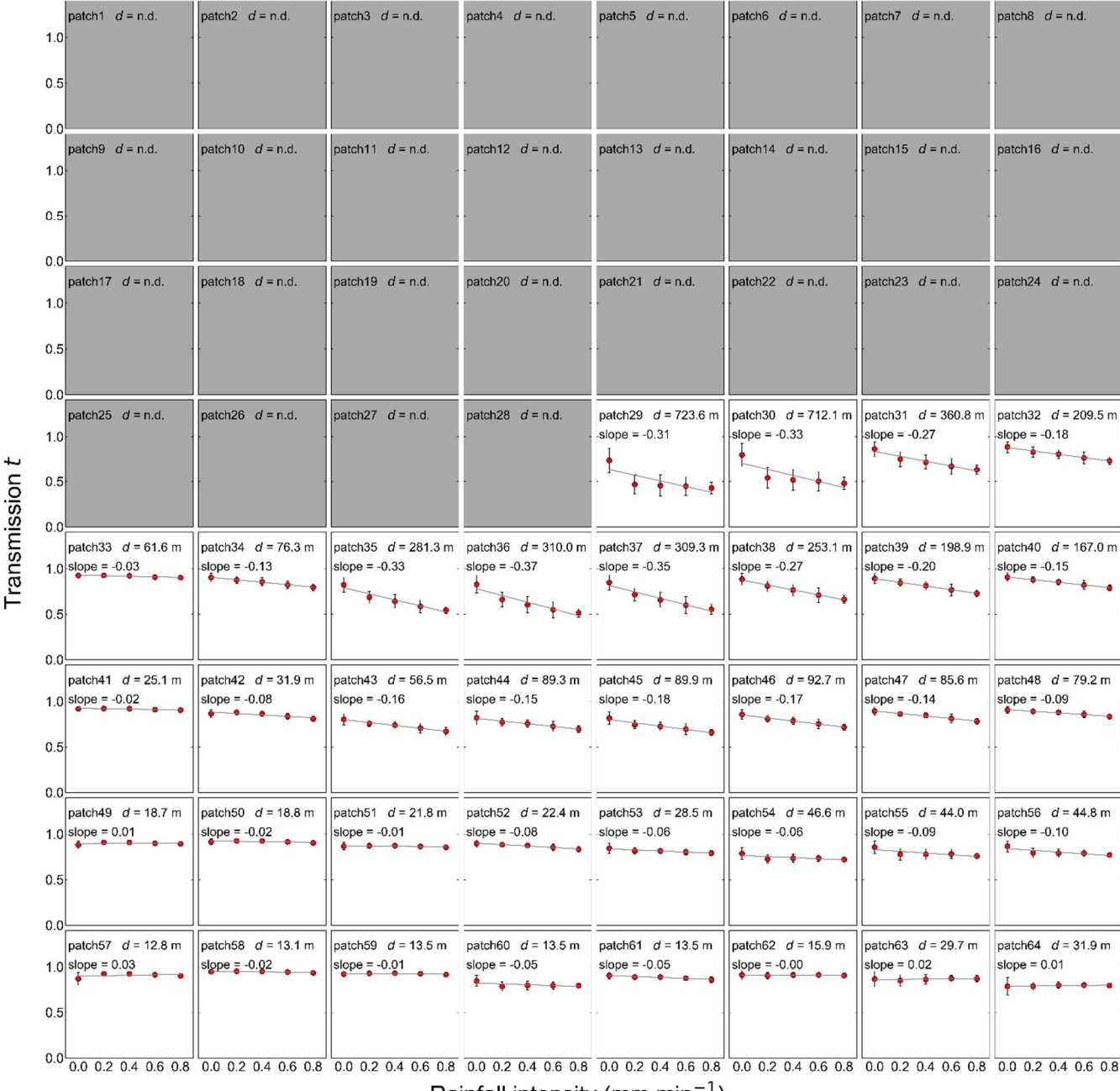

**Figure C-3-c. Distribution of transmission *t* by rainfall intensity of Camera 3.**

## Data availability

Images of all cameras and data used for analysis in this study are available at https://doi.org/10.5281/zenodo.7163149, https://doi.org/10.5281/zenodo.7166150 and https://doi.org/10.5281/zenodo.7166178.

## Supplement

Time series variations of rainfall intensity estimates for all camera patches during the three rain events as shown in 5.3 are available at https://doi.org/10.5281/zenodo.13337020.

## Author contributions

AK and TU designed the experiments, and AK and TU carried them out. AK developed the model code and analyzed the data. AK wrote the manuscript draft and TU reviewed and edited the manuscript.

## Competing interests

The authors declare that they have no conflict of interest.

## Acknowledgements

This work was supported by JSPS KAKENHI grant number 20H03019, Fujigawa Sabo Office, Kanto Reginal Bureau, Ministry of Land, Infrastructure, Transport and Tourism.

## Financial support

This work was supported by JSPS KAKENHI grant number 20H03019.

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
