# Peer review of "Rainfall intensity estimations based on degradation characteristics of images taken with commercial cameras"

_EGUsphere, 2024_

## Referee Comment (RC2)

**Title : Degradation of Commercially Available Digital Camera Images due to Variation of Rainfall Intensity in Outdoor Conditions**

The manuscript addresses commercial camera-based rainfall observation is a useful technology that contributes to the densification of rainfall observation networks. The study investigates the main and interactional effects of different commercial interval cameras (outdoor images) and rainfall intensity, which is interesting for measuring rainfall with high spatiotemporal resolution and low cost. The topic is important, and manuscript fits with the scope of the journal. but it has some weaknesses associated with the presented data and discussion have shortcomings as discussed below. Therefore, the current version of the manuscript needs major revision to be published in (HESS Journal). There are several issues must be addressed.

**Minor comments**

1. Regarding the title of manuscript, the title should show the novelty of the research and tell the main finding of the study.

The title explains the problem... For example, you could write a title like this: "The effect of fluctuation and change in rainfall intensity when using commercial cameras on the accuracy of rainfall measurement"

**Major comments**

2. For all tables and figures, no SD or SE. How the statistical analysis has been done with replicates.
   How many replicates are used for each camera? Please describe it in materials and methods section.

3. The experiment has been conducted at outdoor sitting using three commercial camera, which are the brand type and specifications of each camera separately (country of origin and description number of the device) such as "the UV visible spectrophotometer (model T80 × UVNIS Spectrometer PG Instruments Ltd, England)". This must be included in the material section. And what is the camera's shooting range (km)?

4. Details about the monthly meteorological data (wind speed, relative humidity, max and min temperature) for the experiment period are missing. Please describe it in figure…

5. As for the figures (4,5,7) and the table (2), there is very dense data in them. Please simplify the presentation of the results in a way that makes it easy for the reader to understand and grasp the information easily and without feeling any distraction.

6. The discussion section needs work. There are no comparisons with other studies. The discussion section must be rewritten in-depth highlighting the limitations of the present study.

7. As for the results section ... it shows very valuable and very important results, so it needs to be written in more detail and more clarity.

8. The research is based on how commercial cameras are used to measure rainfall and the effect of this rainfall on the measurement accuracy of each type of camera separately.... But how can we overcome the problem of the inefficiency of commercial cameras in measuring rainfall with high accuracy... Can the efficiency of the camera be improved... What is the best type of the three camera ... How can we help stakeholders in manufacturing a high-resolution surveillance camera at a low price... Please explain this

9. How to solve the problem of commercial cameras deteriorating due to increased rainfall.... please explain

10. Can farmers use a rainfall monitoring camera on their farmland to track rainfall and calculate irrigation rates efficiently.... Or will it be too expensive for them? Please clarify.

11. The research compares types of commercial cameras... please clarify which categories can benefit most from these results and apply the research results on a practical and real-world scale.

12. Please clarify at the end of the discussion section what are the weaknesses and future studies that should be conducted for improvement and to reach the best results that help in solving problems related to hydrology and rainfall.

13. References are generally very good, but they need to be expanded and cite recent research related to the research topic. (The references must be recent, as there are many articles related to this topic that were published during this period).

---

## Author Comment (AC1)

**Response to the comments of Anonymous Referee #1**

We would like to thank anonymous referee#1 for the constructive feedback on our manuscript.

Our responses to the comments are shown below.

The comments of anonymous referee #1 are shown in black. Authors' responses are shown in blue.
* * *
I've read the manuscript "Degradation of commercially available digital camera images due to variation of rainfall intensity in outdoor conditions" with interest. I find it an in-depth study that is well written. I do have a couple of suggestions though to improve (the readability) of the manuscript.

**< Comment 1 >**

- In my opinion, the title sounds a bit negative and not entirely fits the purpose of this manuscript. I suggest to rewrite the title emphasizing rainfall intensity estimation from the degradation of camera images.

Response:

We will revise the title as follows.

  "Rainfall intensity estimations based on degradation characteristics of images taken with commercial cameras"

**< Comment 2 >**

- Introduction: I think this method can be considered opportunistic sensing, which includes crowdsourcing, but the term crowdsourcing seems a bit too specific, since it does not really involve the crowd. I suggest to use the term opportunistic sensing, and to use it consistently throughout the introduction.

Response:

We will revise L. 38 through 42 as follows.

"As an initiative to overcome the issues mentioned above, techniques have been proposed to build sensors using low-cost equipment not used for its intended use and to combine a variety of not fully utilized technologies to make opportunistic observation (Tauro et al., 2018)."

And we will remove L. 96 through 97 and L. 914 through 916.

**< Comment 3 >**

- L. 50: rainfall estimation employing commercial microwave links does not use "cellular phones", although the network is used by these phones. So, remove "of cellular phones".

Response:

We will remove "of cellular phones" in this sentence.

**< Comment 4 >**

- Introduction: Give a (more) clear definition of static and dynamic weather effects. In L. 99 rain streaks may appear as fog, but the process itself is still dynamic, but considered static. This can be explained a bit more.

Response:

We will revise L. 98 through 100 as follows.

"On the other hand, even if the absolute size of raindrops is constant within the camera's angle of view, the size of raindrops in the image varies with their distance from the camera. In particular, raindrops over a certain distance from the camera induce a visual effect as if they were in static weather conditions, because their fall distance within the camera's exposure time is sufficiently small compared to the pixel size that the camera's sensor cannot detect individual raindrops. In fact, it has been pointed out that rain streaks over a certain distance from the camera accumulate on the image and appear as fog (Garg and Nayar, 2007; Li et al. 2018; Li et al., 2019)."

**< Comment 5 >**

- L. 100: define the background. Should this be seen as those cameras capturing a relatively "undisturbed" 2D image? E.g., without persons, animals and traffic moving around? So, just the river, scenery, trees? So the background is just the quite static image that is captured.

Response:

We will add and revise the following text in L. 100 through 102.

"Such raindrops over a certain distance from the camera are likely to induce static weather effects when the camera is mainly capturing a relatively undisturbed background such as rivers, scenery, trees. On the other hand, in the case of a disturbed background with people, animals, or traffic moving around, the static weather effects of raindrops may be difficult to discern because the original background may be disturbed by their movement. Thus, in an outdoor photography system that captures a relatively undisturbed background over a certain distance, not only the dynamic weather effects caused by rain but also the static weather effects caused by rain may be apparent in the images."

**< Comment 6 >**

- Section 2: despite attempts to explain "direct attenuation" and "airlight", it would help to visualize these two effects or explain them more clearly. The first seems to suggest the light going from the background to the camera, whereas the second seems direct & diffuse radiation from the sun interacting with the atmosphere (but not the background) before reaching the lens.

Response:

We will revise L. 126 through 131 as follows.

""Effects of static weather are mainly caused by two scattering phenomena: "direct attenuation" and "airlight" (Fattal, 2008; He et al., 2011; Narasimhan & Nayar, 2002, 2003; Tan, 2008). "Direct attenuation" is the attenuated light received by the camera from the background along the line of sight, caused by the scattering of light by particles such as water droplets in the atmosphere. "Direct attenuation" reduces the contrast of a scene (Tripathi & Mukhopadhyay, 2014). "Airlight" is the total amount of environmental illumination reflected into the line of sight by atmospheric particles, typically direct and diffuse radiation from the sun interacting with the atmosphere in the case of daytime outdoors. "Airlight" results in a shift in color (Tripathi & Mukhopadhyay, 2014). ""

**< Comment 7 >**

- L. 196: replace "chapters" by "sections".

Response:

We will replace "chapters" by "sections".

**< Comment 8 >**

- Section 3.1: what is the typical temporal resolution, or feasible temporal resolution, especially given rainfall retrieval processing time?

Response:

We will add the following text in L. 216.

"In estimating rainfall intensity based on camera images, it is essential to consider the instantaneous intensity at the time of shooting. In contrast, when observing rainfall using a traditional tipping bucket, it is not possible to measure rainfall until it reaches the capacity of one tipping bucket. In other words, it is difficult to measure instantaneous values with a tipping bucket rain gauge with sufficient precision. However, in this study, to validate the accuracy of rainfall intensity estimated based on camera images, we decided to obtain data from a tipping bucket rain gauge with as fine a resolution as possible (one minute)."

This method is relatively computationally inexpensive and can be processed in as little as one minute per image.    This means that the proposed method enables the acquisition of instantaneous rainfall intensity in real time for time intervals of one minute or less. As shown in our response to Comment 9, we will add the following text in 5.4.2.

"On the other hand, data processing time may be an issue in utilizing the data for real-time observations. However, the proposed method is extremely simple, requiring less than one minute to process one image using a typical commercial computer. Although we used the computer having specifications of 80 GB RAM and Intel core i7-10700 @2.90 GHz CPU in this study, such RAM capacity is not necessary for this process. In other words, it is considered that instantaneous rainfall intensity can be estimated with a time resolution of one minute or less using a typical commercial computer."

**< Comment 9 >**

- L. 210: this is an important limitation that should be mentioned in the outlook part of the conclusions.

Response:

We will add the following text in L. 679.

"Moreover, this method is not applicable to nighttime images because it was difficult to distinguish rainfall. Therefore, rainfall estimation methods using nighttime images should be also considered separately."

Furthermore, in relation to the limitation of this method, we will remove L. 611 through 626, and add a new discussion section "5. 4 Ways of forwards" as follows to clearly show the limitation, and add related paper to references.

[revised manuscript text omitted]

**< Comment 10 >**

- L. 233: why not the entire image is used, as can also be seen in Figure 2?

Response:

The calculation process of this method uses scene depth. If the entire image is used, the variation in scene depth become large. Therefore, we set an analysis region large enough to determine the scene depth. Thus, we will add following text in L. 234.

"The degradation magnitudes of the image should be related to the scene depth. If a relatively wide area is analyzed, the scene depth should vary considerably. Therefore, the limited number of pixels are set as analysis patches for each area."

**< Comment 11 >**

- Figure 3 is really helpful in clarifying and summarizing the processing chain.

Response:

We will revise Figure 3 as follows.

[Figure]

Figure 3. The flowchart of estimating rainfall intensity from image information.

**< Comment 12 >**

- L. 269: how do you recognize the "sky background"?

Response:

We did not recognize the "sky background", but we just recognize patches where the appropriate scene depth could not be calculated. Therefore, we will revise L. 269 through 271 as follows.

"Patches where the appropriate scene depth could not be calculated due to the presence of sky background and the application of geometric corrections in the image registration process, such as the upper and rightmost patch of Camera 1, were excluded from the analysis."

**< Comment 13 >**

- L. 275: "rainfall intensity and rainfall intensity" seems a typo.

Response:

We will remove "shown for each rainfall intensity" in L. 275.

**< Comment 14 >**

- Caption Figures 4-7: mention that rainfall intensity is observed by a rain gauge.

Response:

We will add the following sentence in Caption Figures 4-7.

"Rainfall intensity is observed by the rain gauge."

**< Comment 15 >**

- Figures 4, 5, 7 & Table 2: this is a lot of information and I find the figures quite difficult to read. Perhaps the figures could be enlarged and put on two pages per figure (and perhaps move to an appendix).

Response:

We will revise Figures 4, 5, 7 as follows. The revised figures show the top three patches of scene depth for each camera. We will move Figures which include all patches to an appendix. Furthermore, we will remove Table 2 and add the information in Table 2 to Figure 4, 5, 7. We will also revise Figure 6 to have the same description.

[revised manuscript text omitted]

**< Comment 16 >**

- Figures 4 - 7: what classes do the rainfall intensity values on the horizontal axis represent? E.g., 0.2 is 0 - 0.2 mm/min? And do higher values than 0.8 mm/min not occur (since the scale ends at 0.8 mm/min). In Figure 11 the scale ends at 1 mm/min.

Response:

The rainfall intensity values on the horizontal axis are the observed values from the tipping rain gauge used with a resolution of 0.2 mm as shown L. 216 through 217. During the observation period, the maximum one-minute rainfall intensity was 0.8 mm min$^{-1}$ as shown L.222. We will revise the scale ends in Figure 11.

**< Comment 17 >**

- Figure 11: using square plots would make it easier to spot correspondence and deviation between estimated and observed rainfall intensity.

Response:

We will use square plots and revise Figure 11 as follows.

[Figure]

Figure 11-a. Relationship between observed rainfall intensity and estimated rainfall intensity of Camera 1. Each figure is marked with a patch number, scene depth and two MAPE values of rainfall intensity estimates throughout the observation period. $M_{>0.2}$ means MAPE value in cases using data with observed rainfall intensity of 0.2 mm min$^{-1}$ or greater and $M_{>0.4}$ means MAPE value in cases using data with observed rainfall intensity of 0.4 mm min$^{-1}$ or greater. The plots and error bars show the mean value and standard deviation of all data during the observation period. Patches hatched in yellow indicate patches with scene depth of less than 100 m. Patches hatched in gray are patches where the appropriate scene depth could not be obtained due to the presence of sky background and the application of geometric corrections in the image registration process.

[Figure]

Figure 11-b. Relationship between observed rainfall intensity and estimated rainfall intensity of Camera 2. Each figure is marked with a patch number, scene depth and two MAPE values of rainfall intensity estimates throughout the observation period. $M_{>0.2}$ means MAPE value in cases using data with observed rainfall intensity of 0.2 mm min$^{-1}$ or greater and $M_{>0.4}$ means MAPE value in cases using data with observed rainfall intensity of 0.4 mm min$^{-1}$ or greater. The plots and error bars show the mean value and standard deviation of all data during the observation period. Patches hatched in yellow indicate patches with scene depth of less than 100 m. Patches hatched in gray are patches where the appropriate scene depth could not be obtained due to the presence of sky background and the application of geometric corrections in the image registration process.

[Figure]

Figure 11-c. Relationship between observed rainfall intensity and estimated rainfall intensity of Camera 3. Each figure is marked with a patch number, scene depth and two MAPE values of rainfall intensity estimates throughout the observation period. $M_{>0.2}$ means MAPE value in cases using data with observed rainfall intensity of 0.2 mm min$^{-1}$ or greater and $M_{>0.4}$ means MAPE value in cases using data with observed rainfall intensity of 0.4 mm min$^{-1}$ or greater. The plots and error bars show the mean value and standard deviation of all data during the observation period. Patches hatched in yellow indicate patches with scene depth of less than 100 m. Patches hatched in gray are patches where the appropriate scene depth could not be obtained due to the presence of sky background and the application of geometric corrections in the image registration process.

**< Comment 18 >**

- Adding a scatter density plot with metrics, or tables with metrics, such as relative bias in the mean and Pearson correlation coefficient (e.g., by expanding Table 4), would provide more insight into the performance of camera-based rainfall estimation.

A lot of time and figures are spend on the underlying relationships (e.g., Figures 4-7), and relatively little space is reserved for the evaluation of the ultimate goal: rainfall estimation.

Response:

We will revise Table 4 as follows.

The revised Table 4 shows mean, maximum, 75 percentile, median, 25 percentile and minimum values of MAPE over all patches and patches with scene depth of more than 100 m. Furthermore, we will also add MAPE value to Figure 11 for each patch. We will explain these results in response to Comment 21.

Table 4. Comparison of accuracy between five previous studies and this study: The "City, Country" row indicates the city and country where each study was conducted, and the "Köppen climate classification" row indicates the Köppen climate classification of the city. In the "Used data" row, "> 0.2 mm min$^{-1}$" means that data with observed rainfall intensity of 0.2 mm min$^{-1}$ or greater were used, and "> 0.4 mm min$^{-1}$" means that data with observed rainfall intensity of 0.4 mm min$^{-1}$ or greater were used. The "Observed rainfall intensity" row indicates the mean, maximum and minimum rainfall intensity during the observation period for each study. The range of "Observed rainfall intensity" indicates the range due to multiple rainfall events in each study. The rainfall intensity of Jiang et al. (2019) was converted from the duration and the accumulated rainfall of the rainfall event. MAPE is the mean absolute percentage error, and data with observed rainfall intensity of 0 mm min$^{-1}$ were excluded by the definition of MAPE. MAPE values are shown for the case where all patches were used and for the case where only patches with a scene depth of more than 100 m were used. The number of patches used for each camera is shown in parentheses. In the column of the minimum MAPE, the patch number indicating the minimum value is shown in parentheses.

| | | This study | | | | | | Allamano et al. (2015) | Dong et al. (2017) | Jiang et al. (2019) | Yin et al. (2023) | Zheng et al. (2023) |
|---|---|---|---|---|---|---|---|---|---|---|---|---|
| City, Country | | Yamanashi, Japan | | | | | | Torino, Italy | Nanjing, China | Shenzhen, China | Hangzhou, China | Hangzhou, China |
| Köppen climate classification | | Humid subtropical climate (Cfa) | | | | | | Humid subtropical climate (Cfa) | Humid subtropical climate (Cfa) | Monsoon influenced humid subtropical climate (Cwa) | Humid subtropical climate (Cfa) | Humid subtropical climate (Cfa) |
| Used data | | > 0.2 mm min$^{-1}$ | | | > 0.4 mm min$^{-1}$ | | | - | - | - | - | - |
| Camera name | | Camera 1 | Camera 2 | Camera 3 | Camera 1 | Camera 2 | Camera 3 | - | - | - | - | - |
| Data size for validation: Video length (min) | | 3261 | 3015 | 3261 | 120 | 107 | 120 | 104 | 9 | 403 | 170 | 357 |
| Observed rainfall intensity (mm h$^{-1}$) | Mean | 12.6 | 12.6 | 12.6 | 28.5 | 28.9 | 28.5 | 2.8 - 9.3 | 1.1 - 6.5 | 4.1 - 29.5 | 11.0 - 23.6 | 9.7 - 39.3 |
| | Maximum | 48.0 | 48.0 | 48.0 | 48.0 | 48.0 | 48.0 | 6.0 - 38.2 | - | - | 36.0 - 66.0 | 24.0 -156.0 |
| | Minimum | 12.0 | 12.0 | 12.0 | 24.0 | 24.0 | 24.0 | 1.3 - 3.2 | - | - | - | - |
| Accuracy: MAPE (%) — All patches (Camera 1: 37 patches, Camera 2: 53 patches, Camera 3: 36 patches) | Mean | 1163.4 | 2131.4 | 1087.2 | 459.7 | 923.8 | 466.3 | 26.0 | 31.8 | 21.8 | 13.5 - 21.9 | 10.4 - 18.0 |
| | Maximum | 5981.9 | 12290.1 | 2178.9 | 2823.4 | 5474.7 | 951.9 | | | | | |
| | 75 percentile | 783.5 | 1989.6 | 1296.7 | 333.2 | 853.9 | 565.7 | | | | | |
| | Median | 170.1 | 242.2 | 546.6 | 66.1 | 111.7 | 237.1 | | | | | |
| | 25 percentile | 55.6 | 95.9 | 62.2 | 32.4 | 36.9 | 49.9 | | | | | |
| | Minimum | 39.1 (patch 42) | 41.7 (patch 29) | 36.0 (patch 39) | 22.5 (patch 37) | 25.3 (patch 28) | 28.6 (patch 48) | | | | | |
| Accuracy: MAPE (%) — Patches with scene depth of more than 100 m (Camera 1: 21 patches, Camera 2: 23 patches, Camera 3: 10 patches) | Mean | 88.9 | 148.3 | 47.1 | 43.1 | 70.6 | 44.3 | | | | | |
| | Maximum | 219.0 | 829.4 | 75.2 | 111.8 | 408.1 | 56.9 | | | | | |
| | 75 percentile | 123.8 | 172.5 | 46.1 | 54.4 | 83.2 | 52.5 | | | | | |
| | Median | 65.0 | 96.5 | 41.3 | 35.1 | 38.7 | 45.9 | | | | | |
| | 25 percentile | 47.7 | 58.0 | 40.0 | 28.4 | 33.3 | 36.4 | | | | | |
| | Minimum | 39.1 (patch 42) | 41.7 (patch 29) | 36.0 (patch 39) | 22.5 (patch 37) | 25.3 (patch 28) | 29.3 (patch 35) | | | | | |

**< Comment 19 >**

- Table 4 and discussion of other studies: are metrics computed in the same way across all these studies, i.e., with the same or no threshold? And are there differences in climatology that may lead to differences between studies?

Response:

As shown in our response to Comment 18, we will revise Table 4. The revised Table 4 shows the city and country where each study was conducted, the Köppen climate classification of the city, and the mean, maximum and minimum rainfall intensity during the observation period for each study. Table 4 shows that although only Jiang et al. (2019) was conducted in Monsoon influenced humid subtropical climate, this study and all five previous studies were conducted in a humid subtropical climate, and that there are no significant climatic differences. Furthermore, regarding the mean rainfall intensity during the observation period, the mean rainfall intensities of Allamano et al. (2015) and Dong et al. (2017) were lower than those in this study, while the mean rainfall intensities of Jiang et al. (2019), Yin et al. (2023), and Zheng et al. (2023) were comparable to this study. Overall, there is no significant difference in mean rainfall intensity between this study and the five previous studies. Moreover, in all studies, MAPE, a metrics of model performance, was calculated from observed values and model prediction. Given these facts, it seems reasonable to compare this study with five previous studies.

Therefore, we will add the following text in L. 603.

"Table 4 shows that although only Jiang et al. (2019) was conducted in Monsoon influenced humid subtropical climate, this study and all five previous studies were conducted in a humid subtropical climate, and that there are no significant climatic differences. Furthermore, regarding the mean rainfall intensity during the observation period, the mean rainfall intensities of Allamano et al. (2015) and Dong et al. (2017) were slightly lower than those in this study, while the mean rainfall intensities of Jiang et al. (2019), Yin et al. (2023), and Zheng et al. (2023) were comparable to this study. Overall, there is no significant difference in mean rainfall intensity between this study and the five previous studies. Moreover, in all studies, MAPE, a metrics of model performance, was calculated from observed values and model prediction. Given these facts, it seems reasonable to compare this study with five previous studies."

**< Comment 20 >**

- Table 4: From the data size I conclude that your study encompasses a much longer dataset. That could be emphasized in the introduction, since this stands out.

Response:

We will add the following sentence in L. 119.

"The estimation of rainfall intensity used over 3,000 images from rainfall events, and this data size is a unique aspect of this study."

**< Comment 21 >**

- Table 4: You honestly describe that you selected the values for the path with the lowest MAPE for each camera. What is the performance when the MAPE values over all patches are used? You could, e.g., provide the median and mean values found for MAPE, or just compute the MAPE over all patches. And did the other studies also select the patches with the highest performance?

Response:

As shown in our response to Comment 18, we will revise Table 4. And we will revise L. 603 through 607 as follows.

"All five of these studies did not separate the patches. This is because these previous studies focused on the dynamic weather effects and scene depth was not relevant. As shown in Table 4, the mean value of MAPE using data with observed rainfall intensity of 0.2 mm min$^{-1}$ or greater for all patches was 1163.4 %, 2131.4 %, and 1087.2 % for Camera 1, Camera 2, and Camera 3, respectively, and the median value of MAPE using data with observed rainfall intensity of 0.2 mm min$^{-1}$ or greater

for all patches was 170.1 %, 242.2 %, and 546.6 % for Camera 1, Camera 2, and Camera 3, respectively. The mean value of MAPE was considerably larger than the median value of MAPE, because it was heavily influenced by larger values such as the maximum value of MAPE. On the other hand, the mean value of MAPE using data with observed rainfall intensity of 0.2 mm min$^{-1}$ or greater for patches with scene depth of more than 100 m was 88.9 %, 148.3 %, and 47.1 % for Camera 1, Camera 2, and Camera 3, respectively, and the median value of MAPE using data with observed rainfall intensity of 0.2 mm min$^{-1}$ or greater for all patches was 65.0 %, 96.5 %, and 41.3 % for Camera 1, Camera 2, and Camera 3, respectively. Thus, the results indicate that the accuracy of rainfall intensity estimation can be improved by restricting the data to patches with scene depth of more than 100 m. Therefore, it is important to select patches with a scene depth of more than 100 m for rainfall intensity estimation. Next, we compare results of patches with scene depth of more than 100 m in this study with results of five previous studies. The median, 25th percentile, and minimum value of MAPE using data with observed rainfall intensity of 0.2 mm min$^{-1}$ or greater for patches with scene depth of more than 100 m was higher than the MAPE value in the five previous studies. In contrast, the median value of MAPE using data with observed rainfall intensity of 0.4 mm min$^{-1}$ or greater for patches with scene depth of more than 100 m was slightly higher than the MAPE value in the five previous studies, but the 25th percentile and minimum value of MAPE using data with observed rainfall intensity of 0.4 mm min$^{-1}$ or greater for patches with scene depth of more than 100 m was similar to those of the five studies. Therefore, the proposed method in this study is considered to have a certain degree of effectiveness as a rainfall intensity estimation method, although there may be some error when the rainfall intensity is small."

**< Comment 22 >**

- Particle size distribution is known to vary between rainfall types, and hence between climates. How representative is the used distribution for your study? What do you expect from the suitability of your method for other climates, e.g., tropical climates in the Global South, where often few ground-based observations are available?

Response:

As shown in L. 706 through 709, the particle size distribution we used is the Marshall and Palmer distribution. We are thinking that the Marshall and Palmer distribution is a very good approximation to the raindrop size distribution referred to natural rainfall and widely used for describing the midlatitude particle size distribution that are characterized by low to moderate intensity (e.g., Serio et al. (2019)). However, when using the particle size distribution different from the Marshall and Palmer distribution, the change of the value of $N_0$ and $\lambda$ may lead to the change in the value of extinction coefficient $\beta$, resulting in an overestimation or underestimation of rainfall intensity estimate.

Therefore, we will add following text in L. 180.

"As shown in Appendix A, the particle size distribution of raindrops used in this study is that presented by Marshall and Palmer (1948), hereafter referred to as the M-P distribution. The M-P distribution is a very good approximation to the raindrop size distribution referred to natural rainfall and widely used for describing the midlatitude particle size distribution that are characterized by low to moderate intensity (e.g., Serio et al., 2019). However, particle size distribution is known to vary between rainfall types and climates. Therefore, it should be noted that when using the particle size distribution different from the M-P distribution, the change of the value of $N_0$ and $\lambda$ may lead to the change in the value of extinction coefficient $\beta$, resulting in an overestimation or underestimation of rainfall intensity estimate."

Also, we will add following paper to references.
- Serio, M. A., Carollo, F. G., and Ferro, V.: Raindrop size distribution and terminal velocity for rainfall erosivity studies. A review, J. Hydrol., 576, 210–228, https://doi.org/10.1016/j.jhydrol.2019.06.040, 2019.

**< Comment 23 >**

- L. 613: How to select an appropriate background in urban areas? Any suggestions, e.g., buildings?

Response:

We will revise L. 613 through 614 as follows and also refer to in our response to comment 9.

"Therefore, it is preferable to choose a static background such as building walls, tree canopies, and ground surface without people or vehicles, especially when applying this method in urban areas."

**< Comment 24 >**

- L. 670: Would this method be generally applicable to the abundant webcam or video images around the globe? Would it in principle be possible to obtain all the necessary data by simply downloading images from public websites? And could this technique potentially become a gap filler for (tropical) areas in the Global South lacking surface rainfall observations?

Response:

As shown in L. 671 through 675, this method only uses the camera image taken of the background over a certain distance and background scene depth information. Therefore, if they are available, it would be generally applicable to the abundant webcam or video images around the globe. As shown in L. 674 through 675, it would be possible to obtain the scene depth information measuring distances in a GIS. Of course, this means that this technique potentially becomes a gap filler for areas in the Global South lacking surface rainfall observations. Therefore, we will add the following sentence in L. 675 and in 5.4.2. as shown in our response to Comment 9.

"Therefore, this technique potentially become a gap filler for areas in lacking surface rainfall observations."

**< Comment 25 >**

- L. 670 & L. 678: what is the difference between "a single static image" and "a single individual image"? I find it a bit confusing to read that the method cannot be applied to a single individual image, but is applied to a single static image. Do you mean that the method can be applied to a single image, but that a sequence of images is needed to perform all necessary processing?

Response:

This method can be applied to a single individual static image. But the text in L. 677-679 was not well expressed. This text was intended to say that the overall trend in the applicability of the method was analyzed in this study, but the specific causes of the errors in each individual image were not validated. Therefore, we will revise L. 677-679 as follows. Note that this revision also includes a response to comment 26.

"Furthermore, this study examined the overall trend in the applicability of the method across the entire data set, but the specific causes of the errors in each individual image were not validated. For example, the presence of dew formation and raindrops on the camera lens itself could cause significant blurriness on the image and affect the rainfall estimation results, but this was not validated in this study. Therefore, validation of the specific causes of the errors when the proposed method is applied to each individual image is an issue to be addressed in the future."

**< Comment 26 >**

- What about dew formation (could be due to fog) and rain drops on the camera lens itself? Couldn't this cause significant blurriness, which could be interpreted as extinction? Or is the lens protected to rain drops by some cover above the lens? This could be mentioned around L. 677 or discussed in the Discussion section.

Response:

Dew formation and raindrops on the camera lens itself could cause significant blurring of the image and affect the rainfall estimation results. Therefore, we have included a revised version in our response to Comment 9 and 25.

**< Comment 27 >**

- You end the conclusions with the prospect of upscaling this technique. Would privacy issues, e.g., persons on the image, pose an obstacle for this rainfall estimation technique?

Response:

In many outdoor surveillance cameras, it may be inevitable that persons will be captured. Therefore, when making data public, it is necessary to pay careful attention to privacy issues.

We will add the following text in 5.4.2. as shown in our response to Comment 9.

"Furthermore, there are concerns about privacy issues in the actual use of this method. In many outdoor surveillance cameras, it may be inevitable that persons will be captured. Therefore, when making data public, it is necessary to pay careful attention to privacy issues."

**< Comment 28 >**

- What are the prospects for differentiating between precipitation types (e.g., rain, snow) and fog? This entails knowing which precipitation type (or fog) occurs, which can be highly relevant for early warnings (i.e., not focussed on intensity but only on type), but also improving rainfall estimates, by not taking into account other precipitation types and fog.

Response:

As you have pointed out, we consider differentiating between rain and fog a very important problem. At present, however, there is no method to determine whether it is rain or fog. Therefore, as a further study, it is necessary to investigate a method to determine whether the whiteness in the image under bad weather conditions is caused by rain or fog.

We will add the following text in 5.4.1. as shown in our response to Comment 9.

"Since this method estimates rainfall intensity from image whiteness, image whiteness caused by fog is misidentified as the effect of rainfall. At present, however, there is no method to determine whether it is fog or rain. Therefore, as a further study, it is necessary to investigate a method to determine whether the whiteness in the image under bad weather conditions is caused by rain or fog."

---

## Author Comment (AC2)

**Response to the comments of Anonymous Referee #2**

We would like to thank anonymous referee#2 for the constructive feedback on our manuscript.

Our responses to the comments are shown below.

The comments of anonymous referee #2 are shown in black. Authors' responses are shown in blue.
* * *
The manuscript addresses commercial camera-based rainfall observation is a useful technology that contributes to the densification of rainfall observation networks. The study investigates the main and interactional effects of different commercial interval cameras (outdoor images) and rainfall intensity, which is interesting for measuring rainfall with high spatiotemporal resolution and low cost. The topic is important, and manuscript fits with the scope of the journal. but it has some weaknesses associated with the presented data and discussion have shortcomings as discussed below. Therefore, the current version of the manuscript needs major revision to be published in (HESS Journal). There are several issues must be addressed.

**Minor comments**

**< Comment 1 >**

- Regarding the title of manuscript, the title should show the novelty of the research and tell the main finding of the study. The title explains the problem... For example, you could write a title like this: "The effect of fluctuation and change in rainfall intensity when using commercial cameras on the accuracy of rainfall measurement"

Response:

We will revise the title as follows because the original title was unclear.

"Rainfall intensity estimations based on degradation characteristics of images taken with commercial cameras"

**Major comments**

**< Comment 2 >**

- For all tables and figures, no SD or SE. How the statistical analysis has been done with replicates. How many replicates are used for each camera? Please describe it in materials and methods section.

Response:

We will revise Figures 4, 5, 6, 7 and 11 with SD as follows. The revised Figures 4, 5, and 7 show the top three patches of scene depth for each camera. In terms of Figures 4, 5, and 7, we will move Figures which include all patches to an appendix C-1, C-2, and C-3.

As shown Table 1, there are multiple images at each rainfall intensity for each camera. SDs were calculated from these images. We will describe this in the captions of all figures with SD.

[revised manuscript text omitted]

**< Comment 3 >**

- The experiment has been conducted at outdoor sitting using three commercial camera, which are the brand type and specifications of each camera separately (country of origin and description number of the device) such as "the UV visible spectrophotometer (model T80 × UVNIS Spectrometer PG Instruments Ltd, England)". This must be included in the material section. And what is the camera's shooting range (km)?

Response:

We will revise L.203 through 204 as follows.

"Photography was taken using three commercially available interval cameras (TLC200Pro Brinno inc., Taiwan)."

We consider shooting range and focus distance to mean almost the same thing and it is from 40 cm to infinity as shown L. 206.

We will also revise L. 215 to have the same description about a tipping bucket rain gauge used as follows.

"One-minute rainfall intensity was also observed using a tipping bucket rain gauge (RG3-M Onset Computer Corporation, USA) at almost the same locations where the cameras were installed."

**< Comment 4 >**

- Details about the monthly meteorological data (wind speed, relative humidity, max and min temperature) for the experiment period are missing. Please describe it in figure…

Response:

We will add the following figure and text in L. 203.

"The meteorological observations in 2021 around the observation site are shown in Figure **. The figure shows data from a weather station about 24 km southeast of those cameras. The Köppen climate classification of the area around the observation site is humid subtropical climate, with hot, humid and heavy precipitation in summer and cool to mild in winter."

[Figure]

Figure **. The meteorological observations in 2021 around the observation site.

**< Comment 5 >**

- As for the figures (4,5,7) and the table (2), there is very dense data in them. Please simplify the presentation of the results in a way that makes it easy for the reader to understand and grasp the information easily and without feeling any distraction.

Response:

As shown in our response to Comment 2, we will revise Figures 4, 5, and 7. The revised Figures 4, 5, and 7 show the top three

patches of scene depth for each camera. In terms of Figures 4, 5, and 7, we will move Figures which include all patches to an appendix C-1, C-2, and C-3. Furthermore, we will remove Table. 2 and add the information in Table 2 to Figure 4, 5, and 7. We will also revise Figure 6 to have the same appearance as shown in our response to Comment 2.

**< Comment 6 >**

- As for the results section... it shows very valuable and very important results, so it needs to be written in more detail and more clarity.

Response:

We will revise the results section 4.1 as follows. As shown in response to Comment 5, we will revise Figures 4 ,5 ,6 and 7 for better clarity.

[revised manuscript text omitted]

In addition, we will add the following sentence in L.325.
"As described in 4.1, the fact that scene depth and rainfall intensity may interrelate to determine the extent of decrease in transmission $t$ is also in the same sense."

**< Comment 7 >**

- The discussion section needs work. There are no comparisons with other studies. The discussion section must be rewritten in-depth highlighting the limitations of the present study.

Response:

In terms of the validity of calculated extinction coefficient $\beta$, we compared our results to other studies as shown in 5. 2. 3, and 5. 2. 4. Furthermore, in terms of the accuracy of estimated rainfall intensity, we compared our results to other previous studies as shown in Table 4. We will revise Table 4 as follows. The revised Table 4 shows mean, maximum, 75 percentile, median, 25 percentile and minimum values of MAPE over all patches and patches with scene depth of more than 100 m.

[revised manuscript text omitted]

Therefore, we will add the following text in L. 603.
"Table 4 shows that although only Jiang et al. (2019) was conducted in Monsoon influenced humid subtropical climate, this

study and all five previous studies were conducted in a humid subtropical climate, and that there are no significant climatic differences. Furthermore, regarding the mean rainfall intensity during the observation period, the mean rainfall intensities of Allamano et al. (2015) and Dong et al. (2017) were slightly lower than those in this study, while the mean rainfall intensities of Jiang et al. (2019), Yin et al. (2023), and Zheng et al. (2023) were comparable to this study. Overall, there is no significant difference in mean rainfall intensity between this study and the five previous studies. Moreover, in all studies, MAPE, a metrics of model performance, was calculated from observed values and model prediction. Given these facts, it seems reasonable to compare this study with five previous studies."

In addition, we will rewrite the limitations of the present study in the discussion section as shown in our response to Comment 12.

< **Comment 8** >

- The research is based on how commercial cameras are used to measure rainfall and the effect of this rainfall on the measurement accuracy of each type of camera separately.... But how can we overcome the problem of the inefficiency of commercial cameras in measuring rainfall with high accuracy... Can the efficiency of the camera be improved... What is the best type of the three camera ... How can we help stakeholders in manufacturing a high-resolution surveillance camera at a low price... Please explain this

Response:

It is certainly inefficient to install new cameras for rainfall observation. However, as shown in L. 105 through 106, the intention of this study is to efficiently use images from cameras already installed for other purposes to measure rainfall intensity. All three cameras used in this study are the same model. These cameras are commercially available for approximately 300 US dollars per unit and are relatively accessible to everyone.

< **Comment 9** >

- How to solve the problem of commercial cameras deteriorating due to increased rainfall.... please explain

Response:

As shown in our response to Comment 1, we will revise the title because the original title was unclear. This study did not discuss camera performance degradation to increased rainfall.

< **Comment 10** >

- Can farmers use a rainfall monitoring camera on their farmland to track rainfall and calculate irrigation rates efficiently.... Or will it be too expensive for them?   Please clarify.

Response:

As shown in L. 672, this method requires only camera image taken of the background over a certain distance and background scene depth information. The camera we used is a commercial camera costing approximately 300 US dollars per unit and is not so expensive. Therefore, farmers can easily use those cameras and monitor rainfall.

**< Comment 11 >**

- The research compares types of commercial cameras... please clarify which categories can benefit most from these results and apply the research results on a practical and realworld scale.

Response:

As shown in L. 27 through 29, in mountainous areas where flash floods and debris flow occur, rainfall should be measured on fine spatial and temporal scales for effective early warning against these disasters. In such areas, even if rain gauges are not installed, monitoring cameras may be in place. This study attempts to observe rainfall by effectively utilizing such cameras already installed for other purposes. We expect that our research results can be applied on a practical and realworld scale in the category of disaster prevention.

Therefore, we will revise L. 681 through 684 as follows.

"Especially, in mountainous areas where flash floods and debris flow occur, for countermeasures against these disasters, it is desirable to have information on rainfall with high spatio-temporal resolution. In such areas, even if rain gauges are not installed, monitoring cameras may be in place. This study attempts to observe rainfall by effectively utilizing such cameras already installed for other purposes. We expect that our research results can be applied on a practical and realworld scale in such category of disaster prevention."

**< Comment 12 >**

- Please clarify at the end of the discussion section what are the weaknesses and future studies that should be conducted for improvement and to reach the best results that help in solving problems related to hydrology and rainfall.

Response:

We will remove L. 611 through 626, and add a new discussion section "5. 4 Ways of forwards" as follows to clearly show the weaknesses and future studies that should be conducted for improvement. Furthermore, we will add related paper to references.

[revised manuscript text omitted]

**< Comment 13 >**

- References are generally very good, but they need to be expanded and cite recent research related to the research topic. (The references must be recent, as there are many articles related to this topic that were published during this period).

Response:

We will cite some recent related research and add them to the references as follows.

Firstly, we will add the following sentence in L. 88.

"A lot of deep machine learning-based methods have been proposed in recent years as with the trend of studies about static weather effect (e.g., Lin et al., 2020; Lin et al., 2022; Yin et al., 2023; Zheng et al., 2023)."

[Related papers to add to the references]

- Lin, C. W., Lin, M. X., and Yang, S. H.: SOPNet Method for the Fine-Grained Measurement and Prediction of Precipitation Intensity Using Outdoor Surveillance Cameras, IEEE Access, 8, 188813–188824, https://doi.org/10.1109/Access.2020.3032430, 2020.

- Lin, C. W., Huang, X., Lin, M., and Hong, S.: SF-CNN: Signal Filtering Convolutional Neural Network for Precipitation Intensity Estimation, Sensors, 22(2), 511, https://doi.org/10.3390/s22020551, 2022.

Secondly, we will add the following sentence in L. 95.

"In particular, it has been pointed out that the limitation of deep machine learning-based methods is the lack of training data rather than the design of network structure and learning manners (Wang et al., 2021; Yan et al., 2023)."

[Related papers to add to the references]

- Wang, H., Yue, Z. S., Xie, Q., Zhao, Q., Zheng, Y., and Meng, D.: From Rain Generation to Rain Removal, Proc. CVPR. IEEE, 14786–14796, 2021

- Yan, K., Chen, H., Hu, L., Huang, K., Huang, Y., Wang, Z., Liu, B., Wang, J., and Guo, S.: A review of video-based rainfall measurement methods, WIREs Water, 2023;10:e1678., https://doi.org/10.1002/wat2.1678, 2023.

Thirdly, we will mention methods using infrared and near-infrared cameras to estimate rainfall intensity at night in recent years as shown in response to Comment 12.

[Related papers to add to the references]

- Lee, J., Byun, J., Baik, J., Jun, C., and Kim, H. J.: Estimation of raindrop size distribution and rain rate with infrared surveillance camera in dark conditions, Atmos. Meas. Tech., 16, 707–725, https://doi.org/10.5194/amt-16-707-2023, 2023.

- Wang, X., Wang, M., Liu, X., Zhu, L., Shi, S., Glade, T., Chen, M., Xie, Y., Wu, Y., and He, Y.: Near-infrared surveillance video-based rain gauge, J. Hydrol., 618, 129173, https://doi.org/10.1016/j.jhydrol.2023.129173, 2023.

---

## Author Response (AR2)

**Response to the comments of Anonymous Referee #1**

We would like to thank anonymous referee#1 for the constructive feedback on our manuscript.

Our responses to the comments are shown below.

The comments of anonymous referee #1 are shown in black. Authors' responses are shown in blue.
* * *
**< Comment 1 >**

You have seriously replied to all questions and comments and provided an extensive rebuttal including a thorough revision of your manuscript. I only have one remaining remark concerning the particle size distribution of rain drops. Although this is described in Marshall and Palmer (1948), in case of the Z-R relationship between radar reflectivity and rain rate, the coefficients (200 & 1.6) probably should be obtained from: Marshall, J. S., W. Hitschfeld, and K. L. S. Gunn, 1955: Advances in radar weather. Advances in Geophysics, Academic Press, New York, Vol. 2, 1–56. Note that many different coefficients for the Z-R (radar reflectivity factor - rain rate) relationship exist in the literature, some of them also for other than temperate climates, such as for tropical rain. So the use of the M-P ($Z = 200 * R^{1.6}$) relationship has its limitations for other than temperature climates, but the Z-R relationship itself can be applied in other climates preferably with appropriate coefficients for that climate. Here, you do not use the Z-R relationship, but since you are referring to the drop size distribution from Marshall and Palmer, a similar reasoning can be followed for your specific computations.

Response:

We will not refer to the Z-R relationship itself here because we do not use the Z-R relationship, but we will describe the limitation of the use of the M-P distribution and add the proposed paper to references. Therefore, we will revise L. 185 through 186 and L. 193 through 195 as follows and add the following paper to references.

L. 185 through 186

"Since the particle size distribution of raindrops is known to be related to rainfall intensity (Marshall and Palmer, 1948; Marshall et al., 1955), the extinction coefficient can be expressed using rainfall intensity as follows."

L. 193 through 195

"Therefore, the use of the M-P distribution has limitations for other rainfall types and climates. It should be noted that when Eq. (6) is used under different rainfall types and climates conditions than those under which the M-P distribution is applied, the appropriate coefficients for other rainfall types and climates should be used."

[The paper to add to the references]

- Marshall, J. S., Hitschfeld, W., and Gunn, K. L. S.: Advances in radar weather, Adv. Geophys., 2, 1–56, https://doi.org/10.1016/S0065-2687(08)60310-6, 1955.